# Effects of thymidylate synthase inhibitors differ in genomic uracilation and mutagenic potential

Eszter Holub[1,2,3] , Gábor Papp[1] , Hajnalka Laura Pálinkás[1,2], Milda Blanka Szajkó[1,2], Richard Izrael[2], Gergely Róna[4,5] , Beáta G Vértessy[1,2] , Angéla Békési[1,2]

Genomic uracil and its respective repair play key roles in colorectal, gastric, and other solid tumor therapies targeting thymidylate biosynthesis. Previously, we established that treating HCT116 colon cancer cell lines with either raltitrexed (RTX) or 5-fluoro-2′-deoxyuridine (5FdUR), two potent inhibitors of thymidylate synthase, results in characteristic genomic uracil patterns. Here, we focus on drug-specific differences in the genomic uracil profiles and their associations with altered cytotoxicity and drug-induced mutagenesis. We demonstrated that biased uracilation preferentially affects functionally related genes in a drug-specific manner, highlighting the biological significance of genomic uracilation. Mutational analysis further revealed a significant increase in the frequency of C-to-T somatic transitions, selectively in response to high-dose 5FdUR treatment, in DNA repair-deficient cells. The mutational spectra and the clustered nature of these transitions suggested the involvement of APOBEC3 DNA cytidine deaminases, several of which were induced under these conditions. Notably, this mutagenic response coincides with decreased cytotoxicity compared with the low-dose 5FdUR or any efficient doses of RTX, providing insights that may be relevant for personalized cancer therapy.

## Introduction

Inhibition of thymidylate biosynthesis has been widely applied as an anticancer strategy for decades (Heidelberger et al, 1957; Wilson et al, 2014), leading to thymidine depletion and diverse cellular responses. The phenomenon known as thymine-less cell death has been extensively studied in both prokaryotes and eukaryotic cell lines, revealing dNTP imbalance, increased frequencies of single- and double-strand DNA breaks, induced mutagenesis, and recombinational repair (reviewed in Ahmad et al [1998] and Berger et al [2008]). Thymidylate synthase (TS) inhibition was initially thought to cause lethality via a perturbed dUTP/dTTP ratio, leading to uracil incorporation into the genome of actively dividing cells and consequent DNA strand breaks via hyperactivation of uracil-DNA repair mechanisms (Webley et al, 2001; Li et al, 2005; Wyatt & Wilson, 2009; Martín & Guzmán, 2011) followed by DNA damage signaling (Yang et al, 2008). However, when uracil–DNA repair is impaired, TS inhibitors still exert cytostatic effects, most probably because of interference between genomic uracil- and DNA-based regulatory mechanisms (Andersen et al, 2005; Luo et al, 2008; Christenson et al, 2021). Previously, in HCT116 human colon cancer cells expressing a protein inhibitor (UGI) of the main uracil-DNA glycosylase (UNG), we detected massive uracil incorporation with characteristic genomic distributions in response to treatments with TS inhibitors 5-fluoro-2′-deoxyuridine (5FdUR) and raltitrexed (RTX). These cells also exhibited decreased viability coupled with strong S-phase arrest (Pálinkás et al, 2020).

Thymidine deprivation-induced mutagenicity has been described in prokaryotes as a result of imbalanced dNTP pools, uracil incorporation and repair, and increased recombination (Kunz & Glickman, 1985). Because uracil incorporation replacing thymine is not mutagenic per se, the induced mutagenicity must arise indirectly through a complex interplay among uracil-DNA glycosylases, downstream repair factors, and DNA damage signaling. Interestingly, in yeast—where UNG is the only uracil-DNA glycosylase—the frequency of point mutations was not increased, yet sister chromatid exchange was observed upon thymidylate synthase inhibition (Kunz et al, 1986). In cancer cells, treatments with fluoropyrimidine derivatives caused the incorporation of 5-fluorouracil (5FU), resulting in 5FU:G mismatches (Meyers et al, 2005), which can also contribute to additional mutational burden, especially in mismatch repair (MMR)-deficient tumor cells.

As an additional mechanism of therapy-induced mutagenesis, a more recent study reported that chemotherapeutic agents (including 5FU) can induce the expression of APOBEC3 cytidine deaminases

[1]Department of Applied Biotechnology and Food Science, Faculty of Chemical Technology and Biotechnology, Budapest University of Technology and Economics, Budapest, Hungary   [2]Genome Metabolism Research Group, Institute of Molecular Life Sciences, HUN-REN Research Centre for Natural Sciences, Budapest, Hungary   [3]Doctoral School of Biology, Institute of Biology, ELTE Eötvös Loránd University, Budapest, Hungary   [4]MTA-HUN-REN RCNS Lendület "Momentum" DNA Repair Research Group, Institute of Molecular Life Sciences, HUN-REN Research Centre for Natural Sciences, Budapest, Hungary   [5]NYU Grossman School of Medicine, Department of Biochemistry and Molecular Pharmacology, New York, NY, USA

Correspondence: bekesi.angela@vbk.bme.hu; vertessy.beata@ttk.hu

(Periyasamy et al, 2021). The AID/APOBEC family comprises multiple proteins (AID, A1, A2, seven members of A3s, and A4), each differing in substrate preferences, catalytic activity, and biological function. Among these, AID, A1, and A3s have well-documented DNA-editing activities (Harris et al, 2002). APOBEC3 enzymes primarily function within innate immunity, playing roles in antiviral defense, clearance of foreign DNA, and restriction of mobile genetic elements via deaminase-dependent or deaminase-independent mechanisms (Bogerd et al, 2006; Stenglein et al, 2010; Sadeghpour et al, 2021). Dysregulation of APOBEC3s has been linked to cancer development (Venkatesan et al, 2021; Dananberg et al, 2024) and drug resistance (Venkatesan et al, 2018; Periyasamy et al, 2021; Mertz et al, 2022). APOBEC mutational signatures, resembling kataegis, are detected in most sequenced tumor genomes (Burns et al, 2013; Roberts et al, 2013; Taylor et al, 2013), frequently resulting from episodic APOBEC expression (Petljak et al, 2019). In line with their role in antiviral defense and protection of genome integrity, APOBEC3 enzymes can be induced by foreign RNA and DNA molecular patterns (Stenglein et al, 2010) or by genotoxic stress caused by various chemotherapeutic agents (Oh et al, 2021; Isozaki et al, 2023). Short-term treatments of cancer cell lines with etoposide, cisplatin, or 5FU were shown to induce p53-independent expression of A3B, as well as the expression of other APOBEC3s (A3C, A3H, and to a lesser extent A3D and A3F) via p53-dependent mechanisms (Periyasamy et al, 2021). Interestingly, A3B expression is repressed by p53 and by the MDM2 inhibitory drug, Nutlin3A (Menendez et al, 2017; Periyasamy et al, 2017). Notably, the effects of the thymidylate synthase inhibitors RTX and 5FdUR on APOBEC3 expression have not yet been investigated.

Previously, we described genomic uracil patterns in DNA repair-deficient HCT116 cells treated with 5FdUR and RTX (Pálinkás et al, 2020). We found that drug-induced genomic uracil profiles strongly correlated with genomic regions of early replication timing, and, to a lesser extent, with more actively transcribed euchromatin (cf. Fig 4 in Pálinkás et al [2020]). In addition, both MMR-deficient and MMR-proficient versions of this model exhibited strong S-phase arrest and an induced DNA damage response (cf. Figs 5 and 5-figure supplement 1 in Pálinkás et al [2020], respectively). On the one hand, the S-phase arrest phenomenon helps explain the observed strong correlation between uracil-enriched and early replication timing regions, as replicative DNA synthesis occurs preferentially in these regions, allowing efficient uracil incorporation into the genome when cellular dTTP is limited. On the other hand, sites of active transcription may also be coupled with repair synthesis (e.g., transcription-coupled repair), which could further contribute to uracil incorporation (Owiti et al, 2018). In summary, we suggested that uracil incorporation occurs at sites of replicative or repair synthesis. Interestingly, the two drug treatments resulted in distinct genomic uracil distributions that were differentially affected by cellular MMR status, as reflected in the observed correlations with replication timing and transcriptional regulatory features (Pálinkás et al, 2020). Based on these findings, we asked whether the differences in uracil distributions caused by the two TS inhibitors, RTX and 5FdUR, might reflect altered molecular mechanisms of cellular drug response. Our specific questions were as

follows: (i) Where are the sites of altered uracilation? (ii) Do these differences affect specific genes? (iii) What might explain the altered impact of MMR status? Are there induced mutational processes that produce extra MMR substrates? (iv) If so, what factors are induced that cause mismatches? (v) Are these differences manifested in altered cytotoxicity or cell cycle arrest?

In the present study, we provide a deeper insight into the drug-specific genomic uracilation patterns in the context of the cellular MMR status and induced mutagenicity. Using genome segmentation, we identified regions that were differentially uracilated in response to the two drug treatments in both MMR-deficient and MMR-proficient cells. We introduced a quantitative measure for uracil enrichment of genes (called U-score) to investigate functional relationships within the groups of genes that are either the most or most differentially uracilated in response to the two drug treatments. Furthermore, variant analysis of genome sequencing data revealed an increased frequency of C-to-T transitions selectively occurring in response to high-dose 5FdUR treatment of MMR-deficient and UNG-inhibited HCT116 cells. The mutational spectra indicated the involvement of APOBEC3 DNA cytidine deaminases, which was experimentally confirmed. The coincidence between increased genomic variability and decreased cellular response to 5FdUR further emphasizes the impact of the observed drug-specific differences. Our results allow new insights into the mechanism of action of RTX and 5FdUR, with potential relevance for personalized anticancer therapy.

# Results

## Certain genomic segments exhibit altered uracil enrichment depending on drug treatment and cellular MMR status

As we previously reported, the two thymidylate synthase inhibitors (RTX and 5FdUR) resulted in similar, but not identical, genomic uracil patterns in the UNG-inhibited HCT116 cell line and in its MMR-proficient variant (Figs 3 and 3-figure supplement 3 in Pálinkás et al [2020]). Specifically, RTX treatment (0.1 $\mu$M) resulted in a strong correlation between genomic uracilation and early-replicating regions, which was weaker in response to 5FdUR treatment (20 $\mu$M) and further reduced in the MMR-proficient cells.

To evaluate the significance of these drug-specific differences, we now applied an unsupervised genome segmentation analysis using Segway (Hoffman et al, 2012; Chan et al, 2017). In this approach, the genome is divided into segments (clusters of genomic regions) that share similar patterns of uracil enrichment across the seven published U-DNA-seq samples (Pálinkás et al, 2020). We allowed Segway to define 12 distinct segment types, each representing a characteristic uracil enrichment pattern across the samples. For simplicity, we refer to these segment types as "segments" throughout the article. Segmentation analysis of merged replicates confirmed previously observed trends in replication timing-dependent uracilation, which, in 5FdUR-treated cells, was strongly influenced by the cellular MMR status (Fig 1).

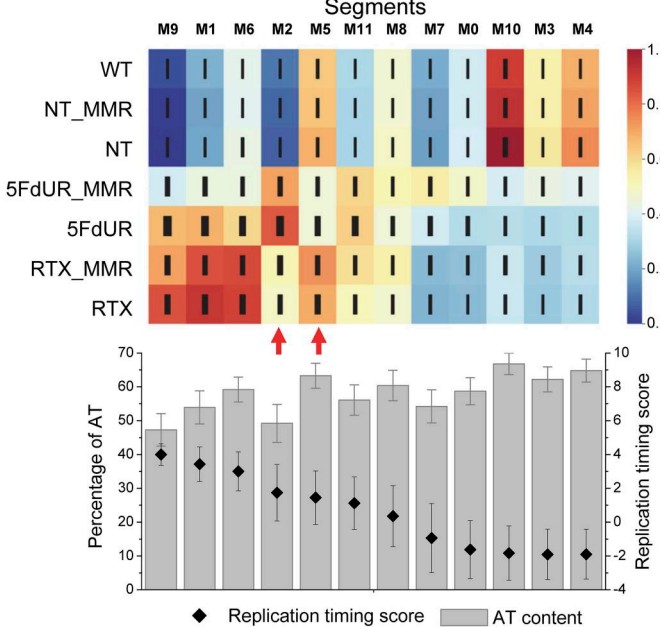

**Figure 1. Drug-specific differences in genomic uracil-DNA patterns.**
Unsupervised genome segmentation was performed on uracil-DNA enrichment profiles (merged replicates) previously published in Pálinkás et al (2020) using Segway 2.0 (Hoffman et al, 2012; Chan et al, 2017) with 12 labels as described in the Materials and Methods section. WT, wild-type HCT116 cells, all the other samples are from UNG-inhibited HCT116 derivatives; NT: nontreated; RTX: 0.1 μM raltitrexed treated; 5FdUR: 20 μM 5-fluoro-2′-deoxyuridine treated. When MMR is indicated, it refers to the mismatch repair-proficient derivatives of the HCT116 cell line (cf. also in Table 3). The heatmap (generated using Seaborn [Matplotlib module of Python]) represents the uracil enrichment signal distribution among 12 genomic segments (M0–M11) within the seven samples (relative scale is provided at the right; the thickness of the black lines indicates SD). The segments were arranged according to their decreasing average replication timing (RT) score (black symbol on the bottom graph). RT scores and the percentage of AT content (gray bars on the bottom graph) were calculated as described in the Materials and Methods section, and represented on the bottom graph precisely aligned to the uracil enrichment signal distribution map. Error bars indicate the SD calculated for genomic intervals of the corresponding segments. Data were plotted by Origin 8.6 (OriginLab Corporation). Red arrows indicate the segments exhibiting major drug-specific differences in their uracil content.
Source data are available for this figure.

In addition, the analysis revealed two genomic segments (M2 and M5, indicated by red arrows in Fig 1) with markedly altered uracil content between the two drug treatments.

Genome segmentation analyses were performed on individual replicates to estimate the robustness and reliability of the approach (Fig S1A). In nontreated (NT and WT) and in RTX-treated samples, the individual replicates display high similarity in their uracil enrichment patterns. In contrast, the 5FdUR-induced genomic uracilation was less stable; however, the main tendencies remained consistent, and the two segments with the most pronounced drug-specific differences were identifiable in these data as well (R5 and R3 in Fig S1A).

The two sets of genomic segments derived from the merged data (M0-M11) and the individual replicates (R0-R11) were approximately matched based on uracil enrichment distributions, AT content, and average replication timing scores. This was

further refined by measuring overlaps between the two sets of segments using BEDTools Jaccard indices (Quinlan & Hall, 2010) (Fig S1B). The M and R segments with the most characteristic uracil enrichment distributions across samples also exhibited the best correspondence (Jaccard indices above 0.5: M9(R4), M1(R9), M2(R5), M11(R1), M8(R2), M7(R8), M0(R7), M10(R0)). Despite lower overlap, the correspondence of M6-R6 and M5-R3 pairs could still be unequivocally established. Segments M3 and M4, with highly similar uracil enrichment distributions, together correspond to R10, whereas R11 was identified as an additional segment in the replicate data.

Both sets of genomic segments presented balanced length distributions, as calculated using Segtools (Buske et al, 2011) (Fig S1C). Their correlations with replication timing and AT content (Figs 1 and S1A) were similar to those of previously published segments defined using a larger set of distinct genomic profiles (cf. Fig 4-figure supplement 3 in Pálinkás et al [2020]). The most uracilated regions in nontreated cells corresponded to late-replicating regions with the highest AT content (segments M10(R0), M3-M4(R10) in Figs 1 and S1). In contrast, RTX-induced uracilation occurred primarily in the early-replicating segments, with only minor changes once MMR is restored (cf. M9(R4), M1(R9), M6(R6)). In comparison, 5FdUR-induced uracilation displayed a weaker correlation with early-replicating regions, which decreased further in the MMR-proficient cells. Overall, the cellular MMR status influences the uracilation profiles more strongly after 5FdUR treatment, whereas RTX-induced uracilation was more resistant to MMR (Fig 1). Segment M2(R5), selectively uracilated in response to 5FdUR treatment, was associated with high GC content and early-to-mid replication timing (Figs 1 and S1A). Segment M5(R3), among the most uracilated segments in RTX-treated cells, also reflected early-to-mid replication timing but was characterized by low GC content and lacked uracil enrichment after 5FdUR treatment.

The most uracilated segments in drug-treated cells (M9(R4), M1(R9), M6(R6)), as well as M2(R5), were enriched in genes, whereas those in the nontreated cells (M10(R0), M3(~R10), and M4(~R10)) were depleted in gene elements (Fig S2). This is consistent with our previous interpretation that TS inhibitors induce uracil enrichment primarily in early-replicating, actively transcribed euchromatin regions, whereas in nontreated cells, the minimal amount of uracil accumulates mainly in the AT-rich, compact heterochromatin that typically lacks genes. The two segments (M2(R5) and M5(R3)) identified here as loci of drug-specific uracilation are slightly enriched in gene parts, but with different patterns. Segments M5(R3) displayed selective enrichment in gene bodies, including internal exons and internal introns (Fig S2).

In summary, RTX and 5FdUR treatments result in distinct, drug-dependent U-DNA profiles, primarily manifested in two genomic segments (M2(R5) and M5(R3)), both of which are enriched in gene-coding regions.

### Drug-specific differences in genomic uracil profiles are reflected at the level of genes

Based on the above results, we assume that differential uracilation might also be reflected at the level of genes. To assess this, we

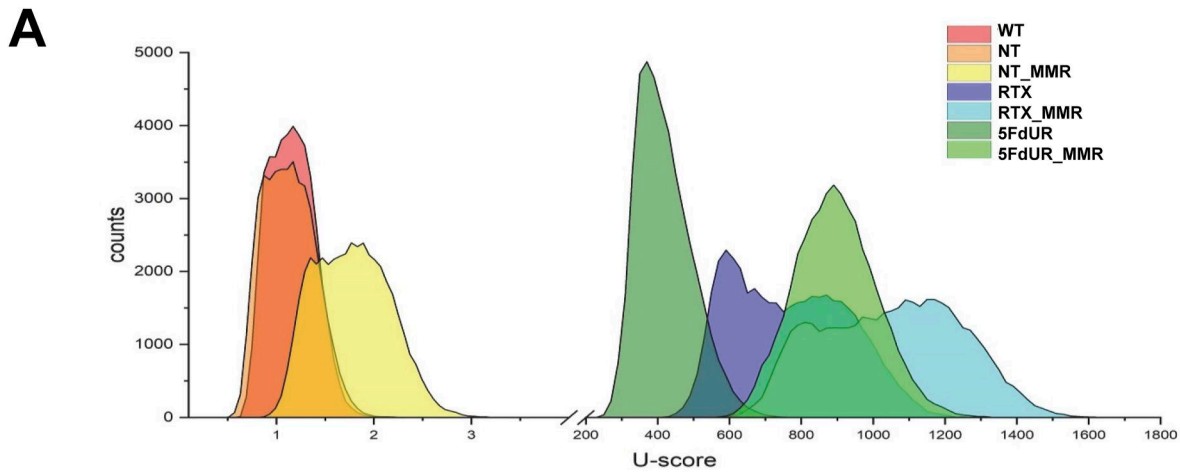

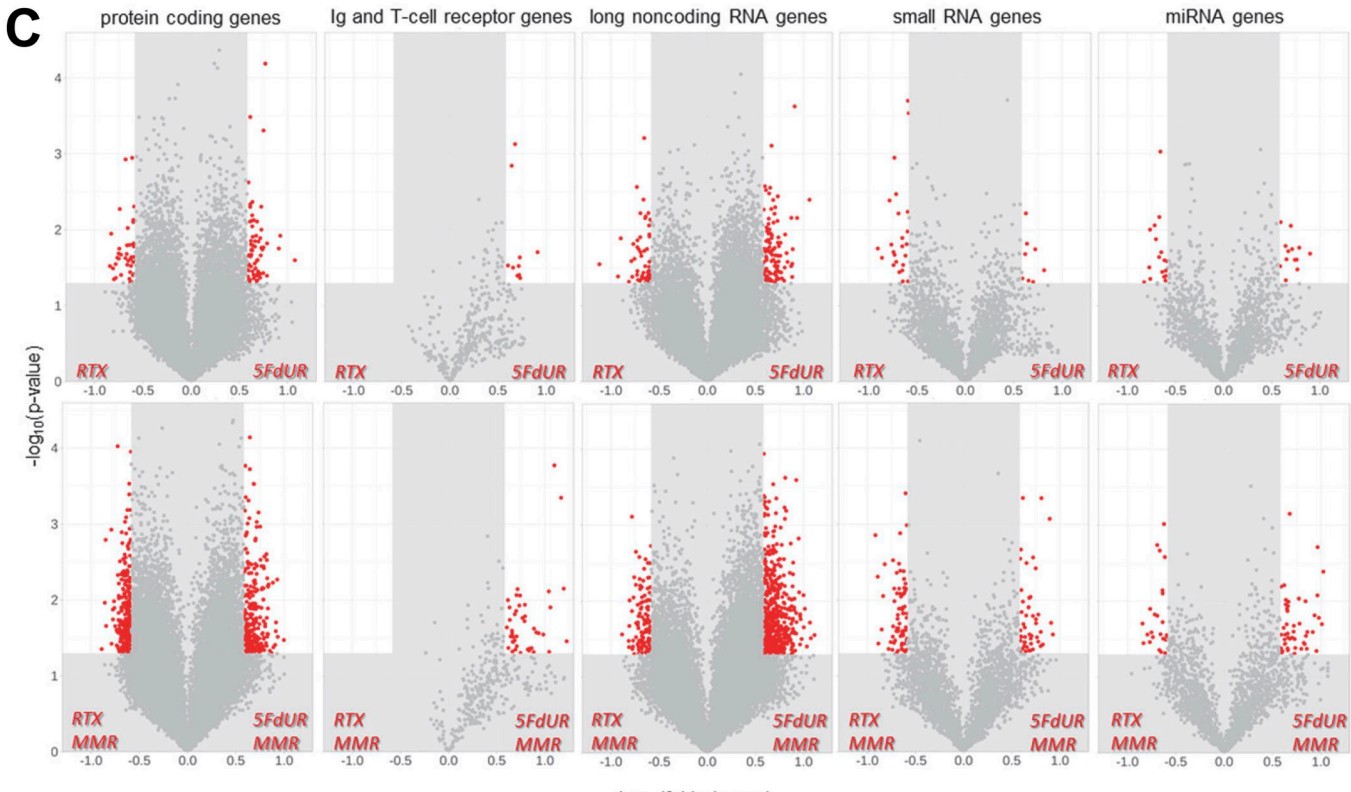

introduced a quantitative measure (termed U-score) to characterize the extent of gene-specific uracilation. For its calculation, the uracil enrichment tracks were rescaled according to the global uracil content measured previously by dot blot (e.g., in RTX samples, ~700 uracil per million base pairs, meaning 1.9 million per genome, as listed in Table S1 [Pálinkás et al, 2020]). In this way, the absolute uracil amount was proportionally distributed across the genome according to the enrichment signal tracks (see the Materials and Methods section and Supplemental Data 1). Each gene (annotated in GENCODE v34 [Harrow et al, 2006, 2012]) was considered in its longest isoform together with its 1,000-bp upstream region as the core promoter. As expected from the two orders of magnitude difference in global uracil contents, dramatic differences in uracilation were observed between the nontreated and drug-treated samples (Fig 2A). In addition, marked differences in U-scores were visible between RTX and 5FdUR treatments, which were further modulated by the cellular MMR status. These differences were apparent from the distribution of U-scores (Fig 2A): in RTX-treated cells, genes were more clearly separated into two populations with high or modest U-scores, whereas 5FdUR treatment did not cause such separation. Notably, MMR proficiency caused a broadening of the U-score distributions after both drug treatments. Given the higher global uracilation in MMR-proficient cells (Pálinkás et al, 2020), an overall upward shift of the distributions was expected. Interestingly, MMR proficiency influenced U-score distribution even in nontreated cells, whereas UGI expression did not result in detectable changes.

Drug-specific differences are further reflected in the distinct over- or underrepresentation of certain gene classes (as defined in the GENCODE v34 list) among the top 2,000 genes with the highest U-scores (Fig 2B). In nontreated cells, which exhibit low overall genomic uracil content, the most uracilated genes are relatively enriched in pseudogenes and small RNA (sRNA) genes, whereas protein-coding genes are underrepresented, unlike in the drug-treated cells. Interestingly, immunoglobulin (IG) and T-cell receptor (TR) genes are almost absent in the top 2,000 of RTX-treated samples, but enriched in response to 5FdUR treatment—particularly in MMR-proficient cells. Conversely, sRNA genes are underrepresented in the most uracilated genes of 5FdUR samples, but are slightly overrepresented after RTX treatment, a pattern again more pronounced in MMR-proficient cells.

To identify those genes that are significantly differentially uracilated in response to the two drug treatments, we calculated U-scores across independent replicates. Log$_2$(fold change) values were determined for 44,000 genes, and raw P-values were obtained using Welch's two-sample $t$ tests. To evaluate the significance of drug-specific differences, we also applied FDR correction for multiple testing; however, because of small effect sizes and the large number of tests, none of the comparisons reached statistical significance. Therefore, we present volcano plots using raw P-values for each major gene class (Fig 2C), marking genes with |log$_2$(fold change)| > 0.585 and P < 0.05, and interpret these results with appropriate caution.

Protein-coding and microRNA (miRNA) genes appeared balanced between the two treatments, whereas long noncoding RNA (lncRNA) and small RNA genes exhibited mild biases toward 5FdUR and RTX treatments, respectively. In contrast, IG and TR genes displayed a strong bias toward 5FdUR treatment, in good agreement with their enrichment among the top 2,000 most uracilated genes (cf. Fig 2B). Notably, MMR proficiency further increased the number of genes displaying differential uracilation between the two drug treatments (Fig 2C).

### Drug-specific biases in U-scores of the genes correlate with certain functional properties

To get a better insight into the biological relevance and the possible functional connections of the differentially uracilated genes, we performed a complex functional enrichment analysis (gene set enrichment analysis, GSEA) on the protein-coding gene lists in the STRING database (version 12.0, https://string-db.org [Szklarczyk et al, 2019]). Both the most uracilated genes and those that feature the highest drug-specific differences in their relative U-scores (i.e., most DU genes) were addressed. Functional enrichments were performed either within the group of top 200 U-score genes against a background (analysis: "Multiple proteins") or within hierarchical lists of genes with values (analysis: "Proteins with Values/Ranks"). The raw results of these three approaches are available at permanent links (Table 1). The scheme in Fig 3 represents the main tendencies we found to be conclusive from the complex analysis detailed below (Fig S3 and Source Data File).

**Figure 2. U-scores for genes reflect treatment-dependent alterations also influenced by the cellular MMR status.**
U-scores were calculated for each gene (cf. GENCODE v34 annotation [Harrow et al, 2006, 2012]) in each sample as described in the Materials and Methods section. Sample names are as above (cf. Table 3): WT HCT116 (WT), UNG-inhibited mismatch repair-deficient (not labeled) or mismatch repair-proficient (MMR) HCT116, either nontreated (NT or NT_MMR) or treated with 0.1 μM RTX (RTX or RTX_MMR) or 20 μM 5FdUR (5FdUR or 5FdUR_MMR). **(A)** U-score distribution among genes. Histograms (counts = number of genes per U-score bin) were calculated for a range of U-scores from 0 to 13,000 with a bin size of 20 for the drug-treated samples, and from 0 to 39 with a bin size of 0.06 for the nontreated samples. The colors applied to the samples (as indicated in the figure) are consistent throughout the entire article, following the color scheme in Pálinkás et al (2020). **(B)** Representation of different gene classes within the top 2,000 most uracilated genes. The gene classes (left) are derived from the GENCODE v34 annotation (TR, T-cell receptor; IG, immunoglobulin; lncRNA, long noncoding RNA; miRNA, microRNA; snRNA, small nuclear RNA; snoRNA, small nucleolar RNA; sRNA, small RNA; other: all other categories as defined in GENCODE v34). Fold enrichments relative to the expected numbers are given for each sample; cells are colored accordingly: depletion (blue), enrichment (red). **(C)** Differentially uracilated genes in RTX- versus 5FdUR-treated cells. Relative U-scores were calculated for individual replicates and compared between RTX- and 5FdUR-treated samples in either MMR-deficient (upper row of volcano plots) or MMR-proficient (bottom row) cases. Significance was tested using Welch's two-sample $t$ test. The negative logarithm of raw P-values (on the y-axis) was plotted against log$_2$(fold change) values for each gene. Genes with P-value < 0.05 and fold change > 1.5 (red dots) were considered to be significantly differentially uracilated in response to the two drug treatments. For details, see Supplemental Data 1. For panel (B), the full lists of U-score data are accessible through GEO Series accession number GSE285931. Source data are available for this figure.

**Table 1.  Permanent links for functional enrichment analysis results in STRING.**

| | | |
|---|---|---|
| "Proteins with Values/Ranks" | 1. Permanent links for hierarchical U-score lists: | Source Data File |
| | WT | |
| | NT | |
| | NT_MMR | |
| | RTX | |
| | RTX_MMR | |
| | 5FdUR | |
| | 5FdUR_MMR | |
| "Multiple proteins" | 2. Permanent links for the top 200 genes with the highest U-score: | |
| | RTX top200 | |
| | 5FdUR top200 | |
| | RTX_MMR top200 | |
| | 5FdUR_MMR top200 | |
| "Proteins with Values/Ranks" | 3. Permanent links to the differential uracilation list (hierarchical list by combined score of $-\log_{10}[P\text{-value}]$ and $\log_2[\text{fold change}]$): | Source Data File |
| | RTX vs 5FdUR (RTX-specific uracilation – top of the list – lower scores) | |
| | RTX_MMR vs 5FdUR_MMR (RTX_MMR-specific uracilation – top of the list – lower scores) | |
| | 5FdUR vs 5FdUR_MMR (5FdUR-specific uracilation –top of the list – lower scores) | |
| | RTX vs RTX_MMR (RTX-specific uracilation – top of the list – lower scores) | |

Two search options were applied: "Proteins with Values/Ranks" and "Multiple proteins" as indicated on the left. "Proteins with Values/Ranks" for the full hierarchical list of protein-coding genes was ranked by either the U-scores (most uracilated genes (1)) or a combined score derived from the fold enrichment and the *P*-values (most DU genes (3)) (cf. the Materials and Methods section). "Multiple proteins" was applied for the top 200 most uracilated genes (2).

In the hierarchical U-score lists (cf. section 1 in Table 1), the most uracilated genes in nontreated cells, regardless of their DNA repair status, were enriched in chemoreceptors (taste and odor sensing), cadherins, keratin-associated proteins, ion channels, and interferons, whereas the least uracilated ones were enriched mainly in homeobox domain-containing transcription factor genes. In contrast, in drug-treated cells, depending on the applied drug and the MMR status, the genes with higher U-scores were different and functionally diverged. Chemoreceptors and cadherins consequently appeared in the low U-score range, except the MMR-proficient 5FdUR-treated cells, which constitute the most outlier sample. In addition, after RTX treatment, the keratin-associated proteins also become enriched among the least uracilated genes. In the MMR-deficient 5FdUR sample, genes encoding homeobox domains, keratin filament components, and some immunoglobulins exhibited strong enrichment toward the higher U-scores. In comparison, in MMR-proficient 5FdUR-treated cells, the enrichment of homeobox genes was weakened, and the immunoglobulins and immunity-related functionalities were strengthened within the high U-score range. The tendencies that some ion channels also appear within the high U-score range, and the chemoreceptors and the cadherins disappear from the low U-score range in this MMR-proficient sample, bring it closer to the nontreated cases. In response to RTX treatment, the functional coherence among the most uracilated genes is lower, and homeobox proteins are rather depleted than enriched in the high U-score range. In the MMR-deficient cells, the cluster of metallothionein genes and components of intracellular intermediate filaments, whereas in the MMR-proficient cells, only splicing-related genes, display strong enrichment within the high U-score range (Fig 3, Source Data File).

Similar functional enrichment analysis on **differential uracilation** (on hierarchical lists, cf. section 3 in Table 1, and the Materials and Methods section) confirmed 5FdUR-biased uracilation in homeobox transcription factor genes regulating developmental processes, as well as for immunoglobulins, ion channels, and cadherins, all of which are involved in extracellular processes and cell–cell communication. In contrast, RTX-biased differential uracilation affects genes related to intracellular processes, such as RNA processing, cell cycle regulation, DNA repair, and viral host interactions. In the MMR-proficient cells, for the RTX-biased uracilation, most of these drug-specific differences remain similar, whereas the splicing and the stress response genes appear even more enriched. As expected, MMR status has a higher impact on the 5FdUR-biased uracilation: enrichment for homeobox transcription factors and immunoglobulins is decreased, whereas keratins and chemoreceptors are rather enriched.

The aforementioned tendencies were further confirmed in the **functional network analysis of the 200 most uracilated genes** (cf. section 2 in Table 1). These networks are more interconnected and

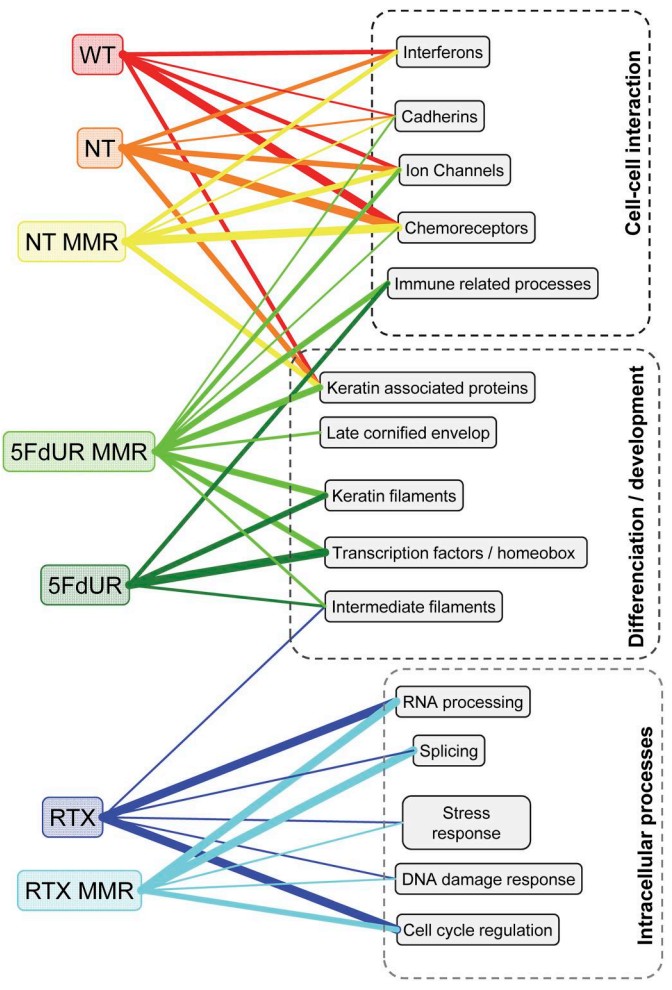

**Figure 3. Differential uracilation induced by RTX and 5FdUR is reflected in altered biological functions enriched among gene sets with the highest U-scores.**
Samples are shown on the left, color-coded as in Fig 2A. Enriched biological functions associated with the most uracilated genes are listed on the right. Connecting lines are colored by sample type, with increasing thickness indicating stronger enrichment. RTX primarily affects genes involved in intracellular processes, whereas 5FdUR impacts transcription factor genes that regulate differentiation and developmental processes. The MMR status has little effect on RTX-treated samples, but 5FdUR-treated, MMR-proficient cells display gene uracilation enrichment patterns resembling those of nontreated samples.
Source data are available for this figure.

functionally coherent in response to the 5FdUR treatment as compared to the RTX-treated cases, regardless of the cellular MMR status (Table 2).

To further visualize the results of the complex functional enrichment analysis above, networks of the top 200 most uracilated protein-coding genes were extracted from the STRING database, and visualized in Cytoscape (Shannon et al, 2003; Excoffier et al, 2017), colored according to the major enriched functionalities and subcellular localization (Fig S3). The higher interconnectivity in "5FdUR" networks, and the overrepresentation of membrane channel (pink) and membrane receptor genes (purple-violet) are obvious, as well as the presence of the cytokine/hormone/interferon genes and other extracellular components (blue) is also more pronounced here (Fig S3B and D) as compared to the "RTX" networks (Fig S3A and C). The overrepresentation of transcription factor genes (green) in the MMR-deficient 5FdUR-treated sample is also obvious, which is decreased in the corresponding MMR-proficient sample. In contrast, RTX causes uracilation in functionally more diverse genes; still, the RNA processing genes (orange), especially the splicing-related genes (dark yellow), seem to be more affected (Fig S3A and C). In summary, the above analysis demonstrates that the two TS inhibitory drugs induce uracilation in groups of functionally related genes in a drug-specific manner.

## The frequency of C:G-to-T:A transitions is increased selectively in 5FdUR-treated, UNG-inhibited, and MMR-deficient HCT116 cells

The drug-specific differences in genomic- and gene-level uracilation presented above suggest markedly altered molecular mechanisms of action of the two TS inhibitory drugs, 5FdUR and RTX. One possible source of these variations could be that 5FdUTP, the metabolite of 5FdUR, can lead to 5FU incorporation into the genomic DNA opposite to either A or G (Meyers et al, 2005). The appearance of 5FU:G mispairs upon DNA synthesis would result in an increased number of C:G-to-T:A and also T:A-to-C:G transitions in 5FdUR-treated, UNG-inhibited, and MMR-deficient cells. Hence, we performed somatic variation calling in the genome sequencing datasets (input samples of the U-DNA-seq published in Pálinkás et al [2020], seven combinations per treatment [cf. the Materials and Methods section and Source Data File]) using the Mutect2 module of the GATK package (McKenna et al, 2010; Van der Auwera et al, 2013). On the lists of filtered variants, we measured relative occurrences of the six types of single-base substitutions

**Table 2. Network statistics from STRING "Multiple protein" search (cf. sections 2 in Table 1).**

| | Top 200 genes with the highest U-score (2) | | | |
|---|---|---|---|---|
| | **RTX top200** | **5FdUR top200** | **RTX MMR top200** | **5FdUR MMR top200** |
| Number of nodes | 200 | 200 | 200 | 200 |
| Number of edges | 118 | 185 | 89 | 208 |
| Average node degree | 1.18 | 1.85 | 0.89 | 2.08 |
| Avg. local clustering coeff. | 0.366 | 0.343 | 0.353 | 0.371 |
| Expected number of edges | 82 | 78 | 75 | 80 |
| PPI enrichment $P$-value | $1.33 \times 10^{-4}$ | $<1.0 \times 10^{-16}$ | 0.0576 | $<1.0 \times 10^{-16}$ |

The raw results of the functional enrichment analysis are provided in Source Data File.

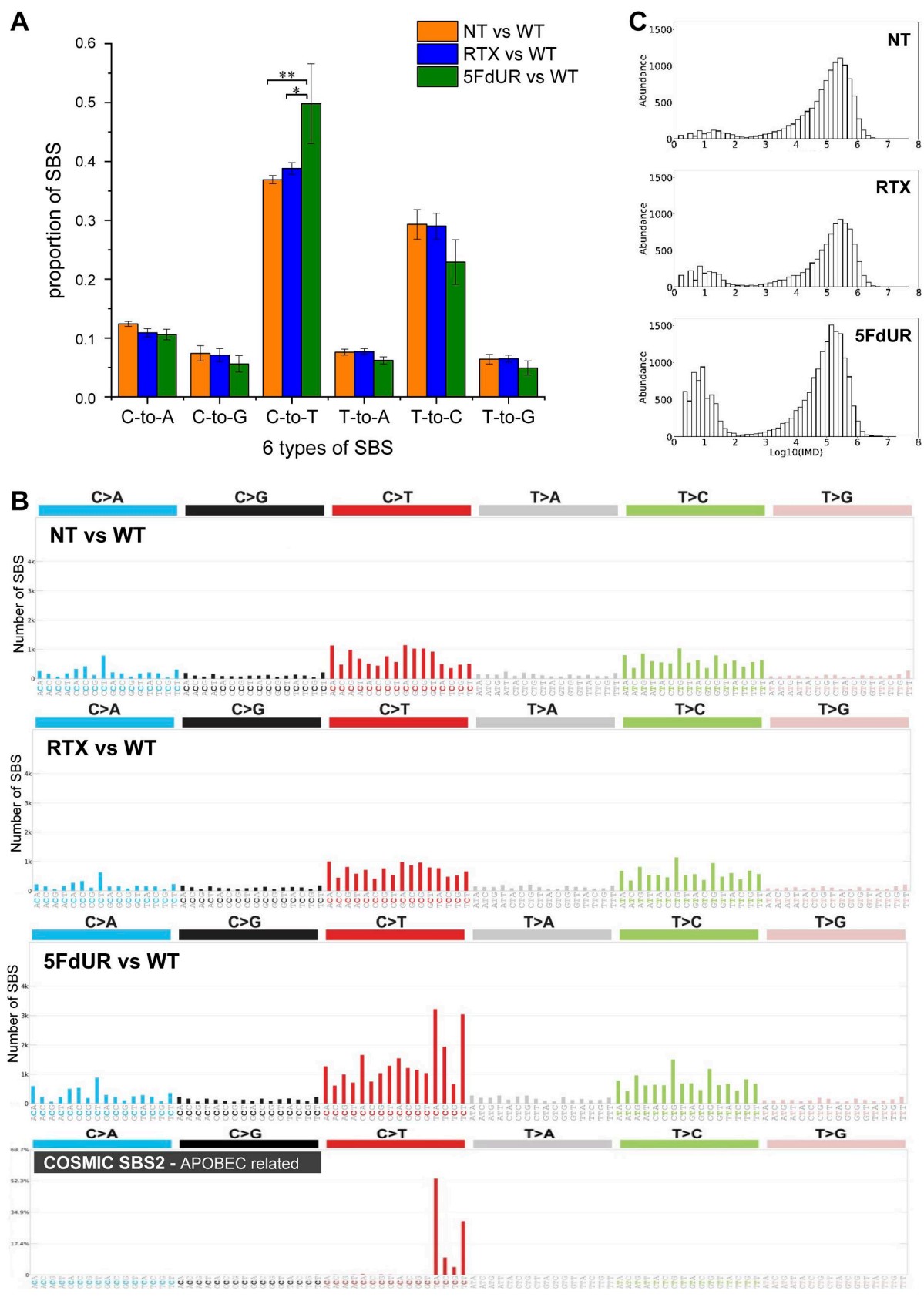

(SBSs), also weighted with their detected allele frequencies (cf. the Materials and Methods section). The ratios of SBSs were reproducible, although the exact numbers were varied among individual variant calling experiments. We have found a significantly elevated ratio of C:G-to-T:A transitions in 5FdUR-treated cells compared with RTX-treated or nontreated ones (Fig 4A). Significance was checked by Welch's two-sample $t$ tests ($n$ = 7, with $P$-values of 0.00244 and 0.00519 for the comparisons of the 5FdUR sample with NT and RTX samples, respectively). Notably, the opposite transition from T:A to C:G did not display an increase in 5FdUR-treated cells compared with the untreated samples. Hence, 5FU incorporation and mispairing with G might not be the major source of mutagenesis in these MMR-deficient and UNG-inhibited cells in response to 5FdUR treatment.

### Mutational spectra suggest 5FdUR-induced APOBEC activation in DNA repair-deficient cells

The detected C:G-to-T:A transitions can also be explained by unrepaired cytosine deamination events, especially in this cell line, where both UNG-initiated repair of uracil-DNA and MMR-dependent repair of U:G mispairs are abolished. Cytosine deamination might occur spontaneously or be catalyzed by the AID/APOBEC DNA cytidine deaminases. Enzymatic deamination can lead to characteristic mutational signatures (cf. SBS2 in COSMIC database, https://cancer.sanger.ac.uk/signatures/sbs/sbs2/ [Alexandrov et al, 2020; Campbell et al, 2020]) and also to short intermutational distances (IMDs, cf. kataegic mutational profile [Nikkilä et al, 2017]) detectable in the genomic data.

To check such traces of enzymatic deamination, triplet SBS signatures were determined (using the SigProfiler tool [Bergstrom et al, 2019]). Notably, the signature of the 5FdUR-treated sample, aside from the background, displays a discernible similarity to the AID/APOBEC-related SBS2 signature (Fig 4B). Furthermore, the distribution profiles of the IMDs calculated for the C-to-T (from this point, we use this simple term for C:G-to-T:A) mutations revealed that 5FdUR treatment caused a marked increase in the frequency of short distances (two to several tens of nucleotides) as compared to RTX-treated or NT samples (Fig 4C). Similar extreme clustering of APOBEC-induced transitions was described recently for APOBEC3C (Brown, 2024). Moreover, a specific double-base substitution (DBS), the CC-to-TT, was also found to be increased in response to 5FdUR treatment (Fig S4). This DBS might be explained by two adjacent cytosine deamination events (i.e., extremely short IMD) that further strengthen the hypothesis of enzymatic deamination over a spontaneous one.

To further confirm these results, similar variation calling experiments were performed on the genome sequencing data of the MMR-proficient cells (input samples of the U-DNA-seq experiments published previously [Pálinkás et al, 2020]). To make a proper comparison, variant calling was performed on data from drug-treated MMR-deficient and MMR-proficient cells relative to their corresponding nontreated samples (see the Materials and Methods section and Source Data File). This approach confirmed the previously observed increase in C-to-T transition frequency and the characteristic signature associated with 5FdUR treatment in MMR-deficient cells. In contrast, neither the RTX-treated cells nor any of the MMR-proficient samples exhibit a similar increase (Fig S5).

Furthermore, the increased frequency of CC-to-TT DBS in response to 5FdUR treatment was also observed exclusively in MMR-deficient cells (Fig S6). Based on these results, either 5FdUR treatment does not cause APOBEC activation in MMR-proficient cells or the restored MMR can efficiently repair the U:G mispairs eventually generated by APOBEC enzymes.

### 5FdUR treatment-induced C-to-T transitions occur preferentially in early- to mid-replicating genomic regions, but not necessarily in the most uracilated genomic segments

We also investigated whether the genomic distribution of these C-to-T transitions correlates with certain genomic features or uracilated segments. Previously, a strong correlation was detected between genomic uracil enrichment and early replication timing, which could be well explained by the profile of replicative DNA synthesis in these S phase-arrested cells (Pálinkás et al, 2020). We now demonstrate that the 5FdUR-induced elevation of C-to-T frequency also correlates with early-to-mid replication timing regions (Fig 5A). This observation provides additional support for the notion that these additional transitions indeed occur during the 48-h cell cycle-arresting treatment. We also addressed the possible correlation with genomic uracil enrichment; however, its major source could be the thymine replacement in this model. C-to-T transition frequencies were calculated for the segments derived by the Segway analysis (cf. Fig 1), and we found that segment M2 (the most uracilated after 5FdUR treatment) did not exhibit a considerable increase (Fig 5B). Instead, the most affected segment is the M5, which is characterized by a much lower uracilation signal. The same tendency is also confirmed by the corresponding segments R5 and R3 (Fig S7). Such observation can easily be explained by methyl-cytosine deamination not resulting in uracil, but thymine. However, methylated CpG islands are unlikely to be involved, as the proportion of regulatory CpG islands and the promoter flanking regions overlapping with the segment M5 is rather low, similar to other segments characterized with higher C-to-T frequencies (Fig 5B, bottom graph). In

---

**Figure 4. Increased ratio of C:G-to-T:A transitions in response to 5FdUR, but not RTX treatment in UNG-inhibited and MMR-deficient HCT116 cells.**
Variants were called using Mutect2 (McKenna et al, 2010; Van der Auwera et al, 2013) from U-DNA-seq input data derived from nontreated (NT), or 0.1 $\mu$M RTX- or 20 $\mu$M 5FdUR-treated, UNG-inhibited HCT116 cells compared against genome sequencing data from nontreated WT HCT116 (WT) samples (as described in the Materials and Methods section). The corresponding filtered variant data are accessible through the GEO Series accession number GSE285931. **(A)** Elevated C:G-to-T:A transitions in the genomic DNA of the 5FdUR-treated cells. The six possible SBS types (labeled in simplified form, e.g., "C-to-T" for "C:G-to-T:A") were weighted by allele frequencies, and their fractions relative to all SBS were calculated (as described in the Materials and Methods section). Statistical analysis using Welch's two-sample $t$ tests on SBS ratios from seven variant call datasets yielded $P$-values of 0.00244 (**) and 0.00519 (*). Error bars represent SD. **(B)** APOBEC-related origin of 5FdUR-induced C-to-T transitions indicated by mutational spectra. Mutational signatures calculated by SigProfiler (Bergstrom et al, 2019) for merged data from NT, RTX, and 5FdUR samples. The APOBEC cytidine deaminase-related SBS2 mutational signature from the COSMIC v3.1 database (Alexandrov et al, 2020; Campbell et al, 2020) is also presented. **(C)** Distribution of intermutational distances (IMDs). IMDs were calculated for neighboring C-to-T events in the merged data, and their distribution is shown on a logarithmic scale. Source data are available for this figure.

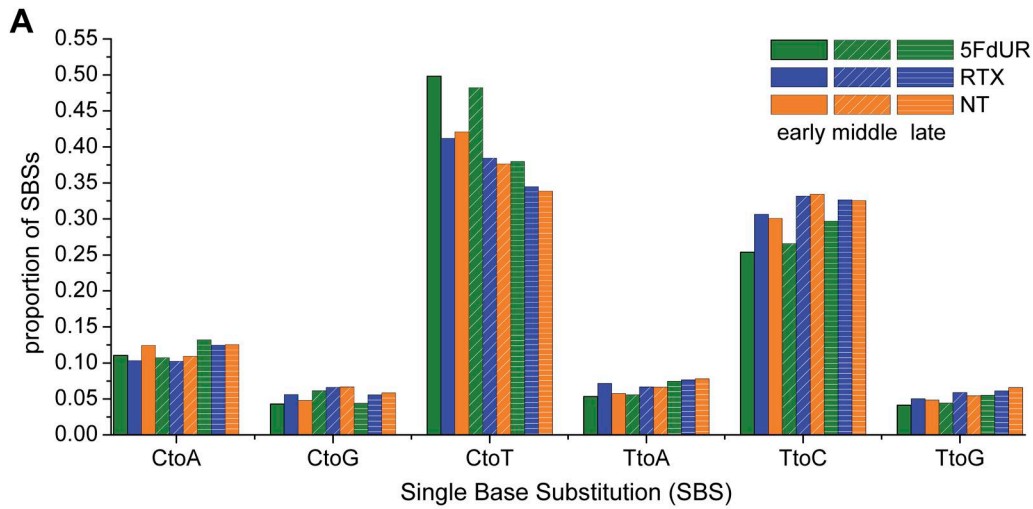

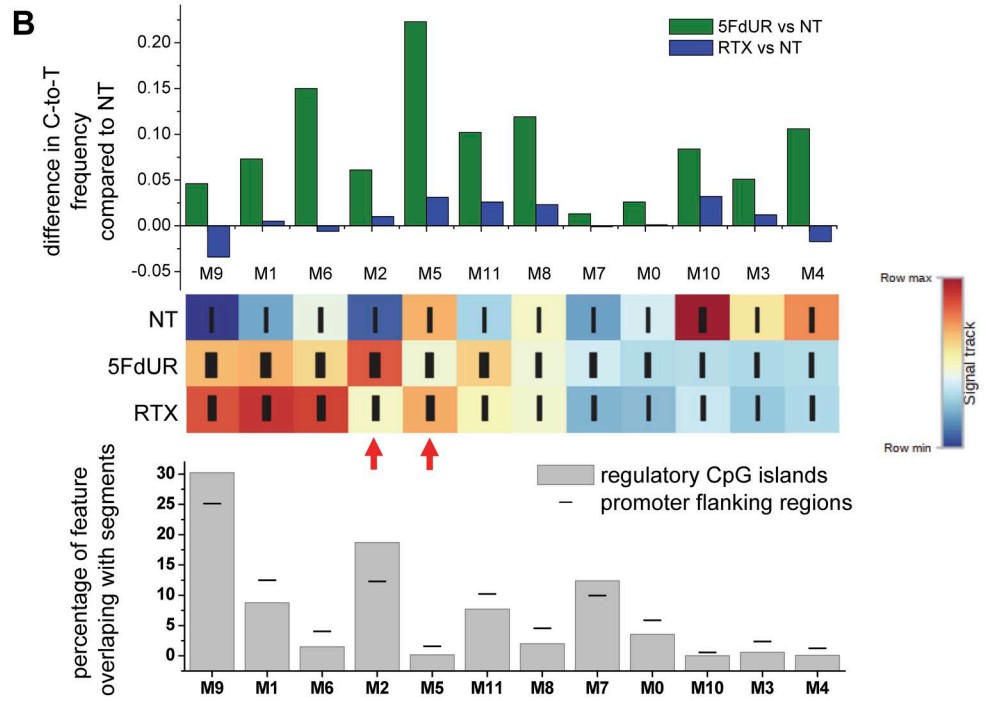

**Figure 5. 5FdUR-induced C-to-T transitions are enriched in early and middle replication timing regions and differentially uracilated genomic segments.**
Filtered variants from merged data were used (cf. Fig 4, GEO Series GSE285931). **(A)** Increased frequency of C-to-T transitions in early and middle replication timing (RT) regions. The occurrence of the six possible SBS types was weighted by allele frequencies, and their relative fractions were calculated for early, middle, and late RT genomic segments (as described in the Materials and Methods section). **(B)** C-to-T transitions in differentially uracilated genomic segments. The ratio of C-to-T transitions was calculated for genomic segments defined by the genome segmentation analysis of U-DNA-seq data (cf. Fig 1; relevant signal distribution patterns are shown at the bottom for comparison). Overlap ratios of regulatory CpG islands and promoter flanking regions were also calculated and plotted (bottom) for each segment. Data were plotted by Origin 8.6 (OriginLab Corporation).
Source data are available for this figure.

contrast, CpG islands mainly overlap with those segments (including M2) that are not affected by the induced C-to-T transitions. Notably, these two segments, the M2 and the M5, have been identified as the most differentially uracilated in response to treatments with the two TS inhibitors, highlighting significant drug-specific differences in their mechanism of action.

**The two TS inhibitors induce distinct cellular responses**

Given the observed drug-specific differences in genomic and gene-level uracilation, as well as in cytosine deamination events, we investigated whether these differences are reflected in or correlate with altered cellular responses to the two TS inhibitors, RTX and 5FdUR.

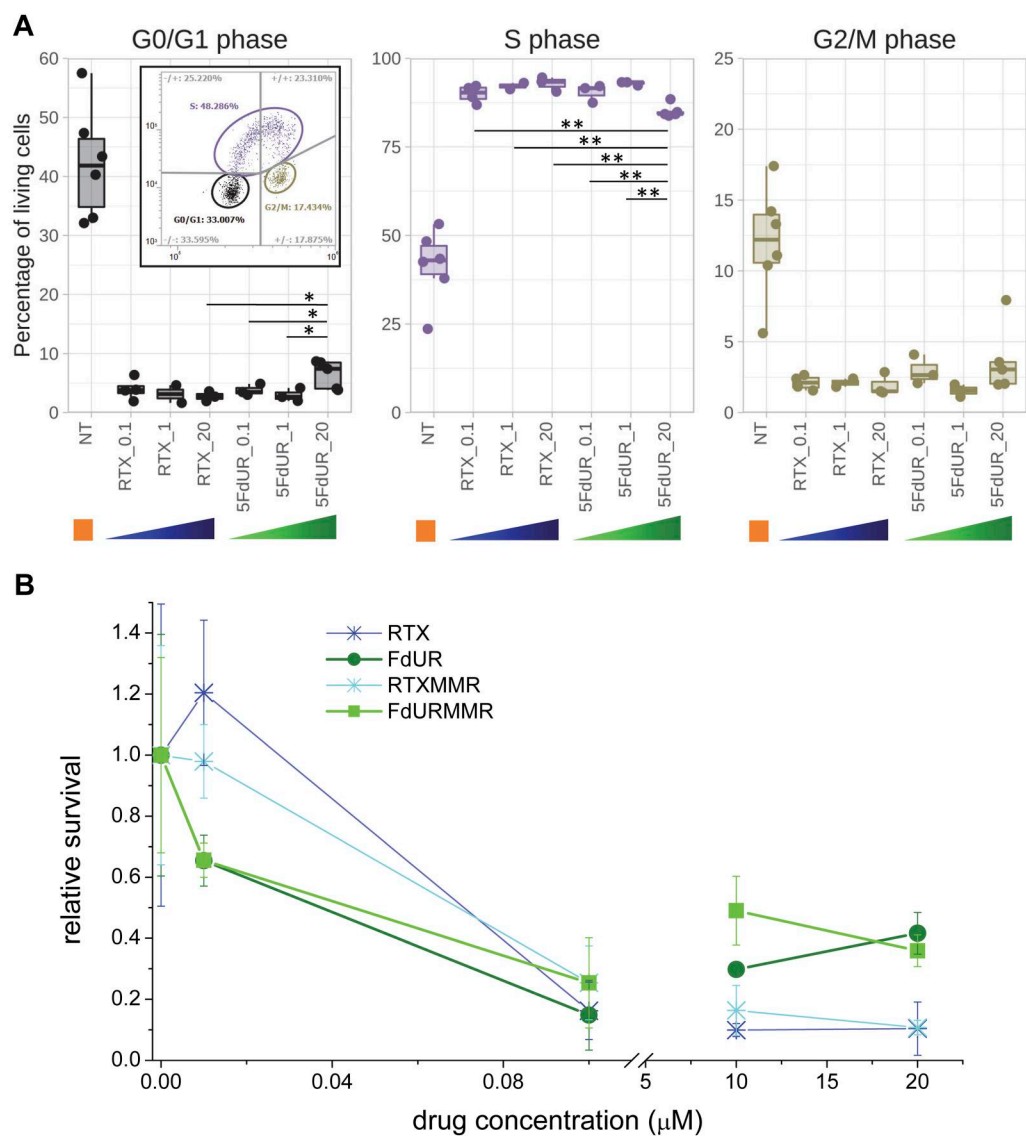

**Figure 6. High-dose 5FdUR treatment causes milder cell responses than RTX or low-dose 5FdUR treatments.**
UNG-inhibited, MMR-deficient HCT116 cells were analyzed. **(A)** High-dose 5FdUR treatment causes milder S-phase arrest. Cells were treated with 0.1, 1, or 20 μM of RTX or 5FdUR for 24 h, pulse-labeled with BrdU before harvesting, and analyzed by FACS after staining with anti-BrdU antibody and propidium iodide (as described in the Materials and Methods section). Flowgrams (BrdU signal versus DNA signal) were recorded for each cell. Inset is the same flow cytometry plot as one at the top of the Fig S8. Biological replicates were evaluated by manual clustering (G0/G1—black; S—purple; G2/M—khaki green). The boxplots were prepared using the ggplot2 package of R. The P-values were calculated using one-tailed t tests for two samples with unequal variance: * and ** are below 0.05 and 0.01, respectively. **(B)** High-dose of 5FdUR treatment causes decreased cytotoxicity. Cultured cells treated with either 0 (NT), 0.01, 0.1, 10, or 20 μM of the drugs (RTX or 5FdUR) for 48 h, stained with CCK8 (Dojindo), and absorbance at 450 nm was measured. The plotted relative survival values are based on at least three biological replicates (cf. also the Materials and Methods section). Data were plotted by Origin 8.6 (OriginLab Corporation). Time-course data and a detailed statistical analysis are provided in Fig S9.
Source data are available for this figure.

Cell cycle analysis experiments by flow cytometry (cf. the Materials and Methods section) were performed on UNG-inhibited, MMR-deficient HCT116 cells at 24-h treatments using different drug concentrations (0.1, 1, and 20 μM) for both drugs (note that the genome sequencing data were obtained from cells treated with 0.1 μM RTX or 20 μM 5FdUR [Pálinkás et al, 2020]). As compared to the nontreated and actively proliferating cell population (cf. orange label and inlet graph in Fig 6A), a strong S-phase arrest is observed (similar to Fig 5 in

Pálinkás et al [2020]) in all of the treated samples already at the lowest applied drug concentration (0.1 μM in Fig 6A). In response to RTX treatment, the proportion of cells in G0/G1 and S phases still exhibits a slight dose dependency, as expected. In contrast, in the case of the high-dose 5FdUR treatment, a small but reproducible shift can be observed toward a healthier phase distribution (more cells were in G1/G0 and G2/M phases, and fewer cells in the S phase). Summary data are presented in Fig 6A, and representative flowgrams in Fig S8.

**A**

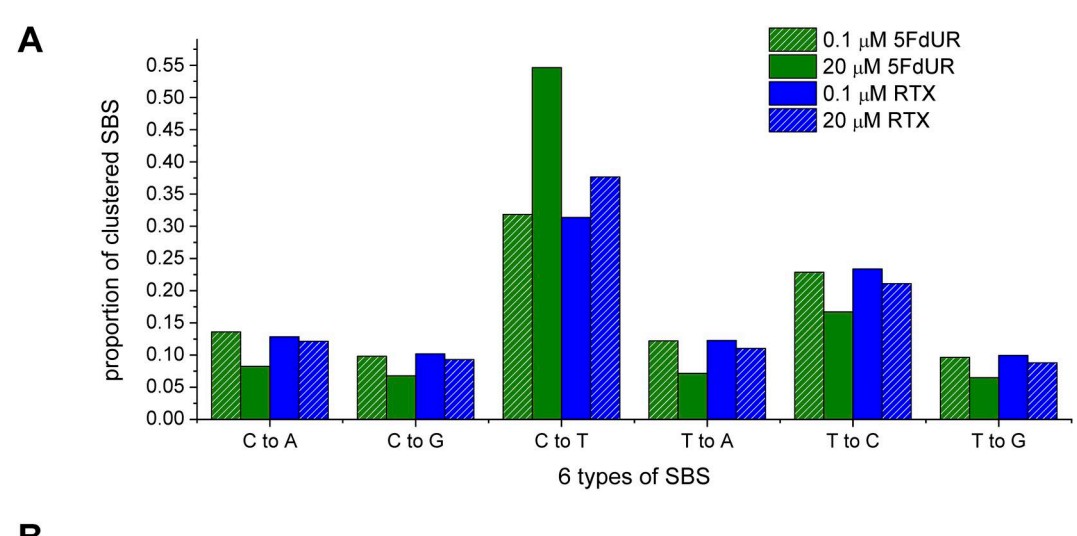

**B**

Consistently, an unusual dose dependency of the 5FdUR-induced cytostatic/cytotoxic effect was also detected when we monitored cell proliferation by the reducing capacity of living cells measured with the CCK8 assay (Dojindo) in a time- and dose-dependent manner (Figs 6B and S9). The lowest applied dose (0.01 $\mu$M) of either RTX or 5FdUR does not lead to a significant decrease in cell growth, except for the 72 h of 5FdUR treatments in both the MMR-deficient and MMR-proficient cells (cf. *P*-value panels in Fig S9). At a 0.1 $\mu$M dose, significant cytostatic effects are observed, which are similar for both drugs starting from the 48-h treatment onward. This effect is slightly weaker in MMR-proficient cells than in MMR-deficient ones, but the effects of RTX and 5FdUR still do not differ (Fig 6B). Interestingly, at 48 h, high doses (10 and 20 $\mu$M) of the 5FdUR treatment cause significantly less pronounced growth inhibition as compared to the lower (0.1 $\mu$M) dose of 5FdUR or any effective doses of RTX treatments (Fig 6B). This drug-specific difference moderately decreases, but it remains significant in the 72-h samples (Fig S9). MMR-proficient cells display similar tendencies, although the effects are slightly delayed in time: the growth inhibition at 48 h is minor in general, whereas the high-dose 5FdUR-specific milder effect is still clearly observable at 72 h (Fig S9).

In summary, low-dose (0.1 $\mu$M) treatments with both drugs produce similar effects throughout the experiment. However, at higher doses (10 or 20 $\mu$M), a reduced growth inhibition by 5FdUR becomes evident after 48 h in both MMR-proficient and MMR-deficient cells. A detailed statistical analysis was also performed to assess the significance of these differences, and *P*-values are reported for 320 pairwise comparisons (Fig S9).

### Drug-specific differences in phenotypes partially correlate with genomic variability

Given the intriguing phenomena of decreased cytostatic effects of high-dose 5FdUR treatment as compared to low-dose 5FdUR, we addressed the question of whether this correlates with the elevated frequency of C-to-T transitions. Hence, we performed new genome sequencing experiments on UNG-inhibited and MMR-deficient HCT116 cells after a 48-h treatment with high and low doses of the two drugs. Genetic variants were called from each of the four samples against the nontreated one using Mutect2 (cf. the Materials and Methods section). The frequency of clustered C-to-T transitions is dramatically increased again only in the case of a high-dose 5FdUR treatment (Fig 7A), which coincides with a milder phenotypic response. The triplet mutational spectrum is highly similar to that calculated from the U-DNA-seq input data (cf. Figs 4B and S5). Notably, these sequencing experiments were fully

independent of our previous study, yet the spectrum characteristics of high-dose 5FdUR treatment of DNA repair-deficient cells are reproducible and highly similar to the APOBEC-related spectrum, SBS2 (Fig 7B). The previously observed elevation of CC-to-TT double-base substitutions (cf. Figs S4 and S6) was also detected, potentially corresponding to the clustered nature of C-to-T transitions (Fig S10).

To assess the potential contribution of different AID/APOBEC family members based on their known sequence preferences, we calculated the pentanucleotide (1,536-type) mutational spectra of the clustered C-to-T transitions (Fig S11A). This analysis allowed us to exclude the involvement of AID, which preferentially targets the WRC motif (W = A or T; R = purine) and avoids the SYC coldspots (S = C or G; Y = pyrimidine) (Pham et al, 2003; Bransteitter et al, 2004). Most of the C-to-T transitions occurred in a TC context, consistent with the activity of APOBEC1 (Petersen-Mahrt & Neuberger, 2003; Beale et al, 2004; Saraconi et al, 2014) and several APOBEC3 family members (Harris et al, 2002; Langlois et al, 2005; Starrett et al, 2016; Ito et al, 2017). The substrate preferences of A1 and APOBEC3 enzymes have been extensively characterized both in vitro and in cellulo approaches, each with distinct advantages and limitations. Although in vitro assays using recombinant or engineered proteins on defined DNA substrates provide detailed structural and functional insights, they may underestimate the true diversity of genomic targets. Conversely, mutational spectra derived from genome sequencing of cells or patient samples better reflect physiological conditions but are complicated by uncontrolled expression of other APOBECs and regulatory factors. Most APOBECs, including A1 and several A3 enzymes, prefer the TC sites, with A3G targeting CCC in vitro (Schumacher et al, 2005), or YCC in viral cDNA (Langlois et al, 2005), a pattern that is also weakly visible in our pentanucleotide spectrum (violet box in Fig S11). A3A and A3B both prefer T at the −1 position (Hoopes et al, 2016), but differ at −2, favoring YTC and RTC motifs, respectively (Chan et al, 2015). A3F and A3D target TTC and WTC motifs, respectively (Sato et al, 2014), whereas A3C is more permissive at −1, producing a preferred WYC motif (Langlois et al, 2005)—consistent with the appearance of TCC sites in our spectra (light yellow box in Fig S11A). The A3H haplotype I footprint (CTCA) overlaps with motifs preferred by A3A and A1 (Starrett et al, 2016). Importantly, these partially overlapping sequence preferences can be further refined—or overwritten—by the DNA secondary structure (Buisson et al, 2019; McDaniel et al, 2020; Butt et al, 2024).

Recently, mutational spectra for A3A, A3B, and A3C were reported from transfected yeast cells (Fig 3B of Brown [2024]). Using these reference spectra, we performed a reconstitution analysis with the SigProfilerExtractor tool (Islam et al, 2022), treating the low-dose 5FdUR sample as a background (Fig S11B). This revealed

**Figure 7. Dose dependency of 5FdUR-induced C-to-T transitions.**
Genome sequencing experiments were performed on either nontreated (NT), RTX, or 5FdUR-treated, UGI-expressing MMR-deficient HCT116 cells. Both drugs were applied at 0.1 and 20 $\mu$M for 48 h. Variants were called using Mutect2 (McKenna et al, 2010; Van der Auwera et al, 2013) from the treated samples against the nontreated ones (as described in the Materials and Methods section). The corresponding filtered variant data are accessible through the GEO Series accession number GSE285767. **(A)** Increased frequency of C-to-T transitions occurs only in response to high-dose 5FdUR treatment. The six possible SBS types weighted by allele frequencies were counted for the clustered mutations in filtered VCF files, and their fractions relative to all SBS were calculated (as described in the Materials and Methods section). Data were plotted by Origin 8.6 (OriginLab Corporation). **(B)** Triplet mutational spectrum indicates APOBEC-related origin of 5FdUR-induced C-to-T transitions. Mutational signatures were calculated from the clustered mutations of filtered VCF files using SigProfiler (Bergstrom et al, 2019).
Source data are available for this figure.

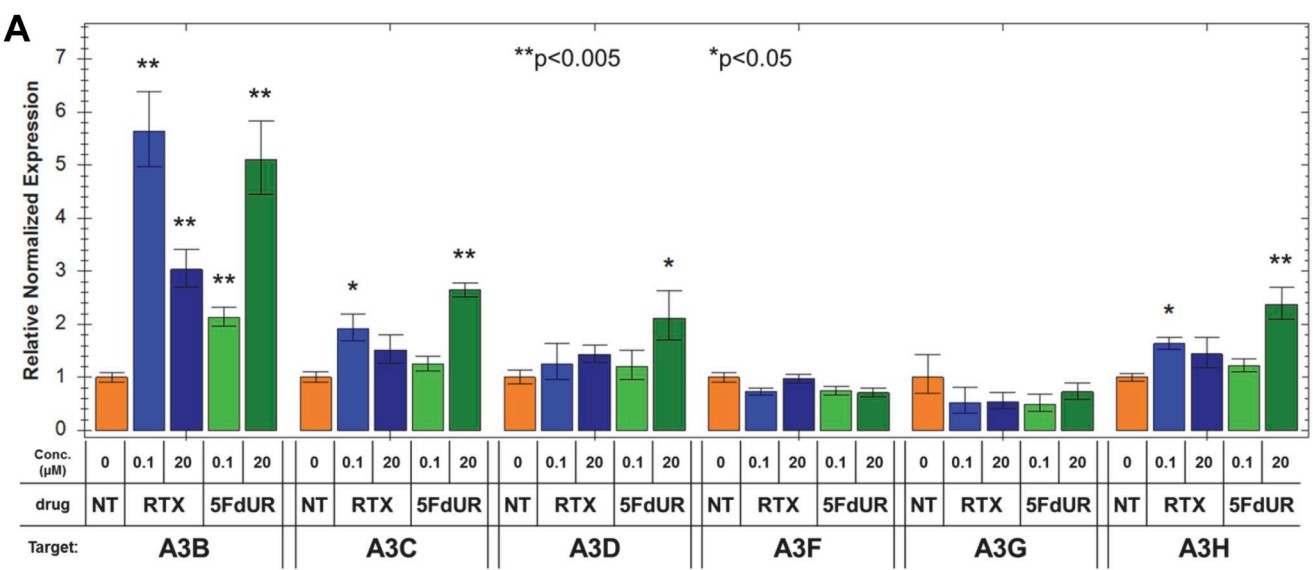

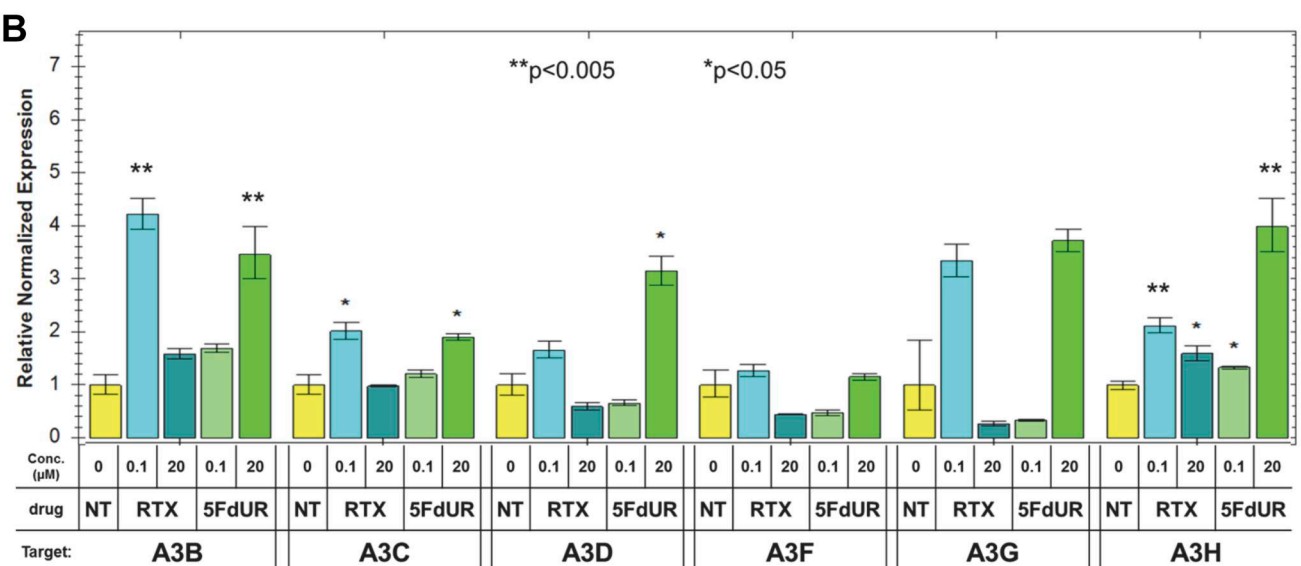

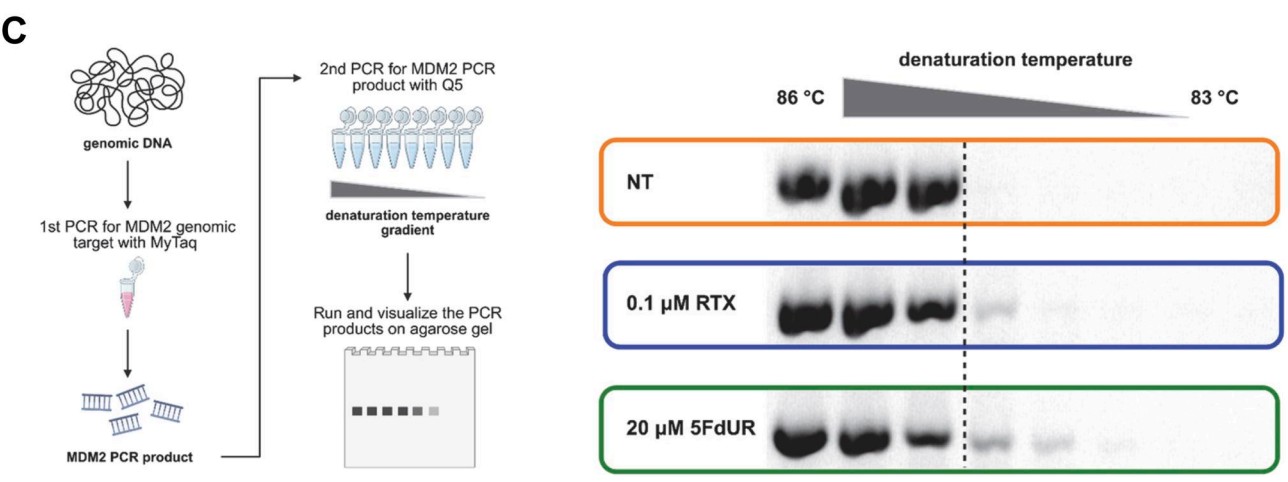

major contributions from A3B (35%) and A3C (38%), and no detectable A3A signature in the high-dose 5FdUR sample. In contrast, ~70% of the spectrum from 20 µM RTX-treated cells could be attributed merely to the background.

As A3B (and A3A also) was previously reported to prefer short hairpin loops 6–9 times more than nonhairpin DNA (Butt et al, 2024), we examined whether the local sequence context of the detected C-to-T transitions exhibited such secondary structures (see the Materials and Methods section and Supplemental Data 1). Our analysis demonstrated that 5FdUR-induced C-to-T transitions do not preferentially occur within predicted stem-loop regions (Fig S11C). Instead, these mutations display a pronounced tendency to cluster, a feature previously found to be more characteristic of A3C than A3B activity (Islam et al, 2022).

In summary, the above analysis of the measured mutational spectra (cf. Figs 4B, 7B, S5B, and S11), as well as the available expression profile of the HCT116 cell line (cf. long-read sequencing data from the ENCODE database, ENCFF242HKF [ENCODE Project Consortium, 2012]), focused our attention on the APOBEC3 family, whose activity on nuclear DNA and potential impact in carcinogenesis and/or cancer therapy have already been established.

### Some of the APOBEC3 family members are induced during drug treatments

To identify which APOBEC3 enzymes contribute to the 5FdUR-induced genomic C-to-T transitions, we performed quantitative reverse transcription-coupled real-time PCR (qRT-PCR) experiments. Specific primers for individual APOBEC3 members were used as described previously (Refsland et al, 2010), whereas *CNOT4* and *PUM1* served as reference genes (Rácz et al, 2021) (cf. the Materials and Methods section). In UNG-inhibited HCT116 cells, *A3A* expression could not be detected, as no specific PCR product was obtained, suggesting a negligible transcript level. This result is consistent with previously published long-read sequencing (ENCFF242HKF [ENCODE Project Consortium, 2012]) and qRT-PCR data (Periyasamy et al, 2021). In contrast, measurable mRNA expression was observed for the remaining APOBEC3 enzymes, with significant induction of *A3B*, *A3C*, *A3D*, and *A3H* in drug-treated samples (Fig 8A). Technical details of RNA quality, PCR specificity, and efficiency of reverse transcription and PCR are provided in Fig S12, and reproducibility across at least three biological replicates is demonstrated in Fig S13.

In DNA repair-deficient HCT116 cells, *A3C*, *A3D*, and *A3H* exhibit a 5FdUR-biased, minor increase, whereas the level of *A3B* mRNA is dramatically elevated after both RTX and 5FdUR treatments (Figs 8A and S13A). Notably, MMR-proficient UNG-inhibited HCT116 cells display similar tendencies: although *A3B* induction is a bit weaker

after both drug treatments, *A3D* and *A3H* are induced by 5FdUR more dramatically, increasing the drug-specific difference as well (Figs 8B and S13B). The drug-induced increase in APOBEC3 levels, together with the absence of increased C-to-T transitions in MMR-proficient cells, suggests that intact MMR efficiently repairs the U/T:G mismatches generated by APOBEC activity. Interestingly, the high-dose RTX reduces rather than enhances APOBEC3 induction.

The indicated APOBEC-mediated cytosine deamination was further examined using both a cell-free activity assay and 3D-PCR analysis targeting genomic DNA. In cell-free cytoplasmic and nuclear extracts, cytidine deamination activity was detected independently of drug treatments (Fig S14), which is consistent with the expected background activity in a cancer cell line expressing several APOBECs under basal conditions. However, this in vitro assay (based on an oligonucleotide substrate and potentially affected by dysregulated APOBEC or nonspecific nuclease activities) cannot reliably model the true genome-editing potential of cellular APOBECs. Therefore, we applied differential DNA denaturation PCR (3D-PCR) to a defined genomic region within the *MDM2* gene (Stenglein et al, 2010; Hultquist et al, 2011). We observed a slight decrease in the limiting denaturation temperature in the 5FdUR-treated sample compared with the RTX-treated and untreated samples (Fig 8C), indicating the presence of a minor fraction of genomic templates with elevated mutation load.

We next sought to estimate the contribution of APOBEC3 enzymes, induced at the mRNA level, to the observed genome-editing activity. Because A3B mRNA displayed similarly strong induction in response to both drug treatments, its selective contribution to the 5FdUR-specific deamination events likely involves additional regulation at the protein or chromatin level. To examine this, we analyzed the subcellular localization pattern of A3B protein by immunocytochemistry and found comparable nuclear accumulation after both treatments (Fig 9A and B). This observation suggests that further regulatory mechanisms, such as protein-protein interactions, posttranslational modifications, or chromatin accessibility, may influence A3B's genomic activity. Given these findings, we also considered the possible role of the other APOBEC3 enzymes, including A3C, A3D, or A3H, which displayed minor but 5FdUR-biased induction (cf. Fig 8). A3C, in particular, drew our attention because of its relatively high basal expression in HCT116 cells (cf. ENCODE data and our qRT-PCR Cq values) and reconstitution of the mutational spectrum (cf. Fig S11), which supports its contribution to the observed deamination pattern. We therefore examined A3C and A3D protein levels in subcellular fractions of untreated and drug-treated cells using specific antibodies (Fig 9C–E). A3C exhibited a modest increase in the chromatin-bound nuclear fraction in response to 20 µM 5FdUR treatment, whereas A3D levels remained unchanged.

**Figure 8.   Induction of APOBEC3 enzyme family members.**
**(A, B)** Gene expression levels of APOBEC3s in MMR-deficient, UNG-inhibited cells (A) and MMR-proficient, UNG-inhibited HCT116 cells (B). The mRNA levels of APOBEC3 family members (*A3B, A3C, A3D, A3F, A3G, A3H*) were measured by qRT-PCR after 48 h of treatment with low (0.1 µM) and high doses (20 µM) of RTX and 5FdUR. The relative normalized expression values were calculated as described in the Materials and Methods section using reference genes *CNOT4* and *PUM1*. Statistical analysis was performed on at least three biological replicates using CFX Maestro (Bio-Rad); * and ** represent *P*-values below 0.05 and 0.005, respectively. **(C)** Endogenous APOBEC3 activity at *MDM2* genomic target. 3D-PCR (scheme on the left) was used to detect genomic deamination events, as reflected in the decreased denaturation temperature (gradient indicated by the triangle at the top) of the PCR. Products were analyzed on 1% agarose gel (right); the specific PCR product, as expected, appears at 570 bp apparent size. Uncropped gel images and ladder positions are provided in the Source Data File.
Source data are available for this figure.

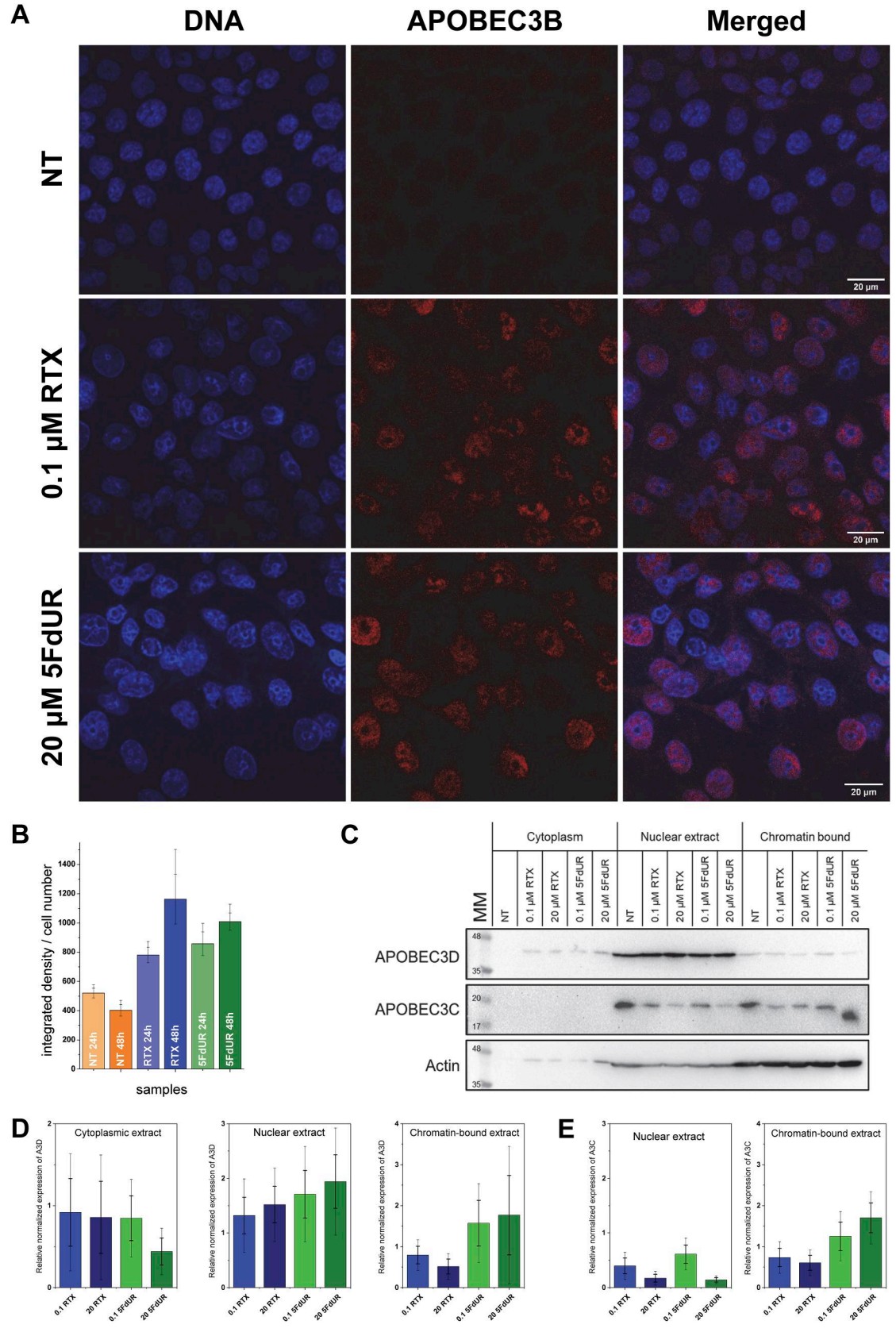

A3H could not be reliably detected with the available antibody. Although densitometric analysis of three biological replicates did not reveal statistically significant changes, the observed trend for A3C is consistent with its potential involvement in 5FdUR-specific genomic cytosine deamination. Because of the semi-quantitative nature of immunocytochemistry and Western blotting, as well as the limited sensitivity of commercially available antibodies, small endogenous changes in these enzymes remain challenging to quantify.

Given these technical limitations, we next sought indirect support for the potential relevance of APOBEC3 expression. Hence, we aimed to assess which APOBEC3 of our candidates might have higher biological or clinical significance in colon adenocarcinoma. We queried publicly available clinical datasets to determine whether APOBEC3 expression correlates with patient outcome in a broader clinical context. Using colon adenocarcinoma data from The Cancer Genome Atlas (TCGA) analyzed via the online tool, TCGExplorer (Kus et al, 2023 Preprint), we generated Kaplan-Meier plots for relapse-free survival based on APOBEC3 expression levels. Elevated *A3C* expression correlated with significantly shorter relapse-free survival, whereas other APOBEC3 family members presented no significant associations (Fig S15). These findings highlight A3C as a promising candidate for further investigation into APOBEC-driven genomic instability and drug response.

# Discussion

In the present study, we demonstrated that the two thymidylate synthase inhibitors, RTX and 5FdUR, act in markedly different ways in DNA repair-deficient HCT116 colorectal cancer cells, particularly with respect to genomic uracilation, induced mutagenicity, and phenotypic responses. These drug-specific differences, further modulated by cellular DNA repair capacity, form a coherent pattern that provides insight into distinct cellular consequences of TS inhibition and may have implications for personalized cancer therapy.

Firstly, using a genome segmentation approach, we identified two genomic segments that were differentially uracilated during the two drug treatments in UNG-inhibited, MMR-deficient, or MMR-proficient cells. We also introduced the U-score as a quantitative measure of gene-level uracil enrichment and identified groups of genes with differential uracilation. Furthermore, we described functional coherence among genes exhibiting the highest or the most differential uracilation in response to 5FdUR or RTX

treatment, highlighting drug-dependent differences that are also affected by the MMR status.

Secondly, we observed an increased frequency of genomic C-to-T transitions specifically in MMR-deficient and UNG-inhibited HCT116 cells treated with 5FdUR. A thorough analysis of mutational signatures, intermutational distances, and replication timing dependencies indicated enzymatic cytosine deamination as the possible source of this drug-specific difference. Indeed, we measured increased mRNA levels of several APOBEC3 DNA cytidine deaminases (A3D, A3C, A3H) with a bias toward 5FdUR treatment, whereas A3B exhibited a substantial increase after both drug treatments. In addition, the induction and nuclear localization of A3B protein were also demonstrated in both cases. These observations suggest that APOBEC induction alone is insufficient to explain mutagenesis and that additional regulatory mechanisms determine whether APOBEC activity translates into detectable genomic mutations.

Thirdly, cell cycle arrest and cell viability measurements revealed an unusual dependency on the applied doses of 5FdUR treatment in both MMR-deficient and MMR-proficient, UNG-inhibited HCT116 cells. Specifically, high-dose 5FdUR exhibited reduced cytotoxicity compared with lower doses of 5FdUR or any effective doses of RTX. This co-occurrence of increased genomic variability and decreased cytostatic potential was further supported by additional genome sequencing experiments performed from DNA repair-deficient cells treated with varying doses of RTX and 5FdUR.

The **functional coherence among the most or the most differentially uracilated genes supports a functional role of genomic uracilation in shaping cellular drug responses**. In response to 5FdUR treatment, the networks of genes with the top 200 U-scores displayed higher interconnectivity compared with the RTX treatment (cf. Table 2 and Fig S3). The enrichment of transcription factor genes—particularly homeobox transcription factors that typically regulate developmental processes—is evident among the most or the most differentially uracilated genes in 5FdUR-treated samples, in contrast to RTX-induced uracilation, which largely avoids these genes (cf. Figs 3 and S3, and Source Data File). In addition, 5FdUR treatment causes biased uracilation of genes encoding immunoglobulins, ion channels, membrane receptors, and cadherins, all of which are involved in extracellular processes and cell-cell communication (cf. Fig 3). The lower interconnectivity of the RTX networks may be explained by the stronger dependency of uracil incorporation on replication timing (cf. Fig 1), also supported by the stronger S-phase arrest (cf. Fig 6A). Hence, the enriched functional terms reflect more diverse

**Figure 9. Expression and localization of A3B, A3C, and A3D proteins.**
**(A)** Expression and localization of A3B. UNG-inhibited HCT116 cells were not treated (NT) or treated with either 0.1 $\mu$M RTX or 20 $\mu$M 5FdUR for 48 h and stained with anti-A3B antibody (ab222330; red; Abcam) and DAPI (blue) as described in the Materials and Methods section. The scale bar corresponds to 20 $\mu$m. **(B)** Quantification of A3B protein levels. Integrated fluorescence intensities were quantified using ImageJ (Fiji [Schneider et al, 2012]) and normalized to manually counted cell numbers from at least three images across three biological replicates. Error bars indicate SD and SEM. **(C)** Representative Western blot demonstrating the protein levels and subcellular localization of A3C and A3D. The same cell line and treatments were applied for 48 h. Cells were lysed and fractionated as described in the Materials and Methods section. Actin was used as a loading control. **(D, E)** Quantification of A3D (D) and A3C (E) Western blots. Densitometric analysis was performed using ImageLab (Bio-Rad). Adjusted volumes were normalized to actin and expressed relative to NT, yielding the relative normalized expression levels shown in the bar graphs. Each expression value represents the mean of at least three biological replicates. Graphs were created by Origin 8.6 (OriginLab Corporation).
Source data are available for this figure.

processes, for example, RNA processing, cell cycle regulation, stress response, and intracellular trafficking. Among these, the RNA-related, especially the splicing-related, functionalities appear to be recurrent (cf. Source Data File), also appearing among the top 200 most uracilated genes (cf. Fig S3A and C). In contrast, in the 5FdUR-treated samples, genes involved in RNA transport are enriched in the low U-score range (cf. Source Data File). Moreover, the sRNA genes known to be involved in splicing and RNA processing are rather depleted within the top 2000 most uracilated genes in the 5FdUR compared with the RTX-treated samples (cf. Fig 2B).

Overall, these functional enrichments might reflect the cellular response to the two drug treatments. Several mechanisms may contribute to the observed genomic uracil patterns: (1) thymine-replacing uracil incorporation during active DNA synthesis, linked to either replication or repair; (2) enzymatic deamination of cytosines; and (3) uracil-DNA repair through base excision repair (BER) or MMR for U:G mispairs. These processes are further shaped by chromatin accessibility and global cellular stress response. In these S phase-arrested and UNG-inhibited cells, replication timing clearly plays a dominant role in shaping uracilation patterns. The early replicated segments harbor higher levels of uracilation, which likely explains the high U-scores of certain gene clusters. For example, metallothionein genes exhibit correlated uracilation most likely because they are located in a single cluster on chromosome 16 and may therefore be regulated by similar processes (cf. Fig S3). However, additional factors likely contribute, particularly **at mutational hotspots** or transcriptionally active regions. One obvious source of genomic uracil enrichment may be the **MMR-coupled DNA synthesis** in response to U:G (or 5FU:G) mispairs in MMR-proficient cells, as indicated by the major MMR-dependent divergence of uracil pattern in 5FdUR-treated cells. Notably, the most uracilated segments M2(R5) in 5FdUR-treated, MMR-proficient and MMR-deficient cells are the same, despite significant differences in their overall uracil signal distribution (cf. Fig 1). This discrepancy cannot be explained by the dependency on replication timing or increased frequency of C-to-T transitions, and this segment appears unaffected by the cellular MMR status. We considered that **local repair-coupled synthesis**, potentially **associated with transcriptional activity**, might contribute to the elevated uracil signal in certain genomic segments (Owiti et al, 2018). Although this interpretation remains speculative, such processes might contribute to uracil enrichment of segment M2(R5) (along with M9) consistently with its GC-rich, gene-dense features (cf. Figs 1, 5B, and S2). However, evaluating the potential correlation between uracilation and transcription is complex, as the arising genomic uracils might also influence transcription, either positively or negatively (Rogstad et al, 2002; Cui et al, 2019; Yang et al, 2021). Thus, genomic uracilation may simultaneously act as both a cause and a consequence of transcriptional changes during the mechanism of drug actions.

In the present study, we focus on **the role of cellular DNA repair abilities, particularly the MMR status, which has a greater influence on the genomic uracil pattern after 5FdUR as compared to RTX treatment**. This phenomenon can be attributed to the 5FdUR-induced cytosine deamination. Cytosine deamination can directly contribute to the genomic uracil-DNA pattern; however,

this contribution might be minor compared with the massive thymine-replacing uracil incorporation in these cells, where both TS and UNG are inhibited. It is further confirmed by the observation that segment M5(R3), the most affected by C-to-T transitions, is not the most uracilated one in the 5FdUR sample; in contrast, segment M2(R5), the most uracilated one, is less affected by deamination (Fig 5B). The induced cytosine deamination might result in rather indirect effects involving MMR activation. Although the mutagenic effects are effectively mitigated in MMR-proficient cells, the pattern of APOBEC3 expression and the observed increase in drug tolerance remain largely unchanged. We propose that the 5FdUR-induced APOBEC3s gaining access to the genome resulted in the accumulation of U:G pairs and extensive repair synthesis in MMR-proficient cells. It is well established that repair of a single mismatched base pair by MMR entails resynthesis of a long stretch of DNA (several thousand nucleotides), carried out by processive replicative polymerases that may remain available in cells arrested in the S phase. These conditions may allow extensive repair synthesis coupled with uracil incorporation, potentially overriding the strong replication timing dependency found in the MMR-deficient cells (Fig 1). This mechanism could also contribute to the higher inter-replicate variability of U-DNA patterns and explain the previously reported elevated global genomic uracil content in 5FdUR-treated, MMR-proficient cells (Pálinkás et al, 2020). In contrast, in the case of RTX treatment, MMR proficiency only moderately increased the global uracil content (Pálinkás et al, 2020), and the genomic uracil patterns, low variability, and straightforward concentration dependency remained unchanged, indicating the absence of additional MMR substrates.

MMR substrates, however, might also arise during 5FdUR treatment if 5FU is incorporated into the genomic DNA, likely because of increased cellular levels of the 5FdUTP metabolite. This incorporation can occur opposite to G and A with approximately equal probability, resulting in 5FU:G mispairs (Meyers et al, 2005) that serve as substrates for cellular MMR (Meyers et al, 2005; Li et al, 2009; Pettersen et al, 2011). Notably, 5FU in DNA can be recognized and processed similarly to uracil by all known human UDGs: UNG (Pettersen et al, 2011), SMUG1 (An et al, 2007), TDG (Hardeland et al, 2001; Liu et al, 2002), and MBD4 (Petronzelli et al, 2000), with their activity influenced by the DNA structure and the base-pairing context. In HCT116 cells treated with 5FdUR, a dose-dependent increase in genomic 5FU content has already been demonstrated, which decreased significantly when MMR was restored (Fig 6A in Meyers et al [2005]). In normal HCT116 cells (where UNG efficiently erases uracil–DNA during 5FdUR treatment [Róna et al, 2016]), G2 arrest was reported, which was even more pronounced and associated with increased drug sensitivity in the MMR-proficient version of this cell line (Meyers et al, 2001, 2003, 2004). Here, in our model, where UNG is efficiently inhibited, and genomic uracil content is exceptionally high (reaching several hundred uracils per million base pairs [Pálinkás et al, 2020]), both the MMR-dependent sensitivity and the G2 arrest phenomenon are absent. Instead, the S-phase arrest becomes more pronounced (cf. Fig 6A), consistent with a previous study reporting that UNG inhibition sensitizes cancer cells to 5FdUR through S-phase arrest and replication fork collapse-induced DNA damage (Yan et al, 2016). We argue that in our UNG-inhibited and MMR-deficient HCT116 cellular model, 5FU

appears to account for only a minority of the thymine-replacing uracil incorporations. If 5FU incorporation were a significant factor, its ambiguous base-pairing properties would lead to a similar increase in both C-to-T and T-to-C transitions. However, the 5FdUR-specific mutational spectra do not support this scenario (cf. Figs 4A and B, 7, and S5). Instead, all spectra exhibit minor background elevation in both T-to-C and C-to-T transitions after both treatments.

The 5FdUR-specific mutational spectra, however, exhibit high similarity to the APOBEC-related SBS2 COSMIC signature (cf. Figs 4B, 7B, S5B, and S11). The accumulation of short intermutation distances in response to high-dose 5FdUR treatment (cf. Fig 4C) further supports the activation of APOBEC enzymes in our model HCT116 cells. Based on available long-read RNA sequencing data (ENCFF242HKF [ENCODE Project Consortium, 2012]), nontreated HCT116 cells express several members of APOBEC3s (annotated and novel isoforms): A3C at the highest level, A3B at a moderate level, and A3D, A3F, A3G, and A3H haplotype I at lower levels. Other members of the AID/APOBEC family were not detected in this sequencing dataset; moreover, A3D was represented by only a short fragment of its annotated transcript. **Hence, we focused on the APOBEC3 subfamily, measuring their relative expression levels** in response to drug treatments (cf. Fig 8). A3B, one of the most studied APOBEC3s involved in carcinogenesis (Burns et al, 2013; Butt et al, 2024), exhibits the highest induction after both drug treatments, which is also confirmed by immunocyto-chemistry revealing similarly elevated nuclear protein level (cf. Fig 9). In contrast, the genomic C-to-T frequency is selectively increased in response to high-dose 5FdUR treatment, which can only be explained by additional regulation of A3B. Alternatively, the other APOBEC3 members that display minor but more 5FdUR-biased increase (A3C, A3D, and A3H) might also be responsible for the measured mutational spectra. Intriguingly, A3A, the other most well-known deaminase implicated in tumorigenesis (Petljak et al, 2019; Isozaki et al, 2023), was not detected in our qRT-PCR setup, which is also in agreement with ENCFF242HKF data (ENCODE Project Consortium, 2012). Although ENCFF242HKF data do not support intact A3D expression either, we could reliably measure it by qRT-PCR, which indicates limited sensitivity of the ENCFF242HKF data, restricted by the sequencing depth. Notably, the absence of A3A and the presence of A3D in HCT116 cells have already been confirmed by qRT-PCR in an independent study, where drug-induced expression of several APOBEC3 enzymes in HCT116 cells was also reported (Periyasamy et al, 2021).

Here, we demonstrated that upon similar induction and nuclear localization of A3B, the mutagenic effect on the genome might be restricted and strongly depend on additional regulatory mechanisms. In accordance, many tumors do not display increased SBS2- or SBS13-type C-to-T mutations despite high levels of A3A or A3B expression (Mertz et al, 2022). A3B was demonstrated to deaminate cytosines on the lagging strand during replication, and an increase of ssDNA caused by replication stress might explain the increased mutagenesis in cancer cells (Hoopes et al, 2016; Duan et al, 2020). In our cell line model, both RTX and 5FdUR cause S-phase arrest (stronger in the case of the RTX, cf. Fig 6A) and replication stress, as indicated by the elevated γH2AX signal (Pálinkás et al, 2020), which again fails to explain the 5FdUR-specific elevation of genomic C-to-T transitions. Posttranslational modifications (Matsumoto et al,

2019) or accumulation in heterogeneous large multimeric complexes involving RNA binding (Chen, 2021) can obviously hinder the cytidine deaminase activity of A3s (Adolph et al, 2017, 2018). Moreover, DNA accessibility may also depend on intrinsic features of chromatin architecture, which could be differentially affected during cellular responses to the two drug treatments. Certain members of the AID/APOBEC family were characterized with specific affinities toward certain DNA structures like A3A and A3B to hairpin loops (Butt et al, 2024), AID to G-quadruplexes (Zheng et al, 2015), and A3B to R-loops (McCann et al, 2023). In addition, the **increased level of the less well-characterized APOBEC3 enzymes, A3C, A3D, and A3H, may also contribute to mutagenic effects**. Indeed, dysregulation of A3C (Qian et al, 2022) and haplotype I of A3H (Starrett et al, 2016) has already been shown to contribute to cancer-related genomic mutations. Although A3C is less characterized, it is the most expressed APOBEC in many tumors (Zhang et al, 2023) and in naive HCT116 cells. Importantly, A3C contribution to the induced mutagenesis in our HCT116 model might also be pronounced as it is suggested by the reconstitution of the measured mutational spectra from published characteristic signatures of A3B and A3C (cf. Fig S11), which is further supported by 5FdUR-induced expression with chromatin-bound localization (cf. Fig 9C), and detectable impact in cancer progression reflected in patient survival (Fig S15).

Besides this pronounced elevated APOBEC mutational trait characteristic of 5FdUR treatment, it is worth noting that a background cytosine deamination might also be ongoing even in untreated HCT116 cells, potentially driven by basal APOBEC3 activity or spontaneous events. This is suggested by the slightly elevated C-to-T and T-to-C transitions observed in the background of all our SBS triplet mutational spectra (cf. Figs 4, 7, and S5). Although this phenomenon might originate from oxidative DNA damage (Lózsa et al, 2023), the contribution of an ongoing CpG deamination is also evidenced by the unique TG-to-CA peak found in each of our DBS spectra (cf. Figs S4, S6, and S10). Such CpG deamination can result in four different dinucleotide outcomes: CG (nonmutated or repaired), TG and CA (asymmetrically deaminated), and TA (symmetrically deaminated). In heterogeneous cell cultures, these variants coexist at a given site with certain frequencies in both control and test samples, allowing the detection of multiple variants (Table S2). Among these, the TG-to-CA is the only DBS specifically associated with asymmetric CpG deamination. Interestingly, in the 5FdUR samples, the number of such DBSs was consequently 1.5-fold higher than in the corresponding RTX ones, suggesting a potential contribution of the 5FdUR-induced APOBEC3 activity to CpG deamination as well.

Apparently, these **deaminated cytosines at CpG sites are not efficiently repaired even in the MMR-proficient cells**. In our cellular models, the activity of both UNG and another uracil-DNA glycosylase, MBD4, is compromised. HCT116 cells are heterozygous for a frameshift mutation in MBD4, which normally encodes a UDG specialized in repairing U:G and T:G mismatches at CpG or methyl-CpG sites (Bader et al, 1999; Abdel-Rahman et al, 2008). The mutant MBD4 protein lacks both UDG activity and MLH1-binding capacity but retains methyl-CpG binding, acting in a dominant-negative manner by competing with WT MBD4 and other methyl-CpG–binding proteins (Bader et al, 1999). This likely protects U:G (or T:G) mismatches from repair. These findings raise the question of how DNA methylation influences APOBEC3-mediated cytosine

**Life Science Alliance**

**Table 3. Sequenced samples.**

| Sample | MMR status | U-DNA BER status | Drug treatment | Drug conc. (μM) | |
|---|---|---|---|---|---|
| WT | Deficient (mutant Mlh1) | Normal | None | 0 | U-DNA-seq (GSE126822) GSE285931 |
| NT | | UNG is inhibited by the transgene UGI | None | 0 | |
| RTX | | | RTX | 0.1 | |
| 5FdUR | | | 5FdUR | 20 | |
| NT _MMR | Proficient (+wt chr3) | | None | 0 | |
| RTX _MMR | | | RTX | 0.1 | |
| 5FdUR _MMR | | | 5FdUR | 20 | |
| **NT** | **Deficient (mutant Mlh1)** | **UNG is inhibited by the transgene UGI** | **None** | **0** | WGS, GSE285767 |
| **RTX _01** | | | **RTX** | **0.1** | |
| **RTX _20** | | | **RTX** | **20** | |
| **5FdUR _01** | | | **5FdUR** | **0.1** | |
| **5FdUR _20** | | | **5FdUR** | **20** | |

deamination. A3A and A3H efficiently deaminate methylated cytosines, whereas A3B does so with lower efficiency (Ito et al, 2017). Therefore, we suggest that cytosine methylation shapes the genomic distribution of C-to-T transitions less strongly than the uracil incorporation pattern itself. This may explain the apparent discrepancy where segment M5(R3) exhibited the highest frequency of C-to-T transitions without corresponding uracil enrichment (see Fig 5B).

We observed a coincidence between decreased cytotoxicity after high-dose 5FdUR treatment (cf. Fig 6) and 5FdUR-induced APOBEC-mediated genomic cytosine deamination in our UNG-inhibited HCT116 cell line model. Such impaired drug efficacy, detectable after just 48 h of treatment, may contribute to the emergence of clinically relevant drug-resistance mechanisms. The role of AID/APOBEC enzymes and their mutagenic activity in the emergence of drug resistance is well established (Venkatesan et al, 2018; Periyasamy et al, 2021; Mertz et al, 2022; Isozaki et al, 2023). In addition to promoting mutagenesis and selecting more viable clones, APOBEC enzymes may directly facilitate adaptation and resistance over short periods, as represented in cell line models. For instance, A3B depletion has been demonstrated to re-sensitize resistant cells to cisplatin and enhance sensitivity to several chemotherapy drugs (Periyasamy et al, 2021). Similarly, A3B promoted tamoxifen resistance in ER-positive breast cancer (Law et al, 2016), whereas a CRISPR screen identified A3C and A3D as contributors to gemcitabine resistance in pancreatic cancer, with their knockout restoring drug sensitivity (Ubhi et al, 2024). Conversely, APOBEC activity can also enhance drug efficacy in some contexts, such as mediating the effects of cisplatin (Conner et al, 2020). In addition, APOBEC-mediated mutations may generate neoantigens

that, at least temporarily, might help the immune system recognize malignant cells (Driscoll et al, 2020). Supporting this complexity, A3D has been identified as a positive prognostic biomarker in breast cancer patients (Chen et al, 2022). In our analysis of TCGA colon adenocarcinoma data, only A3C expression is negatively correlated with patient survival (cf. Kaplan-Meier plots in Fig S15). Our findings suggest a potential causative relationship between induced APOBEC3 expression and drug tolerance.

In summary, genomic uracil patterns induced by RTX and 5FdUR treatments are essentially different and can be well explained in light of the differently induced mutagenicity and altered cell response. Genomic uracilation may simultaneously be influenced by and affect the cellular response, implying replication arrest, and regulation of transcriptional and DNA repair activities. Therapy-induced APOBEC3 expression manifested differently in genomic C-to-T mutations in response to the two TS-inhibitory treatments. The 5FdUR-specific increase of these mutations coincided with increased drug tolerance already in a short-term treatment. However, further investigation is required to confirm this link and to explore how reduced cytotoxicity contributes to drug resistance in clinical settings.

# Materials and Methods

### Sequencing data

Sequenced samples are summarized in Table 3. The original U-DNA-seq data were generated in our previous study (Pálinkás

**Table 4.** Names and the accession codes of raw sequencing data used in this study.

| Short name | Sample | Type of sample | SRA experiment ID | Variation calling | U-score calculation |
|---|---|---|---|---|---|
| 5FdUR_rep1_son | 5FdUR_UGI_HCT116_rep1_son | Input | SRX7073986 | + | + |
| 5FdUR_rep2_son | 5FdUR_UGI_HCT116_rep2_son | Input | SRX7073988 | + | + |
| RTX_rep1_son | RTX_UGI_HCT116_rep1_son | Input | SRX7073990 | + | + |
| RTX_rep2_son | RTX_UGI_HCT116_rep2_son | Input | SRX7073992 | + | + |
| NT_rep1_son | NT_UGI_HCT116_rep1_son | Input | SRX7073994 | +, control | + |
| NT_rep2_son | NT_UGI_HCT116_rep2_son | Input | SRX7073996 | +, control | + |
| 5FdUR_MMR_rep1_son | 5FdUR_UGI_HCT116MMR_rep1_son | Input | SRX8631168 | + | + |
| 5FdUR_MMR_rep2_son | 5FdUR_UGI_HCT116MMR_rep2_son | Input | SRX8631170 | + | + |
| RTX_MMR_rep1_son | RTX_UGI_HCT116MMR_rep1_son | Input | SRX8631172 | + | + |
| RTX_MMR_rep2_son | RTX_UGI_HCT116MMR_rep2_son | Input | SRX8631174 | + | + |
| NT_MMR_rep1_son | NT_UGI_HCT116MMR_rep1_son | Input | SRX8631176 | Control | + |
| NT_MMR_rep2_son | NT_UGI_HCT116MMR_rep2_son | Input | SRX8631178 | Control | + |
| WT_rep1_son | WT_HCT116_rep1_son | Input | SRX7073998 | Control | + |
| WT_rep2_son | WT_HCT116_rep2_son | Input | SRX7070400 | Control | + |
| 5FdUR_rep1_IP | 5FdUR_UGI_HCT116_rep1_IP | Enriched | SRX7073985 | – | + |
| 5FdUR_rep2_IP | 5FdUR_UGI_HCT116_rep2_IP | Enriched | SRX7073987 | – | + |
| RTX_rep1_IP | RTX_UGI_HCT116_rep1_IP | Enriched | SRX7073989 | – | + |
| RTX_rep2_IP | RTX_UGI_HCT116_rep2_IP | Enriched | SRX7073991 | – | + |
| NT_rep1_IP | NT_UGI_HCT116_rep1_IP | Enriched | SRX7073993 | – | + |
| NT_rep2_IP | NT_UGI_HCT116_rep2_IP | Enriched | SRX7073995 | – | + |
| 5FdUR_MMR_rep1_IP | 5FdUR_UGI_HCT116MMR_rep1_IP | Enriched | SRX8631167 | – | + |
| 5FdUR_MMR_rep2_IP | 5FdUR_UGI_HCT116MMR_rep2_IP | Enriched | SRX8631169 | – | + |
| RTX_MMR_rep1_IP | RTX_UGI_HCT116MMR_rep1_IP | Enriched | SRX8631171 | – | + |
| RTX_MMR_rep2_IP | RTX_UGI_HCT116MMR_rep2_IP | Enriched | SRX8631173 | – | + |
| NT_MMR_rep1_IP | NT_UGI_HCT116MMR_rep1_IP | Enriched | SRX8631175 | – | + |
| NT_MMR_rep2_IP | NT_UGI_HCT116MMR_rep2_IP | Enriched | SRX8631177 | – | + |
| WT_IP_rep1_IP | WT_HCT116_rep1_IP | Enriched | SRX7073997 | – | + |
| WT_IP_rep2_IP | WT_ HCT116_rep2_IP | Enriched | SRX7070399 | – | + |
| **NT** | **NT_UGI_HCT116** | **Genomic** | **SRX27236862** | **Control** | **–** |
| **RTX_01** | **RTX_0p1_UGI_HCT116** | **Genomic** | **SRX27236863** | **+** | **–** |
| **RTX_20** | **RTX_20_UGI_HCT116** | **Genomic** | **SRX27236864** | **+** | **–** |
| **5FdUR_01** | **5FdUR_0p1_UGI_HCT116** | **Genomic** | **SRX27236865** | **+** | **–** |
| **5FdUR_20** | **5FdUR_20_UGI_HCT116** | **Genomic** | **SRX27236866** | **+** | **–** |

et al, 2020) and deposited in GEO under accession GSE126822, and were reanalyzed in this study under GSE285931. These data were complemented by new, dose-dependent whole-genome sequencing data generated in this study (GEO accession GSE285767).

For all calculations, we started from the cleaned, uniquely aligned reads as defined in Pálinkás et al (2020). Usage of the data for variation calling and/or U-score calculation is indicated in Table 4 by "+" or "–" in the corresponding columns.

## Genome segmentation according to uracil enrichment signals

The genome segmentation analysis was performed on both the merged data (cf. Fig 1A) and the individual replicates (cf. Fig S1A) of the previously published U-DNA-seq data (cf. Tables 4 and 5) using the Segway software package (Hoffman et al, 2012; Chan et al, 2017) as described in Supplementary file 3 of the reference (Pálinkás et al, 2020). Briefly, the corresponding "fold change over control" files of U-DNA-seq data (merged or individual replicates, cf. GEO Series GSE126822) were combined into a single Genomedata file

**Table 5.   Primers for qRT-PCR measurements.**

| Gene | Forward primer | Reverse primer | Reference |
|---|---|---|---|
| APOBEC3A | gagaagggacaagcacatgg | tggatccatcaagtgtctgg | Refsland et al (2010) |
| APOBEC3B | gaccctttggtccttcgac | gcacagccccaggagaag | Refsland et al (2010) |
| APOBEC3C | agcgcttcagaaaagagtgg | aagtttcgttccgatcgttg | Refsland et al (2010) |
| APOBEC3D | acccaaacgtcagtcgaatc | cacatttctgcgtggttctc | Refsland et al (2010) |
| APOBEC3F | ccgtttggacgcaaagat | ccaggtgatctggaaacactt | Refsland et al (2010) |
| APOBEC3G | ccgaggacccgaaggttac | tccaacagtgctgaaattcg | Refsland et al (2010) |
| APOBEC3H | agctgtggccagaagcac | cggaatgtttcggctgtt | Refsland et al (2010) |
| CNOT4 | gtccaaaacctgactgcatgtatc | ggtgtttacccgcctgcat | Rácz et al (2021) |
| PUM1 | tgcgggagattgctggacat | gtgtggcacgctccagtttc | Rácz et al (2021) |

using Genomedata software (Hoffman et al, 2010). This dataset was trained for 12 labels with 100-bp resolution, confined to the nonblacklisted portion of the core chromosomes (1–22, X, and Y) of the GRCh38 reference genome. Then, genomic segments were identified with Segway annotate (Hoffman et al, 2012; Chan et al, 2017), and the signal distribution was calculated using Segtools (Buske et al, 2011), and the heatmaps (in Figs 1 and S1A) were plotted by Seaborn (Matplotlib module in Python [Hunter, 2007], cf. Supplemental Data 1). BEDTools Jaccard indices (Quinlan & Hall, 2010) were calculated between the segments derived from merged data and individual replicates (cf. Fig S1B; the script is provided in Supplemental Data 1). The results were organized according to similarity, and the plot was created using Python's Seaborn module. Length distribution (cf. Fig S1C) and enrichment of genomic segments in gene parts measured as compared to the randomized occurrence of segments in the given gene parts (cf. Fig S2) were calculated using Segtools (Buske et al, 2011).

**Calculation of average replication timing score, AT content, and the proportion of genomic features overlapping with the genomic segments**

Average replication timing scores for the segments were calculated from two replicates of E/L Repli-seq data available for the HCT116 cell line in the Replication Domain database (Int90617792 and Int97243322 [Weddington et al, 2008]). In E/L Repli-seq, the higher score corresponds to earlier replication (Marchal et al, 2018). The 12 genomic segments were separated into distinct BED files (defining genomic intervals), and their intersections with replication timing files were taken (using BEDTools intersect, version 2.26.0 [Quinlan & Hall, 2010]). Average RT scores for each interval were calculated using bigWigAverageOverBed (Kentutils, https://github.com/ucscGenomeBrowser/kent-core). The mean values and the SD were calculated for intervals belonging to the same segments. The AT content of the segments was calculated using BEDTools nuc (Quinlan & Hall, 2010) and the GRCh38 reference genome. Genomic features, including regulatory CpG islands and the promoter flanking regions, were obtained from the UCSC Table Browser (Karolchik et al, 2004; Kuhn et al, 2013), and overlaps were calculated using the BEDTools annotate (Quinlan & Hall, 2010). Details of the calculations

are provided in Supplemental Data 1. Graphs in Figs 1 and S1A were prepared in Origin 8.6 (OriginLab Corporation).

**Calculation of gene-specific U-scores**

The U-scores were calculated based on the enrichment data expressed in the ratio of coverage in the enriched versus in the input samples (calculated by the deepTools package version 3.2.1. [Ramírez et al, 2016], https://github.com/deeptools/deepTools/releases). These uracil enrichment tracks were cleaned, excluding the blacklisted (cf. Fig 2—figure supplement 2 in Pálinkás et al [2020]) and the hard-masked regions. The cleaned ratio.BDG files were converted to probability tracks (p-tracks) based on the values of *total uracil content* published in Pálinkás et al (2020) and listed also in Table S1. Calculation of probabilities of uracil occurrence in the unit of uracil/million bases for each bin in the BDG files (100 bp) was done using standard Unix tools (awk). The BDG files were converted to bigwig using wigToBigWig of the Kentutils package (https://github.com/ucscGenomeBrowser/kent) (Kuhn et al, 2013). These p-tracks represent a quasi-absolute scale of probabilities, with the assumptions that the measured overall uracil content of the samples is correct and that in the blacklisted regions, the mean frequency of the uracil occurrence is similar to that in the rest of the genome. Based on these p-tracks, an average of uracil enrichment (U-score) was calculated for each gene listed in GENCODE v34 annotation (downloaded from UCSC Table Browser, https://genome.ucsc.edu/cgi-bin/hgTables, Harrow et al, 2006, 2012). Intronic segments and a 1,000-nt upstream region, considered as the core promoter, were also included, whereas blacklisted regions were excluded. Average uracil enrichments were calculated for these gene-related intervals based on the p-tracks using bigWigAverageOverBed (https://github.com/ucscGenomeBrowser/kent [Kuhn et al, 2013]). Then U-scores were filtered for a nonredundant and not blacklisted list consisting of the longest isoform of 44,126 genes (each characterized with a unique Ensembl gene ID). The p-tracks and U-score data are accessible through GEO Series accession number GSE285931. Histograms for U-score distribution (cf. Fig 2A and Source Data File) were prepared in Origin 8.6 (OriginLab Corporation). The enrichment data for different gene types, as defined by the GENCODE v34 annotation, within the top 2,000 most uracilated genes (cf.

Fig 2B) were calculated using standard Unix tools (awk), and the heatmap was generated in Excel (Microsoft). All applied scripts are provided in Supplemental Data 1.

### Detection of the differentially uracilated genes

For efficient comparison, relative U-scores were calculated for individual replicates of each drug treatment separately. The datasets were merged based on common gene symbols using the "merge" function of R (https://www.R-project.org [R Core Team, 2018]). The fold change values were calculated by the ratio of the means of the U-scores in the two samples:

$$\frac{\text{Uscore}_{5FdUR\_rep1} + \text{Uscore}_{5FdUR\_rep2}}{\text{Uscore}_{RTX\_rep1} + \text{Uscore}_{RTX\_rep2}} \qquad (1)$$

Statistical significance was assessed using Welch's $t$ test. We considered relative U-scores of a gene in the two compared samples to be significantly different (cf. red data points in Fig 2C) when the $P$-value was smaller than 0.05 (i.e., $-\log_{10}(P\text{-value}) > 1.301$) and the fold change was at least 1.5 (i.e., $\log_2(\text{fold change}) > 0.58496$). The data were visualized by the ggplot2 package (https://ggplot2.tidyverse.org [Villanueva & Chen, 2019]). All applied scripts are provided in Supplemental Data 1. The data are available as Source Data File.

### Functional analysis of differentially uracilated protein-coding genes

A complex functional enrichment analysis was performed in the STRING database (version 12.0, https://string-db.org [Szklarczyk et al, 2019]). For the groups of significantly differentially uracilated and the top 200 most uracilated genes, we applied the search option "Multiple proteins." For the hierarchical list of protein-coding genes ranked by U-scores or a combined score reflecting the differential uracilation, the search option "Proteins with Values/Ranks" was used. The combined scores were calculated as follows (cf. also the Source Data File).

$$\text{combined score} = \log_2(\text{fold change}) * -\log_{10}(p-\text{value}) \qquad (2)$$

In the case of "multiple proteins" outputs, interactions were considered according to the full STRING network, from sources "Textmining," "Experiments," or "Databases" at least with medium confidence (0.4), and enrichment was measured against a relevant set of genes for which U-scores were calculated and were not zero. The network statistics were measured against the whole human proteome (cf. Table 2). In the case of hierarchical U-score lists, genes with a U-score of zero were excluded. The raw results are available at permanent links (cf. Table 1). The enriched terms were further filtered (FDR was not higher than 0.00005, and relevant categories were selected as follows: "COMPARTMENTS," "GO Component," "GO Function," "InterPro," "KEGG," "Pfam," "Reactome," "SMART," "WikiPathways"), and the results are summarized in Source Data File, and visualized on bar graph created in Origin 8.6 (OriginLab Corporation; cf Source Data File). For the top 200 most uracilated protein-coding genes, STRING networks were exported to Cytoscape (Shannon et al, 2003; Excoffier et al, 2017),

where enriched functional terms were also implemented (cf. Source Data File) and visualized (cf. Fig S3).

### Variant calling and analysis of single-base substitution (SBS) frequencies

Raw sequencing data (cf. Table 4) were mapped to the GRCh38 human reference genome (GRCh38.d1.vd1.fa, Genomic Data Common, https://gdc.cancer.gov/about-data/data-harmonization-and-generation/gdc-reference-files) using bwa as described in Pálinkás et al (2020). Variants were called from cleaned aligned reads using Mutect2, the somatic variant caller of the GATK4 (Genome Analysis ToolKit 4) package (https://gatk.broadinstitute.org/hc/en-us/articles/360051306691-Mutect2 [McKenna et al, 2010; Van der Auwera et al, 2013]). Variant calling experiments were performed in seven different arrangements from either merged or individual replicate data for each sample (cf. Source Data File). On the one hand, UNG-inhibited HCT116 cells, nontreated (NT) or treated with RTX or 5FdUR, were compared with the WT HCT116 cells (cf. Figs 4 and S4), and on the other hand, the same UNG-inhibited HCT116 cells, as well as their MMR-proficient version treated with either RTX or 5FdUR, were compared with their corresponding nontreated samples (cf. Figs S5 and S6). Raw VCF files were filtered using FilterMutectCalls of the GATK4 package, and the variants that were labeled by at least one of the following terms ("weak_evidence," "normal_artifact," "slippage," "germline," "base_qual") were excluded from further analysis. All of the filtered variant data are accessible through the GEO Series accession number GSE285931.

The new whole-genome sequencing (WGS) data of dose-dependent treatments (cf. Tables 3 and 4) were processed as described above, and variants were called against the corresponding NT sample (filtered variant data are found under GEO Series accession number GSE285767).

The six types of single-base substitutions (SBS: $C{\rightarrow}T$, $C{\rightarrow}G$, $C{\rightarrow}A$, $T{\rightarrow}A$, $T{\rightarrow}C$, $T{\rightarrow}G$) were extracted from the filtered VCF files. The occurrence (sum of the counts weighted with their allele frequencies) of the six types of SBS, and their frequencies (occurrence of one SBS/occurrence of all types of SBSs) were calculated using Python's Pandas module. For statistical evaluation, the seven variation calling experiments for each comparison (cf. Source data file) were tested by Welch's two-sample $t$ tests. The results were visualized on bar graphs created in Origin 8.6 (OriginLab Corporation) (cf. Figs 4A and S5A, and for the clustered variants found in the new WGS data, Fig 7A).

To address the correlation with replication timing, the genome was split into three subsets according to the replication timing scores derived from E/L Repli-seq data (early: >2.5; middle: between 2.5 and −2.5; and late: <−2.5) for HCT116 in the Replication Domain database (Int90617792 and Int97243322 [Weddington et al, 2008]). Variants from the merged data were split accordingly, and the SBS frequencies were determined for the three subsets separately using Python's Bioframe module (cf. Fig 5A). Frequencies of the C-to-T transitions were also calculated for the genomic segments (cf. Fig 5B). Correlation between segments and genomic features (regulatory CpG islands and promoter flanking regions) was characterized using BEDTools annotate (version 2.26.0) (Quinlan & Hall, 2010) as described in Supplementary file 4 in

Pálinkás et al [2020] (cf. Fig 5B). All applied scripts are provided in Supplemental Data 1.

### Calculation and analysis of mutational signatures

To investigate the extended mutational environment of the SBSs, we applied SigProfiler Matrix Generator 1.1.21 and SigProfiler Plotting Python modules (Bergstrom et al, 2019) to produce "96" type matrices for the triplet SBS spectra (cf. Figs 4B, 7B, and S5B), the "1,536" type matrices for the pentanucleotide SBS spectra (cf. Fig S11), and the "78" type matrices (where the reverse complement transitions are also collapsed, so-called strand-agnostic double-base substitutions) for the DBS spectra (cf. Figs S4, S6, and S10). The applied scripts are provided in Supplemental Data 1.

### Reconstitution of measured triplet spectra for C-to-T transitions

Published spectra for A3A, A3B, and A3C (Fig 3 in Brown [2024]) were extracted using WebPlotDigitizer. These spectra, combined with a background spectrum (normalized data from low-dose 5FdUR-treated sample, cf. Fig 7B), were used to reconstitute the triplet spectra for high-dose (20 $\mu$M) 5FdUR and RTX-treated samples using the SigProfilerExtractor tool (Islam et al, 2022). The cosine similarity values between measured and reconstituted spectra are 0.993 and 0.998 for 5FdUR and RTX samples, respectively. The bar graphs comparing the reconstituted and the measured spectra, as well as the pie charts, were plotted using Origin 8.6 (OriginLab Corporation).

### Analysis of the SBSs' wider sequence context for stem-loop formation potential

The ±15-nt sequence contexts of detected SBSs, clustered and not clustered separately, were derived using the human reference genome sequence (GRCh38) and BEDTools nuc (version 2.26.0) (Quinlan & Hall, 2010), and analyzed for stem-loop formation potential using a custom Python script (provided in Supplemental Data 1). Predicted structures that consist of at least a 4-nt stem and a 3- to 6-nt loop were tested for the position of the SBS. The proportion of C-to-T transitions relative to all types of SBSs was calculated for all contexts, clustered events, and loop-context C-to-T transitions, respectively. Values were plotted for each variant calling (cf. Source Data File) for the untreated, the 0.1 $\mu$M RTX-, and the 20 $\mu$M 5FdUR-treated cases using Origin, version 8.6 (OriginLab Corporation; cf. Fig S11C and the corresponding Source Data File).

### Analysis of intermutational distances

The intermutational distances (IDMs) between neighboring C-to-T transitions (cf. Fig 4C) were calculated using standard Unix tools (awk), and the log histograms were made using ggplot2 (https://ggplot2.tidyverse.org [Villanueva & Chen, 2019]) of the R package (R Core Team, 2018). The applied scripts are provided in Supplemental Data 1.

### Cell culturing and treatments

HCT116 cells (European Collection of Cell Cultures [ECACC]) and HCT116+ch3 subline (carrying the WT gene for hMLH1, a kind gift from C. Richard Boland [Baylor University, Dallas, Texas, US]) were previously used to establish UNG-inhibited stable cell lines (Pálinkás et al, 2020) that were maintained as described in Pálinkás et al (2020). Cell lines used in this study were tested for *Mycoplasma* contamination. For treatments, fresh media complemented with the given dose of either 5FdUR or RTX were added ~20 h after passaging (when the cells attached to the surface), and the cell cultures were incubated for the indicated time without any adjustment.

### Cell cycle analysis

2D cell cycle analysis was performed using 5-bromo-2'-deoxyuridine (BrdU) labeling and propidium iodide (PI) staining. Non-treated and drug-treated (100 nM or 20 $\mu$M for 24 h) HCT116 UGI cells were labeled with 20 $\mu$M BrdU for 1.5 h. After trypsinization, cells were collected in media supplemented with 3 mM EDTA and washed with PBS. Cells were fixed by adding 70% ethanol (taken directly from −20°C and kept on ice) drop by drop while vortexing continuously. The cells were kept in the fixative for 1 h at 4°C or overnight at 4°C. Then, for denaturing the DNA, fixed cells were incubated in 2 M HCl supplemented with 0.5% Triton X for 15 min at RT. Cells were centrifuged at 3,000$g$ for 5 min, then resuspended in 0.1 M sodium borate (pH 8.5), and incubated for 10 min at RT for neutralization. Cells were washed with PBS and incubated in blocking buffer (PBS containing 1% BSA and 0.05% Tween-20) for 10 min. Samples were incubated with Pacific Blue-conjugated anti-BrdU antibody (Invitrogen, Thermo Fisher Scientific) at a 1:8 dilution in blocking buffer, incubated for 40 min at RT in the dark. After diluting the samples with blocking buffer, they were washed with PBS. DNA was stained by adding 10 $\mu$g/ml PI and 20 $\mu$g/ml RNase A in PBS, incubated for ~30 min at RT. Cell cycle analysis was performed using an Attune N x T Flow Cytometer (Thermo Fisher Scientific). The results of at least three biological replicates were evaluated using a custom R script provided in Supplemental Data 1. The box plot (cf. Fig 6A) was created using ggplot2 (https://ggplot2.tidyverse.org) (Villanueva & Chen, 2019) of the R package (R Core Team, 2018).

### Cell viability assay

Cell Counting Kit 8 (CCK8, CK04-11; Dojindo) was used to estimate cell viability during the drug treatments. Cells were counted using Bürker's chamber, and the starting cell number was set to 2,000 cells/well in 96-well transparent plates (Costar). About 22 h after the passaging, the media were changed to new media (NT) or new media complemented with drugs (10, 100 nM, 10, or 20 $\mu$M). Two technical replicates were used for each concentration. The CCK8 assay was performed as suggested by the manufacturer after 0-, 16-, 24-, 40-, 48-, or 72-h treatment. Cell-free wells were measured as negative controls for each case. Absorbance was detected (at 450 nm) after 60-min incubation with the reagent at 37°C using a plate reader (SynergyMx, BioTek). Two reads were done shortly after each other. At least two biological replicates

were measured for each time point. Significance was assessed using Welch's $t$ test with a $P$-value of 0.05.

## Genomic DNA isolation and WGS

Genomic DNA of HCT116 cells was purified using Quick-DNA Miniprep Plus Kit (Zymo Research) according to the supplier's protocol, eluted in nuclease-free water. Then, genomic DNA was fragmented to the range of 300–500 base pairs using an ultrasound device (Bioruptor, Diagenode). Next-generation sequencing was done by iBioScience, and libraries were prepared using NEBNext Ultra II DNA Library Prep Kit for Illumina (NEB) according to the manufacturer's instructions. The quality of the library was checked on a 4,200 TapeStation System using D1000 ScreenTape (Agilent Technologies), and the quantity was measured on Qubit 3.0 (Thermo Fisher Scientific). Illumina sequencing was performed on a NovaSeq 6000 instrument (Illumina) with a 2 × 151 run configuration. Data processing and mutational analysis were done as for the U-DNA-seq input data.

## RNA isolation

Cells were cultured in six-well plates and treated with 100 nM or 20 $\mu$M of RTX or 5FdUR. After 48-h treatment, the media were removed, and the cells were suspended in TRIzol reagent (0.35 ml per well, Cat. no. 15596018; Thermo Fisher Scientific), and incubated for 5 min at room temperature. Chloroform (0.14 ml per sample) was added to the samples, shaken vigorously, and centrifuged at 12,000$g$ at 4°C for 15 min. RNA-containing aqueous phase was collected. RNA was precipitated with ice-cold isopropanol (0.35 ml, incubation for 10 min at 4°C), washed twice with 75% ethanol, dried at 55°C, and then dissolved in RNase-free water. The quality of RNA samples was checked on 1% agarose gel (cf. Fig S12A). RNA concentration was measured with NanoDrop 2000C (Thermo Fisher Scientific), and the A260/A280 ratio was recorded.

## qRT-PCR

Reverse transcription was performed using High-Capacity cDNA Reverse Transcription Kit (Cat. no. 4368814; Applied Biosystems) with 400 ng of total RNA in a 20 $\mu$l reaction, following the manufacturer's protocol. No reverse transcription (NRT) controls were prepared by omitting the reverse transcriptase enzyme for each RNA sample and isolation. Reverse transcription efficiency was evaluated for each target using ½ serial dilutions starting with 800 ng RNA (cf. Fig S12B).

qRT-PCRs were performed in a 10 $\mu$l final volume using MyTaq-HS MIX (BIO-25046), EvaGreen dye (31000; Biotium), nuclease-free water, cDNA template, and appropriate primers. Primers given in Table 5 (Refsland et al, 2010; Rácz et al, 2021) were obtained from Sigma-Aldrich with desalting purification in a dry format and dissolved in nuclease-free water according to the recommendation to make 100 $\mu$M solutions. The concentration of the primer solutions was checked with NanoDrop 2000C to adjust the final concentration in the PCR to 500 nM. In each PCR, 0.157 $\mu$l of the cDNA sample was used. For every sample and every target gene, three technical replicates were used. Controls were measured for

each target: NRT for each isolation and nontemplate control (NTC) on each plate. The differences between the Cq values of the controls and the samples were higher than 4 in all cases. Clear Hard-Shell 96-Well PCR Plates (Bio-Rad) and Microseal "B" PCR Plate Sealing Film (Bio-Rad) were used. Thermal cycling and detection were performed in a CFX96 real-time PCR detection system (Bio-Rad). Thermal cycling conditions were set as follows: 95°C for 5 min followed by 50 cycles of 95°C for 10 s; 58°C for 15 s; and 72°C for 30 s. After amplification, melting curve analysis was performed from 60°C to 95°C with an increment of 0.5°C every 5 s (Fig S12C). PCR products were also checked on 1% agarose gel.

To determine the PCR efficiency, cDNA derived from the nontreated HCT116 cell line was amplified in PCRs for each target gene and purified with the NucleoSpin Gel and PCR Clean-Up kit (740609; Macherey-Nagel). The concentration of the purified PCR products was measured using a NanoDrop 2000C. A seven-point fourfold dilution series in the range of concentration from 4 to 0.0009765 fg/$\mu$l was measured in qPCRs in triplicate. The Cq values were plotted against the $\log_{10}$ of the concentration values, and linear regression was applied (the regression coefficient was used as a quality control, and was accepted above 0.99). The PCR efficiency values were calculated using formula E (%) = $[10^{(1/-slope)} - 1] \times 100\%$ (Fig S12D) and were used for further calculations.

To determine the relative normalized expression of APOBEC3s, two reference genes (CNOT4 and PUM1) were chosen to normalize our data. Interplate calibrator samples were used to compare samples on different plates. The threshold value was set to 250 relative fluorescence units for every plate measured. At least three biological replicates for each target were measured, and analysis was performed using the Bio-Rad CFX Maestro program. Statistical significance thresholds were set at $P = 0.05$ and $P = 0.005$.

## 3D-PCR

Genomic DNA was isolated 48 h after drug treatment (100 nM RTX or 20 $\mu$M 5FdUR) as described for WGS. A specific 570-bp fragment of the MDM2 gene was amplified with Taq polymerase (MyTaq-HS Red Mix, BIO-25047; Bioline) using genomic DNA as a template (2 ng/$\mu$l final concentration) and degenerate primers (forward: 5'-GCAAAGTTGCTAGCATTYYTGTGAC-3'; reverse: 5'-GCA-CATGTAAAGCARGCYATAARATGT-3'); the reaction was performed as recommended by the manufacturer, at an annealing temperature of 48°C for 25 cycles. A second PCR was performed using 2 $\mu$l of the first PCR product as a template, the same primers, and Q5 polymerase (Q5 High-Fidelity DNA Polymerase, NEB, M0491S) in a final reaction volume of 100 $\mu$l, according to the supplier's protocol. The 100 $\mu$l of reaction mixture was divided into eight tubes, each corresponding to a distinct denaturation temperature ($T_d$). $T_d$ gradient was as follows: 86°C, 85.8°C, 85.4°C, 84.8°C, 84.1°C, 83.5°C, 83.1°C, 83°C. Thermal cycling conditions were as follows: initial denaturation at 98°C for 30 s; 25 cycles of denaturation at a $T_d$ gradient (86–83°C) for 10 s, annealing at 68.5°C for 30 s, and extension at 72°C for 30 s; followed by a final extension at 72°C for 2 min. PCR products were separated on 1% agarose gels, stained with GelRed, and visualized using a ChemiDoc XR + system (Bio-

Rad). Band intensities were further analyzed with ImageLab software (Bio-Rad).

## Preparation of cell extracts

For the preparation of whole-cell extracts, PBS-washed cells were lysed by incubating at 4°C for 15 min in 5X diluted lysis buffer. The suspension was supplemented with an equal amount of normal lysis buffer (50 mM Tris–HCl, 150 mM NaCl, 10% glycerol, 0.5 mM DTT, 2 mM PMSF, 0.1% Tween-20, 0.1% Triton X, protease inhibitor tablet, pH 7.5) and sonicated at 4°C in a Bioruptor (Diagenode). After centrifugation, the total protein concentration of the lysate was determined using the Bradford assay.

Nuclear extracts were prepared from drug-treated and control HCT116 cells using differential detergent lysis and high-salt extraction. Briefly, cells were lysed in a hypoosmotic buffer (0.3 M sucrose, 2% Tween-40, 10 mM Hepes, 10 mM KCl, 1.5 mM MgCl$_2$, 0.1 mM EDTA, 0.5 mM DTT, protease inhibitor tablet, pH 7.9), and nuclei were purified by sucrose gradient centrifugation. Nuclear proteins were then extracted in a high-salt buffer (20 mM Hepes, 420 mM NaCl, 1.5 mM MgCl$_2$, 0.2 mM EDTA, 0.5 mM DTT, protease inhibitor tablet, 25% glycerol, pH 7.9) and quantified using the Bradford assay.

## Cytidine deaminase activity measurement in cell-free extracts

Whole-cell or nuclear extracts were prepared from cells treated for 24 and 48 h with 0.1 $\mu$M RTX or 20 $\mu$M 5FdUR. Each deaminase activity assay contained 50 $\mu$g total protein (whole-cell or nuclear extract), and 10 pmol Cy3-labeled 30-nt oligonucleotide substrate in 20 $\mu$l reaction. The substrates contain a single C or U at their 13th position and are fluorescently labeled at their 5′ end (Cy3—5′-AGGAATTGGTAT$\underline{X}$ATGATTAGAGTGGAGTAA-3′, where X is C or U). The U-substrate was used as a technical positive control. After the deaminase reaction (37°C, 10 h), samples were heat-denatured (95°C, 20 min) and centrifuged. The supernatant was transferred to fresh tubes and processed sequentially with UDG (37°C, 1 h) and 0.1 M NaOH (95°C, 10 min) to cleave the deaminated product. Reaction products were separated by 20% denaturing TBE-PAGE and detected using a ChemiDoc XR+ imaging system (Bio-Rad).

## Western blot

Sample preparation for Western blotting was performed after treatment with 0.1 or 20 $\mu$M RTX, 5FdUR for 48 h, or from untreated controls. Cells were washed twice with PBS, resuspended in cytoplasmic extraction buffer (1 ml/10$^7$ cells; 20 mM Tris, pH 7.4, 10 mM NaCl, 3 mM MgCl$_2$, 0.5 mM DTT, 0.05% NP-40, protease inhibitor tablet [Roche]), and incubated on ice for 15 min. Cytoplasmic fraction was obtained by centrifugation (3,000$g$, 7 min, 4°C); the pellet was resuspended in nuclear extraction buffer (200 $\mu$l/10$^7$ cells; 50 mM Tris, pH 7.4, 150 mM NaCl, 50 mM NaF, 5 mM EDTA, 1 mM EGTA, 1 mM PMSF, 1% NP-40, protease inhibitor tablet [Roche]) and incubated on ice for 30 min with frequent vortexing. Nuclear extract was collected after centrifugation (16,000$g$, 10 min, 4°C), and the pellet was resuspended in PBS and solubilized in SDS loading buffer as chromatin-bound fraction.

Proteins were separated on 12% SDS–PAGE and transferred to a nitrocellulose membrane (10600125, Cytiva; Amersham Protran). Membranes were blocked with 5% milk powder in TBS-T and incubated overnight at 4°C with primary antibodies diluted in 5% BSA/TBS-T (anti-A3C, 1:875 [10591-1-AP; Proteintech]; anti-A3D, 1:580 [antibodies-online, ABIN7238535]; anti-actin, 1:2,500 [A1978; Sigma-Aldrich]). HRP-conjugated secondary antibodies (1:10,000) were used to develop the blots, and Immobilon Western Chemiluminescent HRP Substrate (Millipore) was used to detect the chemiluminescent signal in ChemiDoc MP Imaging System (Bio-Rad). Western blot images were analyzed and processed using ImageLab software (Bio-Rad).

## Immunocytochemistry for APOBEC3B

Cells (~20,000 cells per well) were seeded on coverslips. After attachment to the surface, the medium was replaced with fresh medium supplemented with drugs (RTX or 5FdUR). Cells were cultured for 24 or 48 h, then fixed using 4% PFA solution and permeabilized with 0.1% Triton X-100 in PBS. Blocking was performed overnight at 4°C using blocking buffer (1% BSA, 5% FBS in PBS). Primary staining was carried out with an anti-A3B-antibody (ab222330; Abcam, rabbit polyclonal, 1:200 in blocking buffer), followed by secondary staining with anti-rabbit Alexa Fluor 633 (1:1,000 in blocking buffer). DNA was stained with DAPI (1:10,000 in PBS). Coverslips were mounted in FluorSave mounting medium. Confocal images were acquired using a Zeiss LSM 710 laser-scanning confocal microscope, and image analysis was performed using ImageJ (Fiji) software (National Institutes of Health, USA [Schneider et al, 2012]).

## Kaplan-Meier plots

TCGA database was accessed using the online tool TCGExplorer (Kus et al, 2023 *Preprint*). Colon adenocarcinoma data were filtered to include only those genes present in at least 25% of the experimental datasets. Kaplan-Meier plots were generated for tumor samples categorized within the top and bottom 20% of APOBEC3 expression. Cumulative survival data were calculated, and survival propensity values were plotted over 3,000 d.

## Schematic preparation

Figs 8C and S14A, and the Graphical Abstract were created in BioRender, which uses an integrated AI-assisted design model.

# Data Availability

The data discussed in this publication have been deposited in NCBI's Gene Expression Omnibus (Edgar et al, 2002) and are accessible through GEO Series accession numbers GSE285931 (that includes further analysis of previously deposited data under GSE126822) and GSE285767.

# Supplementary Information

# Acknowledgements

We gratefully acknowledge Gábor Tusnády for providing access to computational capacity, to György Váradi and Edit Szabó for the technical help with flow cytometry, to Nikolett Nagy and Gergely Attila Rácz for helpful advice on the qRT-PCR experiments, and to Judit Tóth for advice on the article writing. We also used Grammarly for polishing the English language of the article. Project no. 137867 has been implemented with the support provided by the Ministry of Innovation and Technology of Hungary from the National Research, Development and Innovation Fund, financed under the OTKA_FK_21 funding scheme for A Békési. A Békési was also supported by the János Bolyai Research Scholarship of the Hungarian Academy of Sciences (BO/726/22/8), and by the ÚNKP-23-5-BME-467 New National Excellence Program of the Ministry for Culture and Innovation from the source of the National Research, Development and Innovation Fund. For E Holub: The scientific work and results publicized in this article were reached with the sponsorship of Gedeon Richter Talentum Foundation in the framework of the Gedeon Richter Excellence PhD Scholarship of Gedeon Richter Plc. E Holub was also supported by the ÚNKP-21-2-I-BME-277 New National Excellence Program of the Ministry for Culture and Innovation from the source of the National Research, Development and Innovation Fund. MB Szajkó was supported by the DKÓP-25-1-BME-52 Doctoral Students' Excellence Grant Program of the Ministry for Culture and Innovation from the source of the National Research, Development and Innovation Fund. BG Vértessy was supported by the National Research, Development and Innovation Fund of Hungary (K135231, K146890, NKP-2018-1.2.1-NKP-2018-00005, 2022-1.2.2-TÉT-IPARI-UZ-2022-00003, TKP2021-EGA-02 grant), and the ICGEB Research Grants Programme 2023 (CRP/HUN23-02).

## Author Contributions

E Holub: formal analysis, funding acquisition, validation, investigation, visualization, methodology, and writing—original draft, review, and editing.

G Papp: software, formal analysis, validation, investigation, visualization, methodology, and writing—original draft.

HL Pálinkás: investigation, visualization, methodology, and writing—review and editing.

MB Szajkó: funding acquisition, validation, investigation, visualization, methodology, and writing—original draft, review, and editing.

R Izrael: methodology and writing—review and editing.

G Róna: methodology and writing—review and editing.

BG Vértessy: conceptualization, resources, supervision, funding acquisition, and writing—review and editing.

A Békési: conceptualization, data curation, formal analysis, supervision, funding acquisition, validation, investigation, visualization, methodology, and writing—original draft, review, and editing.

## Conflict of Interest Statement

The authors declare that they have no conflict of interest.

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
