## [Reviewer comments · Life Science Alliance]

Effects of thymidylate synthase inhibitors differ in genomic uracilation and mutagenic potential

Eszter Holub, Gábor Papp, Hajnalka Pálincás, Milda Szajkó, Richard Izrael, Gergely Rona, Beata Vertessy, and Angela Békési
DOI: <https://doi.org/10.26508/lsa.202503352>

Corresponding author(s): Angela Békési, Budapest University of Technology and Economics and Beata Vertessy, HUN-REN Research Centre for Natural Sciences

Review Timeline:

Submission Date:	2025-04-09
Editorial Decision:	2025-06-05
Revision Received:	2025-10-08
Editorial Decision:	2025-12-18
Revision Received:	2026-01-13
Accepted:	2026-01-17

Scientific Editor: Sarita Hebbar

Transaction Report:

June 4, 2025

Re: Life Science Alliance manuscript #LSA-2025-03352

Dr. Angela Békési
Budapest University of Technology and Economics
Faculty of Chemical Technology and Biotechnology
Műegyetem rakpart 3.
Budapest H-1111
Hungary

Dear Dr. Angela Békési

Thank you for submitting your manuscript entitled " The effects of thymidylate synthase inhibitors differ in genomic uracilation and mutagenic potential" to Life Science Alliance (LSA). The manuscript was assessed by three expert reviewers, whose comments are appended to this letter.

All three reviewers commented on the value of this work to the community. However we agree with the reviewers that the manuscript needs additional data and clarification for publication at LSA. A revised manuscript must include the following:

1. Drug-treatments:

Information on doses used, statistical analyses on dose-response curves, and cell viability assays (Reviewer 1, points 1,3 and Reviewer 3, major revisions point 1)

2. Differences in genomic uracilation:

A more succinct contextualisation of gene networks (Reviewer 1, point 2) and a statistical refinement in U-score and gene analyses (Reviewer 3, major revisions point 3)

3. Validation of APOBEC3 involvement:

Increased APOBEC3 mRNA expression should be supported with protein expression and activity (Reviewer 2, point 5 and Reviewer 3, major revisions point 2)

Functional linking of APOBEC to mutational signature by method of choice: overexpression or knockdown (Reviewer 3, major revisions point 2) OR hairpin loop analyses (Reviewer 1, point 4)

4. Edits to eliminate grammatical errors, and improve clarity and presentation as indicated by all the reviewers (Reviewer 1, point 2, Reviewer 2, point 1, and Reviewer 3, major revisions point 4)

In line with the overall recommendations, we invite you to submit a revised manuscript addressing the reviewers' comments. While a rebuttal must respond to all points in some form, additional data to resolve these points (other than ones indicated above) is not required.

Thank you for this interesting contribution to Life Science Alliance. We are looking forward to receiving your revised manuscript.

Sincerely,

Sarita Hebbar, PhD
Scientific Editor
Life Science Alliance
<http://www.lsjournal.org>

B. MANUSCRIPT ORGANIZATION AND FORMATTING:

Reviewer #1 (Comments to the Authors (Required)):

The authors examined the uracilation profiles of HCT116 cell lines to compare the effects of RTX and 5FdUR. This validated known associations, like early replication timing, but identified several genome segments where there were drug specific effects in gene coding regions. They next examined gene specific uracilation and again found differences, as well as a strong effect of MMR status on the effects. They carried out a functional enrichment analysis using STRING and found genes involved in specific processes were differentially affected by the drugs and MMR status. They next looked at the types of mutational differences and found more C:G to T:A transitions with 5FdUR and signature analysis implicated APOBEC3 activity. These, as well as a DBS signature were dependent on MMR deficiency. They examined the effects of the drugs on cell proliferation and found both dose and drug dependent effects on the cell cycle and viability, with a reduced sensitivity at high doses. Analysis of A3 expression and mutational signatures suggested a contribution from A3B and A3C, but not A3A. A3B was the most dramatically upregulated following treatment with either drug, although the clinical correlations were highest with A3C. Finally, they performed IF and directly confirmed A3B induction with either drug.

Building on previous work, the paper identifies an interesting uracilation pattern difference between the 2 clinically relevant drugs that is dependent on MMR status and appears to involve A3B, or potentially other family members. The paper generally follows a logical arc but some of the data analysis is less informative and seems extraneous for making the point (see specific comments below). I think the paper would benefit greatly from a graphical summary (aside from the abstract) that gives a more unifying idea of the proposed mechanistic underpinning of the difference in the context of existing literature, I remain somewhat confused as to why the authors think that 5FdUR yields an MMR deficiency dependent A3 signature when both agents upregulate the protein pretty extensively- what would account for this interesting observation? I think that while the analysis was extensive, this observation is also coming from the background of a single cell line that carries a number of DNA repair related mutations, potentially yielding some cell type specific observations that should be considered.

Specific comments

1. What is the drug dose used in the initial analysis in Figure 1? No where in the text or Figure legend can I locate this and another paper is referenced. As it is relevant to comparing this data with data later in the paper that uses different doses, it needs to be clarified and in the Figure legend of this paper.
2. Given the amount of attention paid to a deep analysis of the gene functional networks in Table 1, 2, Figure 3 and multiple supplemental Figures, a clearer point needs to be made here. To me the 2 most obvious likelihoods are that these are genes that are particularly active in the particular cell line, and that chromatin is thus exposed and accessible to mutagenesis or, less likely, these are selected for function to adapt to the treatments. Is there gene expression or ChIP data that could be applied to this to see if this is related to relative expression in the different cell lines? As it is, it is an enormous amount of data and text that is of unclear relevance to the mechanism and I find that it disrupts the story more than making anything clearer. It makes a minor observational point that there are cell line differences, but beyond that it is unclear to me the purpose of including all of this in the

middle of the story.

3. In Figure 6B, data is provided that there are differences in toxicity at different doses. Are any of these differences statistically significant? Has this been validated using colony forming assays or another approach? I worry somewhat that these agents would have significant mitochondrial effects, particularly at high doses that would impact these types of assays that are measuring metabolites.
4. In Figure 8C, the survival of patients correlated to the expression of individual APOBECs is plotted. This is interesting in itself, but trying to connect this to the data from a single cell line is a pretty massive leap. Is there any evidence in RTX or 5FdUR treated patients that any APOBECs are more highly expressed? This would provide more compelling evidence for a clinical connection than survival that may have nothing whatsoever to do with the mutagenesis observations from cell lines.
5. How do the authors reconcile the observation that RTX at low dose and 5FdUR at high dose (Figure 8B) cause similar levels of A3B upregulation but the mutagenic outcome is distinct related to the A3B signature?
6. The dependency of MMR deficiency on the mutational signature is seemingly at odds with observations from previous studies that found MMR was required for A3 clustered signatures (<https://doi.org/10.1016/j.cell.2017.07.003>). Has this been considered and do the authors have any thoughts on the mechanistic implications?

Reviewer #2 (Comments to the Authors (Required)):

This manuscript extends the results of a previous study (Palinkas et al eLife 2020) in which these authors mapped uracils in the genomes of UNG inhibited MMR-defective cells. The principle conclusion here is that the accumulation and distribution of uracils and mutations following treatment of cells with 5FdUR or raltitrexed (RTX) are substantially different from each other. This is not surprising in itself as the two compounds inhibit dTMP synthesis in very different ways. The more interesting observation is that the two drugs cause a modest increase in APOBEC3s- but not A3A (Fig. 8)!! Somewhat paradoxically 5FdUR treatment, but not RTX treatment, creates a mutational pattern associated with the APOBEC enzymes, the COSMIC signature SBS2. These observations increase our understanding of how two frequently used anticancer compounds, 5FdUR and RTX, affect human cells. Unfortunately, the manuscript contains a lot of low value data and speculation weakening the manuscript. Eliminating these sections would strengthen and focus the manuscript.

1. The significance of the first three sections in the results (Pages 5 through 10) is unclear. As dUTP and 5FdUTP are expected to be incorporated during replication, the extensive analysis of which groups of genes accumulate uracils does not make much sense. Most of these sections involve complex statistical and computational analysis that is difficult to understand. I tried to understand genome segmentation analysis by reading this manuscript, the Palinkas et al manuscript and the original NAR paper, but did not succeed. I suggest that this whole section is eliminated, summarized in the Supplementary material or made part of a separate manuscript.
2. The authors suggest that (page 20) APOBEC3-mediated deamination at CpG sequences may contribute to mutations. This is confusing. If the CpGs are methylated, 5-methylC is a poor substrate for most APOBEC3s except A3A (PMID 22798497, among others). However, A3A is not detectably expressed in these cells (page 15 and Fig. 8). One way to settle this point is to do bisulfite sequencing to map 5mCs in the genome and to correlate the CpG to TpG mutations to methylation or the lack thereof.
3. The authors speculate (page 17) that "...high uracil signal originates from thymine-replacing uracil incorporation during transcription-coupled repair synthesis around active genes...". This seems unlikely. If this were true, the uracils should predominantly occur in the template (transcribed) strand. Can the authors confirm this? Can they create CsbA/CsbB KO cell lines to make the cells TCR defective and show that this uracil enrichment disappears?
4. If the C to T mutations are indeed caused by APOBEC3B, they should be predominantly occur in hairpin loops (PMID 38499542 and 38499553). This should be confirmed.
5. It is relatively straightforward to detect APOBEC3 activity in cell-free extracts. This should be done.
6. Does MMR repair the U:G mispairs created by cytosine deaminations by APOBEC3s in this system? It is well known that uracils created by AID are subject repair by MMR (PMID: 16894013).
7. If the SBS2 signature mutations created as a result of expression of APOBEC3s, why do they not also see SBS13 signature? In summary, this interesting manuscript advances our understanding how and where 5FdUR and RTX cause uracil incorporation in DNA, but the manuscript should be simplified and shortened to make it more understandable and to increase its impact on the field.

Reviewer #3 (Comments to the Authors (Required)):

1. **MANUSCRIPT SUMMARY:** This manuscript investigates the differential genomic uracilation patterns and mutagenic potential elicited by two thymidylate synthase (TS) inhibitors-raltitrexed (RTX) and 5-fluoro-2'-deoxyuridine (5FdUR)-in HCT116 colon cancer cells. The study leverages UNG-inhibited and MMR-deficient cellular models to elucidate how these chemotherapeutics affect genomic uracil accumulation, mutational signatures, and downstream cellular consequences.

The authors demonstrate that the two drugs produce distinct uracilation profiles associated with replication timing and genomic features. Notably, 5FdUR, particularly at high doses, induces a C-to-T mutational signature enriched in APOBEC3-related motifs, suggesting involvement of APOBEC3 deaminases. These changes coincide with reduced cytotoxicity, highlighting potential implications for therapeutic resistance.

The work presents a comprehensive multi-omic analysis (U-DNA-Seq, RNA-Seq, and whole genome sequencing) supported by robust computational approaches, such as Segway genome segmentation and U-score quantification.

This study offers a valuable mechanistic advance by linking TS inhibitor-induced uracil incorporation with mutagenesis, gene expression modulation, and DNA repair status-key for understanding treatment efficacy and resistance in colorectal cancer.

The work is suitable for Life Science Alliance due to its relevance, methodological rigor, and translational implications.

2. EVALUATION OF KEY CLAIMS AND SUPPORTING DATA

Main Points and Experimental Support

A. Drug-specific genomic uracilation patterns (Strong support provided)

- o The use of Segway for genome segmentation is appropriate and well executed. Distinct segment clusters (e.g., M2 vs. M5) are convincingly shown to respond differentially to RTX and 5FdUR.
- o Supplementary Figure S1 and Jaccard analysis support reproducibility across replicates.
- o The correlation with replication timing and AT content adds mechanistic insight.

B. Gene-level uracilation and U-score analysis (Moderate-to-strong support provided)

- o The U-score is a useful quantitative measure. However, further validation (e.g., correlation with independent uracil-detection methods like Excision-seq or qPCR validation of highly uracilated genes) would strengthen conclusions.
- o The statistical testing is appropriate but could benefit from FDR-adjusted p-values to control for multiple comparisons.

C. Induction of C-to-T mutations by 5FdUR (Strong support)

- o Whole-genome sequencing in UNG- and MMR-deficient cells is a major strength. The enrichment of APOBEC-related signatures is well demonstrated.
- o APOBEC3 expression analysis supports mechanistic claims.

D. APOBEC involvement in mutagenicity (Moderate support)

- o While RNA-level data support increased APOBEC3 expression, the conclusions would be bolstered by inclusion of protein-level or activity assays (e.g., deamination assay or western blot of A3B, A3C, A3H).

E. Implications for cytotoxicity and therapeutic resistance (Weak-to-moderate support)

- o Cytotoxicity data are briefly mentioned but should be more fully integrated. Dose-response curves and statistical comparisons between RTX and 5FdUR (with and without MMR) would clarify this key claim.

MAJOR REVISIONS:

1. Expand cytotoxicity data:

- o Include cell viability assays comparing RTX and 5FdUR across MMR/UNG status.
- o Include dose-response curves with statistical analysis.

Estimated timeframe: 2-3 weeks

2. Functional validation of APOBEC3 involvement:

- o Provide protein expression data or enzymatic activity assays for APOBEC3s implicated in mutagenesis.
- o Consider knockdown/overexpression validation to directly link APOBECs to observed mutational signatures.

Estimated timeframe: 4-6 weeks

3. Statistical refinement in U-score and DU gene analysis:

- o Apply false discovery rate (FDR) correction to the Welch's t-tests.
- o Include volcano plots or MA plots for clarity.

Estimated timeframe: 1 week

4. Clarify uracil quantification and normalization:

- o Explain normalization of enrichment signals in more detail (e.g., rescaling using dot blot values).
- o Clarify how background uracil levels are accounted for.

Estimated timeframe: <1 week

3. ADDITIONAL MINOR REVISIONS

Abstract: The term "cellular responses" is vague; specify e.g., "differential cytotoxicity and mutational profiles."

Introduction: Minor typographical issues (e.g., "quenstions" → "questions").

Figure legends: Some are too brief (e.g., Figures 1-2). Please expand to describe axes, data type, and interpretation.

Terminology: Define "U-score" earlier and more precisely.

Methods: Consider moving some code blocks to a supplementary GitHub repository or Zenodo archive with DOI for reuse and transparency.

4. REFEREE CROSS-COMMENTS

I have reviewed the comments provided by Reviewers #1 and #2.

Reviewer #1 provides a thorough summary of the manuscript's key findings and acknowledges the relevance of the drug-specific uracilation and mutational profiles. I agree with their observation that the MMR-dependent difference in APOBEC signatures is intriguing and merits mechanistic clarification. Their request for clearer contextualization of the gene network analyses aligns with my suggestion to statistically refine the U-score and DU gene data. However, I believe that rather than being extraneous, this section adds value if it is streamlined and interpreted more clearly. Their suggestion for a unifying graphical model is a strong one and would help integrate the mechanistic findings.

Reviewer #2 raises valid concerns regarding the complexity of the genome segmentation and uracilation analysis, but I do not support the recommendation to eliminate Pages 5-10 entirely. While the segmentation analysis is computationally dense, it underpins the manuscript's major claims and can be clarified rather than discarded. I agree with Reviewer #2 that functional validation of APOBEC3 activity (e.g., biochemical assay or knockdown models) would substantiate the mechanistic claims, a point I also raised. Their comment regarding the need to reconcile the lack of SBS13 with SBS2 presence is well-taken and should be addressed.

Both reviewers call for clarification of mechanistic links-especially between APOBEC expression and mutational signatures. My own report similarly recommended expanding cytotoxicity profiling, adding APOBEC protein-level validation, and enhancing statistical robustness. These revisions appear to be consistent across reviews.

In summary, while Reviewer #1 and I are largely aligned in assessing the manuscript's strengths and revision needs, Reviewer #2 recommends more drastic structural changes. I would advocate for selective refinement and improved clarity rather than major content removal.

October 7, 2025

Re: Editorial evaluation with Reviewers' comments

To: Sarita Hebbar, PhD, Scientific Editor

Life Science Alliance

<http://www.lsjournal.org>

Life Science Alliance manuscript #LSA-2025-03352

Dear Dr. Sarita Hebbar,

We thank you and the reviewers for the careful and constructive evaluation of our manuscript entitled " The effects of thymidylate synthase inhibitors differ in genomic uracilation and mutagenic potential" #LSA-2025-03352 .

We sincerely appreciate the thoughtful feedback, the editor's efforts in coordinating this thorough review process.

General note on reviewers' cross-comments:

We appreciate the reviewers' constructive engagement and evaluation. We have carefully considered the consensus among Reviewers regarding the need for clearer mechanistic connections, refined statistical analysis, and a more concise presentation. Accordingly, the revised manuscript now includes:

1. Detailed information on the drug doses used, along with statistical analyses of the dose-response curves and cell viability assays.
2. A more succinct contextualization of the gene networks affected by genomic uracilation and a reconsidered statistical analysis of U-scores and gene-level data.
3. Increased APOBEC3 mRNA expression is supplemented with protein-level and activity data. Furthermore, hairpin-loop analyses revealed the highly clustered nature of C-to-T transitions, pointing toward A3C involvement.
4. Thorough language and stylistic revisions to correct grammatical errors and improve clarity, flow, and overall presentation.

Please, find our detailed response below point-by-point.

Sincerely yours,

Angéla Békési

Budapest, October 7th, 2025.

Angéla Békési PhD

Senior lecturer and research fellow
Genome Metabolism and Biostruct Research Group

Department of Applied Biotechnology and Food Sciences
Faculty of Chemical Technology and Biotechnology
Budapest University of Technology and Economics
Műegyetem rakpart, Budapest, 1111, Hungary

Editorial instruction:

phone: +36208234391
e-mail: bekesi.angela@vbk.bme.hu
web: www.biostruct.org

Institute of Molecular Life Sciences
Research Centre for Natural Sciences
Hun-Ren
Magyar Tudósok krt 2, Budapest, 1117

A revised manuscript must include the following:

1. Drug treatments:

Information on doses used, statistical analyses on dose-response curves, and cell viability assays (Reviewer 1, points 1,3 and Reviewer 3, major revisions point 1)

Response to the Editor's point 1:

In the original submission, drug doses were described in the Methods section and summarized in Table 3, with references to our previously published U-DNA-Seq data also. In the revised manuscript, we have made this information more accessible by explicitly adding dose details in the first paragraph of the Results section and in all relevant figure legends (Figures 1, 2, 4, 8, and Supplementary Figures S3 [previously S5] and S5 [previously S7]).

A comprehensive statistical analysis of dose–response curves is now clearly indicated. This analysis, originally presented in Supplementary Figure S11 together with the corresponding source data file, includes Welch's t-test p-values for all 320 pairwise comparisons. To improve clarity, we have also added a sentence to the legend of Figure 6B: "Detailed statistical analysis is provided in Supplementary Figure S11."

2. Differences in genomic uracilation:

A more succinct contextualisation of gene networks (Reviewer 1, point 2) and a statistical refinement in U-score and gene analyses (Reviewer 3, major revisions point 3)

Response to the Editor's point 2:

We thank the reviewers for their valuable suggestions regarding both the contextualization of differential uracilation and the statistical refinement of U-score and gene analyses.

We agree that the original description of gene networks and functional associations could have been more concise. Accordingly, in the revised manuscript, we have substantially shortened and reformulated the relevant *Results* sections (tracked in the revised version and detailed in our response to Reviewer 1, point 2). Specifically, we removed Supplementary Figures S3 and S4, while retaining Figure 3, the permanent links to the STRING analysis (Table 1), and the GSEA network visualizations (now Supplementary Figure S3, previously S5) along with their corresponding Table 2 and source data files.

In line with Reviewer 3's recommendation, we performed FDR correction for the differential uracilation (DU) gene analysis. No genes reached an FDR < 0.05 threshold, likely reflecting the modest effect size and biological variability intrinsic to this dataset. Nevertheless, 312 genes met the criteria of raw $p < 0.05$ and $|\log_2FC| > 0.585$.

Therefore, we now report raw p -values in Figure 2C, clearly indicate this choice in the figure legend, and explicitly state in the text that no genes reached FDR significance.

Originally, only one approach out of the four complementary GSEA approaches relied on the groups of significantly uracilated genes. The other three approaches applied to the U-score data do not depend on individual gene-level significance; their results and interpretations remain valid. We removed the entries corresponding to DU genes from Tables 1 and 2 and from Source Data File 6, without affecting the main conclusions, which have been streamlined in the revised version.

Finally, for the GSEA itself, the FDR correction was already incorporated into the algorithm, and we further reduced redundancy and complexity by applying stricter enrichment filters. All this data is available in the Source Data File 6.

3. Validation of APOBEC3 involvement:

Increased APOBEC3 mRNA expression should be supported with protein expression and activity (Reviewer 2, point 5 and Reviewer 3, major revisions point 2)

Functional linking of APOBEC to mutational signature by method of choice: overexpression or knockdown (Reviewer 3, major revisions point 2) OR hairpin loop analyses (Reviewer 1, point 4)

Response to the Editor's point 3:

We thank the reviewers for their constructive comments regarding the validation of APOBEC3 expression and activity. In the original manuscript, besides the RT-qPCR evidence of increased mRNA levels of APOBEC3s, we provided immunofluorescence microscopy data (Figure 9) about *APOBEC3B* at the protein level, which demonstrated its nuclear localization and induction upon both drug treatments.

In the revised manuscript, we now provide additional data at both the protein and activity levels. For other APOBEC3 enzymes, we tested several commercially available antibodies, which showed variable sensitivity and specificity. Nevertheless, we were able to obtain Western blot data for *APOBEC3C* and *APOBEC3D*. In particular, *APOBEC3C* displayed a modest increase in the chromatin-bound nuclear fraction after 48 h of high-dose 5FdUR treatment (Figures 9C–E). Although this change did not reach statistical significance, these results, along with detailed methodological descriptions, have been incorporated into the revised *Results*, *Materials and Methods*, and *Supplementary Methods* sections.

As protein levels do not necessarily reflect catalytic activity, accepting the reviewers' suggestions, we also addressed APOBEC3 function directly. Following Reviewer 2's suggestion, we performed cytidine deaminase activity assays using cell-free extracts and fluorescently labeled oligonucleotide as substrate. As oligonucleotide-based assays

mainly reflected background deamination activity (Supplementary Figure S14), we set up a 3D-PCR analysis for a defined genomic locus (MDM2). This later experiment revealed a slight shift in denaturation temperature, specifically in high-dose 5FdUR-treated samples (Figure 8C), indicating a small fraction of mutated genomic templates.

To further link mutational spectra to APOBEC3 activity, we analyzed the sequence context and structural environment of the C-to-T transitions. Although no enrichment in hairpin-forming sequences was detected, we observed a strong clustering tendency – an established hallmark of *APOBEC3C* activity. These results are now presented in Supplementary Figure S11C.

Together, these new data strengthen the connection between 5FdUR treatment and APOBEC3-mediated mutagenesis, while acknowledging the technical limitations inherent to detecting small endogenous changes in APOBEC protein levels and activity.

4. Edits to eliminate grammatical errors, and improve clarity and presentation as indicated by all the reviewers (Reviewer 1, point 2, Reviewer 2, point 1, and Reviewer 3, major revisions point 4)

Response to the Editor's point 4:

We thank the reviewers for their careful reading and valuable suggestions regarding grammar, clarity, and presentation. In response, we carefully revised the entire manuscript with a comprehensive grammar and style check (including the use of Grammarly) to eliminate errors. The corrected minor issues are not individually indicated in the revised manuscript in order to maintain readability. We also reformulated several sentences or even whole paragraphs to improve clarity; these revisions are shown in change-tracking mode. (It has to be mentioned that due to fatal conflicts between the citation manager Mendeley Cite and the MS Word track changes function, mislabeling our changes can occur, although we tried our best.) We believe the manuscript is now significantly clearer and polished in line with the reviewers' expectations.

Reviewer #1 (Comments to the Authors (Required)):

The authors examined the uracilation profiles of HCT116 cell lines to compare the effects of RTX and 5FdUR. This validated known associations, like early replication timing, but identified several genome segments where there were drug specific effects in gene coding regions. They next examined gene specific uracilation and again found differences, as well as a strong effect of MMR status on the effects. They carried out a functional enrichment analysis using STRING and found genes involved in specific processes were differentially affected by the drugs and MMR status. They next looked at the types of mutational differences and found more C:G to T:A transitions with 5FdUR and signature analysis implicated APOBEC3 activity. These, as well as a DBS signature were dependent on MMR deficiency. They examined the effects of the drugs on cell proliferation and found both dose and drug dependent effects on the cell cycle and viability, with a reduced sensitivity at high doses. Analysis of A3 expression and mutational signatures suggested a contribution from A3B and A3C, but not A3A. A3B was the most dramatically upregulated following treatment with either drug, although the clinical correlations were highest with A3C. Finally, they performed IF and directly confirmed A3B induction with either drug.

Building on previous work, the paper identifies an interesting uracilation pattern difference between the 2 clinically relevant drugs that is dependent on MMR status and appears to involve A3B, or potentially other family members. The paper generally follows a logical arc but some of the data analysis is less informative and seems extraneous for making the point (see specific comments below). I think the paper would benefit greatly from a graphical summary (aside from the abstract) that gives a more unifying idea of the proposed mechanistic underpinning of the difference in the context of existing literature, I remain somewhat confused as to why the authors think that 5FdUR yields an MMR deficiency dependent A3 signature when both agents upregulate the protein pretty extensively- what would account for this interesting observation? I think that while the analysis was extensive, this observation is also coming from the background of a single cell line that carries a number of DNA repair related mutations, potentially yielding some cell type specific observations that should be considered.

Responses to Reviewer 1's general evaluation:

We thank the Reviewer for the careful and thoughtful evaluation of our manuscript. We appreciate the suggestion to simplify certain parts of the data analysis to make the message more focused. In the revised version, we have substantially shortened some explanatory sections and improved clarity throughout, as can be seen in the track changes.

We were indeed intrigued to find that although both drug treatments strongly induced A3B expression, their mutational spectra differed markedly. We propose that this discrepancy may be explained by additional layers of regulation – such as post-translational modifications of A3B, interactions with protein or RNA partners, or

differences in the accessibility of single-stranded DNA – which together may limit the manifestation of cytosine deamination activity across the genome.

Moreover, our data suggest that A3C may also contribute to the genomic C-to-T transitions, as it shows a more pronounced 5FdUR-specific induction (qPCR, Western blot; see Fig 8A–B and Fig 9C in the revised manuscript). Notably, A3C is the most abundant APOBEC3 family member in HCT116 cells, with expression levels approximately one order of magnitude higher than A3B (based on qPCR Cq values in the Source Data File 16 and SMRT sequencing data from the ENCODE database). In addition, our mutational spectrum reconstitution (Supplementary Fig S11B) and loop analysis (Supplementary Fig S11C in the revised manuscript) further support a prominent role for A3C in the 5FdUR-induced C-to-T transitions. These aspects have now been elaborated and integrated more clearly in the revised *Discussion* section.

Regarding the possibility of cell-type-specific effects, we fully agree with the Reviewer. The HCT116 cell line carries multiple DNA repair deficiencies, creating a background of ongoing mutagenesis. Indeed, several APOBEC3 family members are constitutively expressed in these cells even in the absence of treatment. Consistent with this, the mutational spectra calculated from somatic variants (weighted by allele frequencies) show a stable background signature across all samples (Fig. 4 and Supplementary Fig. S5 in the revised manuscript). Despite the stable background, the 5FdUR-specific increase in C-to-T frequency stands out clearly, with a distinct and characteristic pattern.

Specific comments

Reviewer 1 point 1. What is the drug dose used in the initial analysis in Figure 1? No where in the text or Figure legend can I locate this and another paper is referenced. As it is relevant to comparing this data with data later in the paper that uses different doses, it needs to be clarified and in the Figure legend of this paper.

Answer: We thank the Reviewer for pointing out that the applied doses were not consistently indicated in the text and figure legends. We have now revised the manuscript to ensure that the doses are clearly and uniformly specified throughout. Originally, we referred to the previously published U-DNA-Seq data and included the dose data in the Methods, clearly emphasized in Table 3. We have now inserted dose information into the first paragraph of the Results as well as into each relevant figure legend, namely: Figures 1, 2, 4, 8, and Supplementary Figures S5 and S7.

Reviewer 1 point 2. Given the amount of attention paid to a deep analysis of the gene functional networks in Table 1, 2, Figure 3 and multiple supplemental Figures, a clearer point needs to be made here. To me the 2 most obvious likelihoods are that these are genes that are particularly active in the particular cell line, and that chromatin is thus exposed and accessible to mutagenesis or, less likely, these are selected for function to adapt to the treatments. Is there gene expression or ChIP data that could be applied to this to see if this is related to relative expression in the different cell lines? As it is, it is

an enormous amount of data and text that is of unclear relevance to the mechanism and I find that it disrupts the story more than making anything clearer. It makes a minor observational point that there are cell line differences, but beyond that it is unclear to me the purpose of including all of this in the middle of the story.

Answer: We thank the Reviewer for this valuable comment. We agree that the contextualization of the differential uracilation and its functional implications could have been presented more succinctly.

Based on the strong correlation with early replication timing, we are convinced that the primary mechanism underlying genomic uracil enrichment in these S-phase–arrested cells is uracil incorporation during replication-related DNA synthesis (cf. Fig 1, Supplementary Fig S1A). However, the drug-specific differences in genomic uracil profiles or U-scores cannot be explained solely by S-phase arrest and limited DNA synthesis. We also agree that uracil incorporation is unlikely to represent a targeted event that specifically drives adaptation to the treatments. Instead, we consider that uracilation may reflect transcriptional activity and/or open chromatin states, including replication forks, transcription bubbles, repair synthesis, or recombination events. For example, high transcriptional activity can contribute to uracil incorporation through transcription-coupled repair synthesis (PMID: 30016327). Conversely, elevated uracil levels within gene bodies may interfere with transcriptional processes (PMID: 30892639). Altogether, these points suggest a dynamic and complex interaction between uracilation, transcriptional activity, and chromatin state. As the Reviewer also implied, only time-resolved transcriptional or multi-omics analyses could provide a deeper understanding of the biological relevance of these observations, which would extend beyond the scope of the present manuscript.

Nevertheless, the interconnectivity and functional enrichment we observed among the most uracilated genes suggest that U-score differences reflect traces of molecular events during the cellular response to drug treatment, with higher uracilation serving as an indicator of pathway involvement. Specifically, 5FdUR-related uracilation is enriched in homeobox transcription factor genes and genes involved in cell-cell communication and extracellular processes, whereas RTX-induced uracilation predominantly affects genes linked to intracellular functions, such as RNA processing, cell cycle regulation, and DNA repair. While we cannot fully interpret the meaning of these differences at this stage, we believe that they provide intriguing insights into how the two TS inhibitors may exert distinct effects.

To improve clarity and focus, we have substantially revised this section in the Results (changes are tracked in the revised manuscript). In particular, we removed Supplementary Figures S3 and S4, while retaining Figure 3, the permanent links (Table 1), the source data for the GSEA analysis, and the network figures (now Supplementary Figure S3, previously S5) together with Table 2.

Reviewer 1 point 3. In Figure 6B, data is provided that there are differences in toxicity at different doses. Are any of these differences statistically significant? Has this been

validated using colony forming assays or another approach? I worry somewhat that these agents would have significant mitochondrial effects, particularly at high doses that would impact these types of assays that are measuring metabolites.

Answer: We thank the Reviewer for this important point. A detailed statistical analysis of the dose–response curves is already provided in Supplementary Figure S11, where p-values were calculated by Welch’s t-test for each of the 320 pairwise comparisons. To make this clearer, we have now added the following note to the Figure 6B legend: *“Detailed statistical analysis is provided in Supplementary Figure S11.”*

Regarding the suggestion of colony-forming assays, we have not performed these experiments. In our hands, HCT116 cells do not tolerate high dilution or cell-sorting conditions well, which would compromise the reliability of colony formation as a readout in this model. We therefore consider this approach less suitable for our experimental system.

Instead, we relied on the CCK8 assay, which is widely used and accepted in the field as a robust measure of cell viability, proliferation, and metabolic activity. CCK8 is based on a water-soluble tetrazolium salt (WST-8) with improved sensitivity, stability, and lower toxicity compared to traditional MTT or XTT assays, enabling more accurate measurements over a broad dose range without introducing additional stress conditions. Its reliability and reproducibility are well-documented in the literature (e.g., Gong et al., 2023; Guan et al., 2022; Kurasaka et al., 2022).

Furthermore, the cell cycle analysis presented in Figure 6A provides an independent line of support for these results: at high doses of 5FdUR, we observed a weaker S-phase arrest compared to lower doses or any doses of RTX, which is consistent with the reduced cytotoxicity measured by CCK8. Thus, both assays converge on the same conclusion regarding dose-dependent cellular effects.

Taken together, we are confident that the CCK8 assay, in combination with the cell cycle data, provides a physiologically relevant measure of cell viability in our system, and that the statistical analyses adequately support the observed dose-dependent differences.

Reviewer 1 point 4. In Figure 8C, the survival of patients correlated to the expression of individual APOBECs is plotted. This is interesting in itself, but trying to connect this to the data from a single cell line is a pretty massive leap. Is there any evidence in RTX or 5FdUR treated patients that any APOBECs are more highly expressed? This would provide more compelling evidence for a clinical connection than survival that may have nothing whatsoever to do with the mutagenesis observations from cell lines.

Answer: We thank the Reviewer for this important and thoughtful comment. We fully agree that correlating APOBEC expression with patient survival represents a considerable extrapolation from our single-cell-line model, and we acknowledge that the original presentation in Figure 8C may have overstated its direct relevance. In the revised manuscript, these plots have been moved to Supplementary Figure S15, where they are provided as contextual rather than central evidence.

Our motivation for including these analyses was to explore whether the genome editing activity of endogenous APOBECs observed in our study, particularly that of APOBEC3C, might have broader clinical implications. As small endogenous changes in APOBEC levels and activities are difficult to quantify due to the semi-quantitative nature of immunocytochemistry and Western blotting and the limited sensitivity of available antibodies, we sought complementary in silico data for perspective.

The observed reduction in short-term cytotoxicity may suggest increased survival of cancer cells, in the long term, potentially contributing to disease progression or drug resistance in patients and thereby a worse prognosis. Still, the results of this in silico analysis of patient survival data unexpectedly showed that among the four APOBEC3 enzymes examined, only APOBEC3C had an association with worse patient survival (indicative of stronger cancer progression). This observation is consistent with our experimental data, where APOBEC3C emerged as the most abundant APOBEC3 in HCT116 cells and was implicated as a major contributor to the 5FdUR-induced mutational spectra. While we fully acknowledge that survival correlations cannot be interpreted as direct evidence of drug-specific mutagenesis in patients, they may hint at a clinically relevant role of APOBEC3C in drug response and tolerance.

Unfortunately, we were unable to identify publicly available datasets reporting APOBEC expression specifically in RTX- or 5FdUR-treated patients. While we fully acknowledge that these correlations cannot be interpreted as direct evidence of drug-specific mutagenesis in patients, and discussed these in silico results accordingly in the revised manuscript.

Reviewer 1 point 5. How do the authors reconcile the observation that RTX at low dose and 5FdUR at high dose (Figure 8B) cause similar levels of A3B upregulation but the mutagenic outcome is distinct related to the A3B signature?

Answer: We were indeed surprised to find that strong induction of A3B by both drug treatments did not result in similar mutational spectra. We propose that this discrepancy reflects additional regulatory layers – such as post-translational modifications of A3B, interactions with partners, or differential accessibility of single-stranded DNA – that may limit the manifestation of cytosine deamination activity in the genome. These considerations are more explicitly discussed in the revised manuscript. Moreover, our data indicate that APOBEC3C, which is expressed at much higher levels than A3B in HCT116 cells (Source Data file 16; ENCODE SMRT-seq), also contributes significantly to the 5FdUR-induced C-to-T transitions. This is supported by its drug-specific induction (Fig. 8AB, Fig. 9B) and by mutational spectrum reconstitution and loop analyses (Supplementary Fig. S11B-C). Now, in the revised manuscript this is further strengthened by Western blotting (cf Fig 9C-E) Together, these findings suggest that A3C plays a more prominent role in 5FdUR-induced mutagenesis, thereby could explain the divergence from an A3B-like signature.

Reviewer 1 point 6. The dependency of MMR deficiency on the mutational signature is seemingly at odds with observations from previous studies that found MMR was required for A3 clustered signatures (<https://doi.org/10.1016/j.cell.2017.07.003>). Has this

been considered and do the authors have any thoughts on the mechanistic implications?

Answer:

We thank the Reviewer for directing our attention to this relevant study (*Supek & Lechner, Cell, 2017*), which indeed provides a comprehensive analysis of clustered mutational signatures in tumors, including APOBEC-associated ones. As discussed below, our observations are not in conflict with those findings, but rather reflect the distinct experimental context and time scale of our study.

The cited paper describes three APOBEC-related mutational signatures (C1, C2, and C3), of which only C1 is similar to our 5FdUR-specific spectra. The authors attributed this signature primarily to APOBEC3B, based on its correlation with expression data and the site preference (RTC). Notably, they did not consider the potential contribution of other APOBEC3 family members whose expression also correlated positively with clustered mutational spectra. In their interpretation, the differences between the inter-mutation distances in C1 and C2 signatures reflect processive versus less processive repair processes.

A key difference between that study and ours lies in the biological context. They investigated mutations accumulated during tumor progression, whereas our work focuses on acute, drug-induced mutagenic changes within a 48-hour timeframe. Thus, the variants we detect can originate also from non-repaired U:G mismatches (either instant U:Gs or T:Gs fixed by replication) and not only by long-term accumulation involving error-prone repair mechanisms during tumor progression.

Furthermore, the cited paper implicates MMR activity specifically in the C4 signature, which involves clustered T→C mutations enriched in H3K36me3-marked active chromatin regions of solid tumors. These events were shown to depend on non-canonical MMR involving the translesion polymerase η (pol η). Loss of canonical MMR in tumors abolished enrichment of these clustered T→C events. Moreover, this C4 signature did not directly involve APOBEC-mediated cytosine deamination.

In comparison, our system shows a modest increase in T→C substitutions across both MMR-proficient and -deficient cells treated with RTX or 5FdUR (Fig. 4 and Supplementary Fig. S5), without drug-specific differences. Even non-treated cells give similar background signatures to drug-treated samples when variants are called against wild-type HCT116 data (Fig 4). Hence, we interpret this signal as part of stable background mutagenic processes intrinsic to this cancer cell line, which are independent of drug treatment and resemble those seen in solid tumors (as described in the cited paper).

Importantly, this background does not contradict our key findings. Instead, it underscores that the 5FdUR-induced mutational spectra emerging from this background are driven by acute cytosine deamination events during the 48-hour treatment. The resulting U:G mismatches persist primarily in MMR-deficient and UNG-inhibited (UGI-expressing) cells, giving rise to the characteristic C→T transitions we observe. The

clustered nature of these C→T mutations likely arises from the processivity of APOBEC enzymes, rather than from subsequent error-prone repair. Recently, APOBEC3C has been characterized with a highly clustered C-to-T signature, which was even more clustered than in the cases of A3A or A3B (Brown, Genetics, 2024). In contrast, in MMR-proficient but UNG-inhibited cells, these U:G pairs can be accurately repaired via canonical, error-free MMR, particularly in S-phase–arrested cells where replicative polymerases remain available. Accordingly, 5FdUR-treated MMR-proficient cells lack both the C→T signature and the polη/MMR-associated T→C pattern reported in tumors (cf. Fig S5).

Overall, we consider that the clustered T→C signatures dependent on non-canonical MMR and polη activity, as described in (*Supek & Lechner, Cell, 2017*), reflect slow, cumulative mutational processes during tumor evolution, whereas the drug-induced signatures in our study capture rapid and transient mutagenic events arising from immediate APOBEC-mediated cytosine deamination.

Reviewer #2 (Comments to the Authors (Required)):

This manuscript extends the results of a previous study (Palinkas et al eLife 2020) in which these authors mapped uracils in the genomes of UNG inhibited MMR-defective cells. The principle conclusion here is that the accumulation and distribution of uracils and mutations following treatment of cells with 5FdUR or raltitrexed (RTX) are substantially different from each other. This is not surprising in itself as the two compounds inhibit dTMP synthesis in very different ways. The more interesting observation is that the two drugs cause a modest increase in APOBEC3s- but not A3A (Fig. 8)!! Somewhat paradoxically 5FdUR treatment, but not RTX treatment, creates a mutational pattern associated with the APOBEC enzymes, the COSMIC signature SBS2. These observation increase our understanding of how two frequently used anticancer compounds, 5FdUR and RTX, affect human cells. Unfortunately, the manuscript contains a lot of low value data and speculation weakening the manuscript. Eliminating these sections would strengthen and focus the manuscript.

Response to Reviewer 2's general evaluation: We thank the Reviewer for the thoughtful assessment of our work and for highlighting the key findings regarding the differential effects of 5FdUR and RTX on genomic uracilation and mutagenesis. We agree that the previous version of the manuscript included sections that were overly detailed or speculative, which may have obscured our main message.

In the revised version, we have substantially condensed and refocused the Results and Discussion sections, particularly those describing differential uracilation patterns and their potential functional implications. Redundant or speculative interpretations were minimized or replaced with concise, evidence-based statements. These revisions have significantly improved the overall clarity and readability of the manuscript, allowing the main findings – namely the drug-specific genomic uracil profiles and the distinct mutational outcomes associated with APOBEC activity – to be presented more directly and convincingly.

Reviewer 2 point 1. The significance of the first three sections in the results (Pages 5 through 10) is unclear. As dUTP and 5FdUTP are expected to be incorporated during replication, the extensive analysis of which groups of genes accumulate uracils does not make much sense. Most of these sections involve complex statistical and computational analysis that is difficult to understand. I tried to understand genome segmentation analysis by reading this manuscript, the Palinkas et al manuscript and the original NAR paper, but did not succeed. I suggest that this whole section is eliminated, summarized in the Supplementary material or made part of a separate manuscript.

Answer:

We appreciate the reviewer's thoughtful comment and fully agree that uracil incorporation is strongly influenced by DNA synthesis during replication. Indeed, our data confirm that U-DNA distribution correlates with early replication timing in S-phase–

arrested cells. However, the main purpose of these analyses was to reveal and explain **drug-specific differences** beyond this general replication-associated pattern, pointing also to a contribution from repair synthesis.

We demonstrated that RTX treatment leads to a stricter S-phase arrest (cf. Fig 6A) compared to 5FdUR, consistent with the stronger correlation observed between RTX-associated uracil enrichment and early replication timing (cf. Fig 1). Importantly, we identified specific genomics segments where uracil enrichment differed markedly between RTX- and 5FdUR-treated cells, particularly within early-to-mid replication timing regions.

These differences were also evident at the level of genes, where **highly uracilated genes displayed functional coherence** (cf. Fig 3 and Fig S3), suggesting that uracil incorporation may reflect chromatin accessibility and transcriptional or repair-associated activity rather than replication alone. We therefore consider it important to retain these results in the main manuscript, as they reveal mechanistic distinctions between the two drugs that would be obscured if moved entirely to the Supplementary Material. Importantly, in line with the suggestions of other reviewers, we have markedly shortened this section on differential uracilation and refined it to focus more directly on the key insights.

As a more pronounced drug-specific difference, **5FdUR treatment led to an increased frequency of C-to-T transitions (cf. Fig 4)**, indicative of APOBEC-mediated cytosine deamination. The resulting U:G mispairs in MMR-proficient cells can trigger extensive repair synthesis independently of replication, providing a plausible explanation for the altered uracil distribution observed in these cells.

Thus, differences in both **replication arrest strength and repair synthesis activity** likely account for the distinct genomic uracil patterns, demonstrating that genomic uracil profiling can serve as a sensitive readout of underlying molecular mechanisms.

In summary, while we agree that replication is the major source of uracil incorporation, our analyses highlight important **drug-specific variations** that go beyond this general mechanism. We have also **reformulated the description of the genome segmentation analysis** to improve clarity and readability, as indicated in the change-tracked revised manuscript.

Reviewer 2 point 2. The authors suggest that (page 20) APOBEC3-mediated deamination at CpG sequences may contribute to mutations. This is confusing. If the CpGs are methylated, 5-methylC is a poor substrate for most APOBEC3s except A3A (PMID 22798497, among others). However, A3A is not detectably expressed in these cells (page 15 and Fig. 8). One way to settle this point is to do bisulfite sequencing to map 5mCs in the genome and to correlate the CpG to TpG mutations to methylation or the lack thereof.

Answer: We appreciate the reviewer's insightful comment and fully agree that most APOBEC3 enzymes show limited activity on methylated cytosines, as demonstrated for

AID, A3A, and A3G in the cited study (Wijesinghe & Bhagwat et al., 2012). In the revised Discussion, we have acknowledge that 5-methylcytosine is generally a poor substrate for most APOBEC3s. Nevertheless, subsequent comprehensive analyses (e.g., Ito et al., *J. Mol. Biol.*, 2017) revealed that certain APOBEC3 enzymes, including A3H and, to a lesser extent, A3B, are capable of deaminating 5-methylcytosine in vitro. Since both A3H and A3B are expressed in our model, we consider it plausible – although not directly proven – that these enzymes may contribute to the increase of CpG-associated TG-to-CA transitions observed in the 5FdUR-treated samples.

We have accordingly revised and shortened this speculative part of the Discussion to avoid implying a direct causal relationship. We also shortened and refocused the relevant section to ensure that this point does not distract from our main conclusions.

Reviewer 2 point 3. The authors speculate (page 17) that "...high uracil signal originates from thymine-replacing uracil incorporation during transcription-coupled repair synthesis around active genes...". This seems unlikely. If this were true, the uracils should predominantly occur in the template (transcribed) strand. Can the authors confirm this? Can they create CsbA/CsbB KO cell lines to make the cells TCR defective and show that this uracil enrichment disappears?

Answer: We thank the reviewer for this thoughtful comment. We agree that our interpretation of the uracil enrichment near transcriptionally active genes is speculative, and we have now clarified this in the revised text. We also agree that a strand-specific uracil mapping experiment or the generation of transcription-coupled repair (TCR)-deficient cell lines could, in principle, provide further insight. However, such genetic interventions would represent a major perturbation of the system and could substantially alter the overall mutational and repair landscape, complicating the interpretation of drug-specific effects. Creation of such new cell lines is definitely beyond the scope of the present manuscript.

In the revised version, we have shortened and refocused this part of the Discussion to emphasize that this interpretation remains speculative and to make the section more concise, in line with the reviewer's recommendation.

Reviewer 2 point 4. If the C to T mutations are indeed caused by APOBEC3B, they should be predominantly occur in hairpin loops (PMID 38499542 and 38499553). This should be confirmed.

Answer:

We appreciate the reviewer's insightful comment and fully agree that analyzing the structural context of the C-to-T mutations provides important mechanistic information. To address this point, **we have now performed a loop-structure analysis** for all detected C-to-T transitions, separately evaluating clustered and non-clustered events. Our results (now presented in **Supplementary Figure S11C**) show **no preference of C-to-T**

mutations for hairpin loop regions, but reveal a strong tendency for clustering. This pattern is more consistent with APOBEC3C-mediated mutagenesis rather than APOBEC3B, which aligns with our other observations suggesting a predominant contribution of A3C in the 5FdUR-induced mutational spectra.

Reviewer 2 point 5. It is relatively straightforward to detect APOBEC3 activity in cell-free extracts. This should be done.

Answer: We appreciate the Reviewer's suggestion and indeed made efforts to assess endogenous APOBEC-mediated cytosine deamination activity using both a cell-free activity assay and 3D-PCR targeting genomic DNA. In cytoplasmic and nuclear extracts, cytidine deamination activity was detected irrespective of drug treatment (Supplementary Fig S14), consistent with the expected basal APOBEC activity in HCT116 cancer cells, which express multiple APOBEC3 enzymes – even non-specific nucleases in the nuclear extracts – under normal conditions.

However, this in vitro assay – based on an oligonucleotide substrate and potentially influenced by dysregulated APOBEC or nonspecific nuclease activities – does not reliably reflect the true genome-editing capacity of cellular APOBECs. To probe genomic activity more directly, we performed differential DNA denaturation PCR (3D-PCR) on a defined region of the *MDM2* gene (Stenglein et al, 2010b; Hultquist et al, 2011). A slight decrease in the limiting denaturation temperature was observed in the 5FdUR-treated sample compared with the RTX-treated and untreated controls (Fig 8C), indicating the presence of a small fraction of genomic templates with increased mutation load.

Reviewer 2 point 6. Does MMR repair the U:G mismatches created by cytosine deaminations by APOBEC3s in this system? It is well known that uracils created by AID are subject repair by MMR (PMID: 16894013).

Answer: We thank the reviewer for this insightful question and for the relevant reference. We propose that 5FdUR treatment induces cytosine deamination in our MMR-proficient HCT116 cells as well; however, the elevated C-to-T frequency observed in MMR-deficient cells is absent here (cf. Supplementary Fig. S5). This is most likely because canonical, error-free MMR efficiently repairs U:G mismatches in MMR-proficient cells, in contrast to the MMR-deficient HC116 line.

In the cited study (PMID: 16894013), non-canonical error-prone MMR was shown to repair AID-induced U:G mismatches, resulting in somatic hypermutation. In our case, however, error-free repair is more likely, given the availability of replicative polymerases in S-phase-arrested cells.

This interpretation is further supported by the markedly altered genomic uracil distribution in 5FdUR-treated MMR-proficient cells, which shows a loss of replication-timing dependence – potentially reflecting extensive MMR-mediated repair synthesis around U:G hotspots. Such MMR-dependent changes are not observed upon RTX treatment, where C-to-T frequencies remain low even in MMR-deficient cells.

Reviewer 2 point 7. If the SBS2 signature mutations created as a result of expression of APOBEC3s, why do they not also see SBS13 signature?

Answer: We thank the reviewer for this important question. The appearance of SBS2 and the absence of SBS13 can likely be attributed to the distinct nature of our experimental system compared to the tumor-derived COSMIC signatures. COSMIC mutational signatures represent mutations accumulated during tumor progression, often involving multiple processes including error-prone repair. In contrast, our study examines acute, drug-induced mutagenic events within a 48-hour treatment window. In this short-term setting, the detected C-to-T transitions may originate directly from unrepaired U:G mismatches generated by cytosine deamination, rather than from abasic sites produced during uracil excision repair, especially because the main uracil-DNA glycosylase is inhibited in our cell line model. Consequently, SBS2-type mutations can emerge without the accompanying SBS13 pattern, which is associated with not only APOBEC-induced cytosine deamination, but also with error-prone translesion synthesis (e.g., by REV1) acting at abasic sites.

In summary, this interesting manuscript advances our understanding how and where 5FdUR and RTX cause uracil incorporation in DNA, but the manuscript should be simplified and shortened to make it more understandable and to increase its impact on the field.

Answer: We agree with the reviewer's suggestion that simplifying and shortening the manuscript would improve its clarity and overall impact. Accordingly, the revised version has been substantially streamlined by removing much of the speculative discussion and reformulating overly detailed explanations for greater focus and readability. All these modifications are clearly indicated in the revised manuscript using track changes.

Reviewer #3 (Comments to the Authors (Required)):

1. MANUSCRIPT SUMMARY: This manuscript investigates the differential genomic uracilation patterns and mutagenic potential elicited by two thymidylate synthase (TS) inhibitors-raltitrexed (RTX) and 5-fluoro-2'-deoxyuridine (5FdUR)-in HCT116 colon cancer cells. The study leverages UNG-inhibited and MMR-deficient cellular models to elucidate how these chemotherapeutics affect genomic uracil accumulation, mutational signatures, and downstream cellular consequences.

The authors demonstrate that the two drugs produce distinct uracilation profiles associated with replication timing and genomic features. Notably, 5FdUR, particularly at high doses, induces a C-to-T mutational signature enriched in APOBEC3-related motifs, suggesting involvement of APOBEC3 deaminases. These changes coincide with reduced cytotoxicity, highlighting potential implications for therapeutic resistance.

The work presents a comprehensive multi-omic analysis (U-DNA-Seq, RNA-Seq, and whole genome sequencing) supported by robust computational approaches, such as Segway genome segmentation and U-score quantification.

This study offers a valuable mechanistic advance by linking TS inhibitor-induced uracil incorporation with mutagenesis, gene expression modulation, and DNA repair status-key for understanding treatment efficacy and resistance in colorectal cancer.

The work is suitable for Life Science Alliance due to its relevance, methodological rigor, and translational implications.

2. EVALUATION OF KEY CLAIMS AND SUPPORTING DATA

Main Points and Experimental Support

A. Drug-specific genomic uracilation patterns (Strong support provided)

o The use of Segway for genome segmentation is appropriate and well executed. Distinct segment clusters (e.g., M2 vs. M5) are convincingly shown to respond differentially to RTX and 5FdUR.

o Supplementary Figure S1 and Jaccard analysis support reproducibility across replicates.

o The correlation with replication timing and AT content adds mechanistic insight.

B. Gene-level uracilation and U-score analysis (Moderate-to-strong support provided)

o The U-score is a useful quantitative measure. However, further validation (e.g., correlation with independent uracil-detection methods like Excision-seq or qPCR validation of highly uracilated genes) would strengthen conclusions.

o The statistical testing is appropriate but could benefit from FDR-adjusted p-values to control for multiple comparisons.

C. Induction of C-to-T mutations by 5FdUR (Strong support)

- o Whole-genome sequencing in UNG- and MMR-deficient cells is a major strength. The enrichment of APOBEC-related signatures is well demonstrated.
- o APOBEC3 expression analysis supports mechanistic claims.

D. APOBEC involvement in mutagenicity (Moderate support)

- o While RNA-level data support increased APOBEC3 expression, the conclusions would be bolstered by inclusion of protein-level or activity assays (e.g., deamination assay or western blot of A3B, A3C, A3H).

E. Implications for cytotoxicity and therapeutic resistance (Weak-to-moderate support)

- o Cytotoxicity data are briefly mentioned but should be more fully integrated. Dose-response curves and statistical comparisons between RTX and 5FdUR (with and without MMR) would clarify this key claim.

Response to Reviewer 3's general evaluation:

We sincerely thank the Reviewer for the thorough and insightful evaluation of our manuscript and for recognizing the mechanistic and methodological strengths of our study. We particularly appreciate the positive assessment of our genome segmentation and mutational analyses, as well as the acknowledgment of the study's relevance and rigor.

We have carefully considered all suggestions provided in this general summary and in the detailed comments below. The revised manuscript has been adjusted accordingly to improve clarity, better integrate the cytotoxicity results, and highlight the scope and limitations of our analyses.

MAJOR REVISIONS:

Reviewer 3 point 1. Expand cytotoxicity data:

- o Include cell viability assays comparing RTX and 5FdUR across MMR/UNG status.
- o Include dose-response curves with statistical analysis.

Estimated timeframe: 2-3 weeks

Answer: We thank the Reviewer for this valuable suggestion and for highlighting the importance of the cytotoxicity data. Dose-response analyses directly comparing RTX and 5FdUR treatments in both MMR-proficient and -deficient HCT116 cells were already

included in the original manuscript (Figure 6B). Time dependence has also been addressed and reported in Supplementary Figure S9. All of these data are supported by a detailed statistical evaluation, in which p-values were calculated for all 320 pairwise comparisons using Welch's t-test (Supplementary Figure S9).

To make this more evident, we have now revised the text in the *Results* section to emphasize the time-dependent data and the statistical evaluation, as follows:

"A detailed statistical analysis was also performed to assess the significance of these differences, and p-values are reported for 320 pairwise comparisons (Supplementary Figure S9)."

Additionally, we have updated the legend of Figure 6B to explicitly refer to this supplementary analysis:

"Time-course data and a detailed statistical analysis is provided in Supplementary Figure S9."

We hope these clarifications help ensure that the comprehensive cytotoxicity data and their statistical support are now clearly visible and easily interpretable.

Reviewer 3 point 2. Functional validation of APOBEC3 involvement:

- o Provide protein expression data or enzymatic activity assays for APOBEC3s implicated in mutagenesis.
- o Consider knockdown/overexpression validation to directly link APOBECs to observed mutational signatures.

Estimated timeframe: 4-6 weeks

Answer: We thank the Reviewer for this important point and fully agree that direct functional validation of APOBEC3 involvement strengthens the mechanistic interpretation.

In the revised version, we have included additional protein-level and activity-based evidence supporting APOBEC3 participation in 5FdUR-induced mutagenesis:

- **Protein expression: Western blot analyses of APOBEC3C and APOBEC3D (Figures 9B and 9C)** complement the previously presented APOBEC3B immunocytochemistry (Figure 9A). A modest increase in chromatin-bound A3C was observed specifically in 5FdUR-treated cells, suggesting that A3C contributes to the mutagenic process alongside A3B.
- **Activity assays: We now include APOBEC activity measurements using a fluorescently labelled oligonucleotide** substrate in cell-free extracts (**Supplementary Figure S14**), as well as **3D-PCR analysis** confirming enhanced cytidine deamination in **5FdUR-treated genomic DNA (Figure 8C)**.

Regarding the Reviewer's suggestion to perform knockdown or overexpression experiments, we chose an alternative route proposed by the Editor (based on the

suggestion of Reviewer 2, point 4) to strengthen the mechanistic link between APOBECs and the observed C-to-T mutations. Specifically, we conducted a DNA secondary structure (loop) analysis for clustered and non-clustered C-to-T transitions. This analysis revealed no enrichment for hairpin loops, but a strong clustering tendency, which is characteristic of APOBEC3C activity (Figure S11C).

We consider overexpression studies less informative in our experimental context, as APOBEC3B mRNA is induced by both RTX and 5FdUR treatments, yet only 5FdUR leads to increased genomic cytosine deamination. This suggests that post-transcriptional and even post-translational regulation, rather than transcript abundance alone, determine APOBEC3 mutagenic activity.

Reviewer 3 point 3. Statistical refinement in U-score and DU gene analysis:

- o Apply false discovery rate (FDR) correction to the Welch's t-tests.
- o Include volcano plots or MA plots for clarity.

Estimated timeframe: 1 week

Answer: We thank the reviewer for these constructive suggestions. We have performed false discovery rate (FDR) correction for all Welch's *t*-tests used in the differential uracilation (DU) gene analysis to control for multiple comparisons. This analysis resulted in no significantly differentially uracilated genes, except in the MMR-proficient samples, where we have already observed distinct global uracil distribution in the 5FdUR-treated cells.

Based on these results, we decided to retain the original volcano plots using the raw *p*-values, marking in red those genes that show differences greater than 0.585 log₂ fold change and *p* < 0.05. This choice and its rationale are now clearly stated in both the main text and the corresponding figure legend. *“To identify those genes that are significantly differently uracilated upon the two drug treatments, we calculated U-scores across independent replicates. Log₂(fold change) values were determined for 44000 genes, and raw p-values were obtained using Welch’s two-sample t-tests. To evaluate the significance of drug-specific differences, we also applied FDR correction for multiple testing; however, due to small effect sizes and the large number of tests, none of the comparisons reached statistical significance. Therefore, we present volcano plots using raw p-values for each major gene class (Fig 2C), marking genes with |log₂(fold change)| > 0.585 and p < 0.05, and interpret these results with appropriate caution.”*

To ensure a cautious interpretation, we have also revised the subsequent Gene Set Enrichment Analysis (GSEA): we removed the parts that relied on DU gene groups (cf. section 4 in Table 1 and 2nd half of Table 2) and retained only the three GSEA analyses based on hierarchical lists of differential uracilation (evaluating tendencies; cf. section 3 in Table 1) or on absolute uracilation levels (hierarchical list and top 200 genes; cf. sections 1-2 in Table 1). Notably, this refinement does not affect the previously reported GSEA results shown in Supplementary Figures S3 and S4; however, these figures (figures only, but not the source data) were removed in the revised version to make this

section more concise and focused, consistent with the reviewers' general recommendations.

Reviewer 3 point 4. Clarify uracil quantification and normalization:

- o Explain normalization of enrichment signals in more detail (e.g., rescaling using dot blot values).
- o Clarify how background uracil levels are accounted for.

Estimated timeframe: <1 week

Answer: We thank the reviewer for raising this point. To ensure quantitative comparability across conditions, we normalized the uracil enrichment tracks by rescaling them to the global uracil content determined independently by dot blot (e.g., in RTX samples ~700 uracil per million base pairs, meaning 1,9 million per genome, as listed in the Supplementary Methods Table 1) (Palinkas et al. 2020 eLife). In this approach, the total uracil amount was distributed genome-wide according to the relative enrichment signal intensities obtained from U-DNA-Seq. This procedure allowed us to transform relative enrichment signals into gene-specific quantitative measures, which we termed U-scores (see Supplementary Methods for more details). To improve the clarity of the corresponding Result section, we extend our explanation as follows: “For its calculation, the uracil enrichment tracks were rescaled according to the global uracil content measured previously by dot blot (e.g., in RTX samples ~700 uracil per million base pairs, meaning 1,9 million per genome, as listed in the Supplementary Methods Table 1) [13]. In this way, the absolute uracil amount was proportionally distributed across the genome according to the enrichment signal tracks (see Materials and methods and Supplementary Methods for details). “ In addition, we have corrected grammatical errors and polished the style in the revised manuscript as indicated with change-tracking.”

Regarding the question of background uracil levels, we can distinguish two types. The first is biological background, present in non-treated cells, where 1–2 uracils per million bp accumulate in genomic segments that are clearly distinct from segments enriched in the dramatically higher uracil levels observed in drug-treated cells (cf. genome segmentation in Fig. 1 and Supplementary Fig. S1A). We are confident that this baseline uracilation does not affect the comparison of drug-specific patterns. The second type is a potential technical background introduced by the U-DNA immunoprecipitation procedure. This issue has already been addressed in our previous publication (Palinkas et al., 2020, eLife; see Figure 1–figure supplement 2). Therefore, the differences observed between the two drug-treated samples reflect true biological changes in uracilation rather than artifacts or uncorrected background.

3. ADDITIONAL MINOR REVISIONS

Abstract: The term "cellular responses" is vague; specify e.g., "differential cytotoxicity and mutational profiles."

Introduction: Minor typographical issues (e.g., "quenstions" → "questions").
Figure legends: Some are too brief (e.g., Figures 1-2). Please expand to describe axes, data type, and interpretation.

Terminology: Define "U-score" earlier and more precisely.

Methods: Consider moving some code blocks to a supplementary GitHub repository or Zenodo archive with DOI for reuse and transparency.

Answer to minor revisions:

We thank the reviewer for these helpful suggestions. All typographical and minor grammatical issues have been corrected, and the text has been refined for clarity and conciseness. Figure legends for Figures 1 and 2 were expanded to better describe the axes and data types. The U-score has been introduced earlier in the manuscript (at the end of the Introduction) and described more precisely in the respective session of the Results.

Regarding the suggestion to move code blocks to a public repository, we chose instead to include all relevant code segments in the *Supplementary Methods*. We believe this approach ensures better accessibility and readability for the broader audience of this journal, while still maintaining full transparency and reproducibility.

4. REFEREE CROSS-COMMENTS

I have reviewed the comments provided by Reviewers #1 and #2.

Reviewer #1 provides a thorough summary of the manuscript's key findings and acknowledges the relevance of the drug-specific uracilation and mutational profiles. I agree with their observation that the MMR-dependent difference in APOBEC signatures is intriguing and merits mechanistic clarification. Their request for clearer contextualization of the gene network analyses aligns with my suggestion to statistically refine the U-score and DU gene data. However, I believe that rather than being extraneous, this section adds value if it is streamlined and interpreted more clearly. Their suggestion for a unifying graphical model is a strong one and would help integrate the mechanistic findings.

Reviewer #2 raises valid concerns regarding the complexity of the genome segmentation and uracilation analysis, but I do not support the recommendation to eliminate Pages 5-10 entirely. While the segmentation analysis is computationally dense, it underpins the manuscript's major claims and can be clarified rather than discarded. I agree with Reviewer #2 that functional validation of APOBEC3 activity (e.g., biochemical assay or knockdown models) would substantiate the mechanistic claims, a point I also raised. Their comment regarding the need to reconcile the lack of SBS13 with SBS2 presence is well-taken and should be addressed.

Both reviewers call for clarification of mechanistic links-especially between APOBEC expression and mutational signatures. My own report similarly recommended expanding cytotoxicity profiling, adding APOBEC protein-level validation, and enhancing statistical robustness. These revisions appear to be consistent across reviews.

In summary, while Reviewer #1 and I are largely aligned in assessing the manuscript's strengths and revision needs, Reviewer #2 recommends more drastic structural changes. I would advocate for selective refinement and improved clarity rather than major content removal.

December 18, 2025

RE: Life Science Alliance Manuscript #LSA-2025-03352R

Dr. Angela Békési
Budapest University of Technology and Economics
Faculty of Chemical Technology and Biotechnology
Műgyetem rakpart 3.
Budapest H-1111
Hungary

Dear Dr. Békési,

Thank you for submitting your revised manuscript entitled "The effects of thymidylate synthase inhibitors differ in genomic uracilation and mutagenic potential". We apologise for the delay in communicating our decision due to editor availability issues and delays in securing reviewer comments.

Your revised manuscript was evaluated by two of the original reviewers whose comments are appended below. As you will read, the reviewers are consistent in their views that the revised manuscript is substantially improved and satisfactorily addresses their previous concerns.

In line with the reviewers' evaluation, we would be happy to publish your paper in Life Science Alliance pending final revisions necessary to meet our formatting guidelines. Along with points mentioned below, please tend to the following:

Please confirm if the graph in Figure 6A and S8 is the same. If this is the case, please indicate this information in the legend of both figure panels.

-Please include size information/ladder reference to images of gels.

-We recommend a small modification for your title to, "Effects of thymidylate synthase inhibitors differ in genomic uracilation and mutagenic potential".

-We request the following in the description of methods:

--Please provide primer details for all experiments.

--We thank you for providing all scripts described in the methods in the supplemental section. This section also contains some descriptive details for approaches used in the study. We request that you move the description of methods in the main text from the supplemental section (example for Western blotting and cytidine-deaminase activity measurements in cell -free extracts). Please rename the scripts file as 'Scripts file' and they can remain in the supplemental section.

--We encourage you to provide details for any software used to prepare figures (for example S14A).

-We encourage you to do a complete grammar and spell check. Use of AI-based grammar/language checks is permitted and we encourage you to disclose this in the manuscript.

-Please provide clean manuscript file and supplementary methods file without tracked-changes in .docx file format.

-Please add the X and Bluesky handles of your host institute/organisation as well as your own or/and one of the authors in our system.

-Please add ORCID ID for secondary corresponding author--you should have received instructions on how to do so.

-Please be sure that the authorship listing and order is correct.

-Please remove graphical abstract from main manuscript file.

-Please place Tables at the end of manuscript file or upload them as separate files in .docx or .xlsx file format.

-Please remove Supplementary Materials content table from main manuscript file.

-Please make sure that captions for Supplementary Figures are in consecutive order.

-Please label the Tables in Supplementary Methods file as Supplementary Table 1, 2 etc.

-Please remove Supplementary Figure from Supplementary Methods file and upload it separately. Labeling of the figure should be same as the other Supporting figures (Figure S1,S2 etc.); please include a callout for this figure in main manuscript text.

-Please include callouts for Supplementary Figure S14A-C in manuscript text.

-Please be sure that the authorship listing and order is correct.

LSA now encourages authors to provide a 30-60 second video where the study is briefly explained. We will use these videos on social media to promote the published paper and the presenting author (for examples, see <https://docs.google.com/document/d/1-UWcfbE4pGcDdcgzcmiuJl2XMBJnxKYeqRvLLrLSo8s/edit?usp=sharing>). Corresponding or first-authors are welcome to submit the video. Please submit only one video per manuscript. The video can be emailed to contact@life-science-alliance.org

A. FINAL FILES:

B. MANUSCRIPT ORGANIZATION AND FORMATTING:

Thank you for your attention to these final processing requirements. Please revise and format the manuscript and upload materials as soon as you are able.

Sincerely,

Sarita Hebbar, PhD
Scientific Editor
Life Science Alliance
<http://www.lsjournal.org>

Reviewer #1 (Comments to the Authors (Required)):

The authors made an impressive effort to address the concerns of the reviewers. The extensively edited manuscript and alterations to figure legends and data presentation adequately address the main concerns. The addition of additional protein level and activity data for Apobec3 helps to solidify some of the interesting conclusions and I do not have any further comments.

Reviewer #2 (Comments to the Authors (Required)):

The authors have addressed all my concerns and the manuscript is improved substantially as a result. The expression and involvement of APOBEC3C in response to drug treatment reported in the revised manuscript is novel and makes this a more interesting paper. I have no additional suggestions for improvements of the manuscript.

January 16, 2026

RE: Life Science Alliance Manuscript #LSA-2025-03352RR

Dr. Angela Békési
Budapest University of Technology and Economics
Faculty of Chemical Technology and Biotechnology
Műegyetem rakpart 3.
Budapest H-1111
Hungary

Dear Dr. Békési,

Thank you for promptly responding to our email and submitting your revised research article entitled "Effects of thymidylate synthase inhibitors differ in genomic uracilation and mutagenic potential". It is a pleasure to let you know that your manuscript is now accepted for publication in Life Science Alliance. Congratulations on this interesting work.

DISTRIBUTION OF MATERIALS:

Again, congratulations on a very nice paper. I hope you found the review process to be constructive and are pleased with how the manuscript was handled editorially. We look forward to future exciting submissions from your lab.

Sincerely,

Sarita Hebbar, PhD
Scientific Editor
Life Science Alliance
<http://www.lsajournal.org>